# Measuring Model Robustness via Fisher Information: Spectral Bounds, Theoretical Guarantees, and Practical Algorithms

## Abstract

The robustness of deep neural networks is critical for their deployment in safety-sensitive domains. This paper establishes a novel theoretical framework for quantifying model robustness through the lens of Fisher information. We first start with the known conclusion that maximizing the KL divergence of the posterior probability is equivalent to minimizing half the Mahalanobis distance defined by the Fisher Information Matrix (FIM), and further reveal that the FIM is equal to the variance of the input Jacobian matrix. Based on this insight, we propose the FIM's principal eigenvalue (or its reciprocal) as a principled robustness metric. We derive closed-form spectral bounds for common architectural components (e.g., ReLU, convolution) and theoretically compare the robustness of VGG, ResNet, DenseNet, and Transformer. To enable scalable computation, we resort to efficient algorithms, including power iteration and randomized Hutchinson, to estimate the robustness metric. Furthermore, we propose to use Hutchinson and finite differences to achieve robust estimation in a black-box setting. Extensive experiments validate our theoretical claims and demonstrate the metric's utility in predicting adversarial vulnerability. Code: https://anonymous.4open.science/r/8F4D7E6R/.

## 1 Introduction

As deep learning models are increasingly used in safety-sensitive areas such as autonomous driving Shibly et al. (2023) and medical diagnosis Aggarwal et al. (2021), their robustness to adversarial perturbations has become a critical research topic. Even imperceptible input perturbations can lead to catastrophic prediction errors Zhou et al. (2022), exposing fundamental vulnerabilities of modern neural architectures. Robustness means the ability to maintain consistent performance under input perturbations, including adversarial attacks Carlini & Wagner (2017a), noise, and distribution changes. Therefore, understanding and quantifying robustness is crucial for both theoretical development and practical applications.

**Attack-Dependent Metrics** Existing robustness metrics mainly rely on the strength of adversarial attacks (e.g., bounded perturbations of the $\ell_p$ norm Lin et al. (2020)) or empirical accuracy under attack scenarios Madry et al. (2019). While these heuristics provide practical evaluation benchmarks, they suffer from three major limitations: (i) metrics that rely on attacks cannot reveal the intrinsic properties of the model; (ii) norm-based constraints (adding norm constraints to the loss function) lack probabilistic interpretation; and (iii) most analyses focus on local behaviors without considering global statistical characteristics.

**Heuristic Theoretical Bounds** Although the robustness of models has been demonstrated by estimating the Lipschitz constant Weng et al. (2018) or studied by analyzing input sensitivity, such as the Jacobian norm Sokolic et al. (2016) or gradient variance Agarwal et al. (2022), the connection between these heuristic metrics and the behavior of probabilistic models remains unclear.

To address these deficiencies, we propose a unified information-theoretic framework for robustness evaluation based on the geometry of deep learning input-output manifolds, which has a well-established theoretical foundation and do not depend on any attack algorithm. Our work makes the following **contributions** :

**Theoretical foundation** : We start with the known fact that maximizing the KL divergence of the model posterior probability is approximately equivalent to minimizing half the Mahalanobis distance defined by the Fisher Information Matrix (FIM), analyze the approximation error, and establish the equivalence of the FIM with the variance of the input Jacobian matrix. This bridges the robustness from the geometric (Jacobian) and statistical (FIM) perspectives. Furthermore, we theoretically analyze the relationship between our metric and several classic metrics.

**Practical Metric** : We derive $1/\|F\|_2$ or $\|F\|_2$ as an interpretable and differentiable robustness metric, where $\|F\|_2$ is the largest eigenvalue of the FIM. This avoids attack-specific evaluations and provides a global robustness proof.

**Architecture Analysis** : We characterize the spectral properties of common layers (ReLU, convolution) and theoretically rank the robustness of VGG, ResNet, DenseNet, and Transformer architectures.

**Efficient Algorithms** : We resort to three algorithms including direct eigenvalue decomposition, power iteration, and Hutchinson approximation to handle the estimation of the spectral norm $\lambda_{\max}(F)$ of various scales and guarantee convergence, making it applicable to large-scale models. Furthermore, we propose a new algorithm based on Hutchinson and finite differences to estimate the $\|F\|_2$ value in the black-box setting.

**Application Potential** : Our robustness metric can estimate the robustness of multiple models on the same dataset, and can also compare the volatility of multiple data sets for a model, further helping us use adversarial training to further improve the robustness of the model.

Our experiments verify the correlation of this metric with adversarial vulnerability across datasets (CIFAR-10, MNIST etc.) and demonstrate its practicality in robustness-aware model selection. By unifying geometric sensitivity and probabilistic uncertainty, this work provides a principled toolkit for evaluating and designing robust deep learning (see App. A for more discussion).

## 2 RELATED WORK

### 2.1 ROBUSTNESS METRICS IN DEEP LEARNING

**Adversarial Attack-Based Metrics** Empirical robustness is usually evaluated through adversarial attacks (e.g., PGD Madry et al. (2019) and C&W Carlini & Wagner (2017b)), which create perturbations to induce misclassification. While these methods are effective in exposing vulnerabilities, they are computationally expensive and attack-dependent — their results may not generalize to unknown threat models or real-world noise.

**Lipschitz and Jacobian Norms** Theoretical approaches use Lipschitz continuity Szegedy et al. (2014) or Jacobian matrix norms Sokolic et al. (2016) to bound model sensitivity. However, these methods lack probabilistic interpretation and are difficult to scale to complex architectures (e.g., Transformer) due to fuzzy boundaries or exponential computational complexity.

**Information Theoretic Perspective** KL divergence and mutual information have been used to quantify robustness Alemi et al. (2019), but previous studies have failed to link these metrics to the geometry of the input space. Our work bridges this gap by linking KL divergence to Fisher information, unifying probabilistic and geometric perspectives.

### 2.2 FISHER INFORMATION IN DEEP LEARNING

**Classic Foundations** The Fisher Information Matrix (FIM) is central to statistical estimation and natural gradient descent Amari (1998). In deep learning, it has been used for optimization and uncertainty quantification (e.g., K-FAC Martens & Grosse (2020)), but these studies focus on parameter space properties rather than robustness in the input space.

**FIM for Adversarial Robustness** Recent studies have used FIM for adversarial detection Zhao et al. (2019) or robust training Martin & Elster (2019), but none of them has established a direct relationship between FIM eigenvalues and the inherent robustness of the model. Our key insight—that the largest FIM eigenvalue encodes the worst-case sensitivity—provides a novel, theoretically supported robustness metric.

### 2.3 SPECTRAL ANALYSIS AND EFFICIENT COMPUTATION

**Spectral Methods in Deep Learning** Spectral normalization Miyato et al. (2018) can regulate model complexity, but their applications are mainly limited to generative models. Different from these studies, we analyze the spectral properties of discriminative architectures (e.g., CNN, Transformer) from the perspective of FIM.

**Randomized Algorithms** Hutchinson estimator Hutchinson (1989) and power iteration Golub & Loan (2013) are widely used for large-scale matrix computation. We adapt these algorithms to the special structure of FIM matrices to efficiently estimate $\lambda_{\max}(F)$ with provable convergence, thus enabling scalability to modern architectures.

## 3 METHODOLOGY

### 3.1 PROBLEM FORMULATION

**Robustness as KL-Divergence Maximization** For any model, the cluster of posterior probability distributions of the model output relative to the input $x$ forms a statistical manifold

$$\mathbb{P} = \{p(y|x; \theta)|x \in \mathbb{X}\}, \tag{1}$$

where each input $x$ corresponds to a point on the manifold and $\theta$ is a parameter of the model. In adversarial training, the input sample $x$ is mapped to a point $p(y|x)$ on the manifold by the model, and the perturbation $x \rightarrow x + \delta$ will correspond to a trajectory on the manifold. We try to maximize the distance between two model output points on the manifold

$$x'^* = \arg\max_{x'} \mathcal{D}(p(y|x; \theta), p(y|x'; \theta)), \tag{2}$$

where $\mathcal{D}(\cdot, \cdot)$ represents the distance between the outputs of the two distribution functions.

**Fisher Information and Robustness Metric** For the convenience of discussion, we ignore the model parameter $\theta$. We will introduce the following Theorem 1 as our starting point: The KL divergence between any two conditional distributions $p(y|x)$ and $p(y|x')$ is approximately equal to half of the Mahalanobis distance between $x$ and $x'$, where the covariance parameter matrix is the inverse of the Fisher information matrix (FIM). App. C and D analyze the rationality of the approximation theoretically and experimentally.

**Theorem 1** *For any two conditional distributions $p(y|x)$ and $p(y|x')$, where $x$ and $x'$ are the inputs of the model and $y$ is the class label of the model output, we have*

$$KL(p(y|x), p(y|x')) \approx \frac{1}{2}(x' - x)^T F(x)(x' - x) = \frac{1}{2}\delta^T F(x)\delta, \tag{3}$$

*where $F(x)$ is the Fisher information matrix defined as follows*

$$F(x) = \mathbb{E}_{p(y|x)}[\nabla_x \log p(y|x) \nabla_x \log p(y|x)^T]. \tag{4}$$

$F(x)$ geometrically represents the curvature of the probability distribution manifold at point $x$. From Theorem 1, it is not difficult to see that the perturbation direction $\delta$ ($\|\delta\|_2 = 1$) in adversarial training is approximately equal to the principal eigenvector of the Fisher information matrix. Furthermore, for any deep learning structure, we have the following conclusion (see App. E for proof):

**Theorem 2** *For a deep learning model whose last layer uses a **softmax** function to implement classification tasks, where the input vector of softmax is $f(x)$, the Fisher information matrix is*

$$F(x) = var(J_f(x)), \tag{5}$$

*where $J_f(x)$ is the gradient matrix (Jacobian matrix) of the vector $f(x)$ with respect to the input $x$ and var represents the variance of the matrix random variable.*

Using Theorem 2 and the properties of variance, we immediately get $\frac{1}{2}\delta^T F(x)\delta = \frac{1}{2}\text{var}(\delta^T J_f(x))$. Therefore, the KL divergence also measures the sensitivity of the model output (Jacobian projection)

to the fluctuation of the input in the perturbation direction $\delta$. The experimental results in App. K.1 verify how the variance of the gradient tends to the FIM matrix as the number of samples increases.

Given an input $x$, when $\delta$ is the principal eigenvector of $F(x)$, the KL divergence between the two posterior probabilities is maximum, that is, at this time $\delta$ corresponds to the worst-case perturbation to the model, and $\lambda_{\max}(F(x))$ (or $\|F(x)\|_2$) bounds the worst-case perturbation impact. So for the dataset $S$, we define the following robustness measure based on the spectral norm ($|S|$ represents the number of elements in set $S$):

$$R_{\text{spec}}(S) = \frac{1}{|S|} \sum_{x \in S} \frac{1}{\|F(x)\|_2}, \quad R_{\text{norm}}(S) = \frac{1}{|S|} \sum_{x \in S} \|F(x)\|_2. \tag{6}$$

App. B provides the relationship between our metric and several classical measures and further discussion.

## 3.2 THEORETICAL ANALYSIS

**General Analysis** Given any classifier based on deep learning model, we will discuss how to estimate the upper bound of the spectral norm $\|F(x)\|_2$, where the Fisher information matrix of $x$ for the discrete variable $y$ is defined as ($p_k = p(y_k|x)$)

$$F(x) = \sum_{k=1}^{K} p(y_k|x)[\nabla_x \log p(y_k|x) \nabla_x \log p(y_k|x)^T] = \sum_{k=1}^{K} p_k[\nabla_x \log p_k \nabla_x \log p_k^T]. \tag{7}$$

For any network structures, we further estimate $\|F(x)\|_2$ by the following theorem, where the proof is in App. F.

**Theorem 3** *For any deep network-based classifier $h : x \to softmax(f(x))$, where softmax is the softmax function, the spectral norm $\|F(x)\|_2$ of its Fisher information matrix with respect with the input $x$ has the following upper bound*

$$\|F(x)\|_2 = \lambda_{\max}(F(x)) = \max_{\|v\|_2=1} v^T F(x) v \leq \max_k p_k(1 - p_k)\|J_f(x)\|_2^2, \tag{8}$$

*where $J_f(x)$ is the Jacobian matrix of the output $f(x) \in \mathcal{R}^K$ with respect to the input $x \in \mathcal{R}^d$.*

In a deep neural network, the model $f(x)$ is essentially a composite of m functions

$$f(x) = f_m \circ f_{m-1} \circ \cdots \circ f_1(x), \tag{9}$$

where the Jacobian matrix of each function $f_i : \mathbb{R}^{n_i} \to \mathbb{R}^{n_i+1}$ is $J_i$, then we have (by chain rule) $\frac{\partial f}{\partial x} = J_m J_{m-1} \cdots J_1$. Then, according to the property of the norm $\|AB\|_2 \leq \|A\|_2 \|B\|_2$, we immediately have $\|J_f\|_2 \leq \prod_{i=1}^{m} \|J_i\|_2$. Finally, we get

$$\|F(x)\|_2 \leq \max_k p_k(1 - p_k) \prod_{i=1}^{m} \|J_i\|_2^2. \tag{10}$$

Therefore, the spectral norm analysis of deep network models can be reduced to the analysis of its basic components.

**Spectral Norm $\|J_i\|_2$ of Basic Components** We theoretically analyze the upper bounds of the spectral norms of the basic components of deep neural networks in Table 1 (see App. G for details). We can see that 1) The spectral norm of ReLU and Max Pooling is strictly 1, indicating that they have equidistant propagation of input perturbations; 2) The spectral return of Average Pooling decreases as the kernel increases, which has a certain gradient smoothing effect; 3) BN and LN can actively amplify or suppress perturbations through scaling factors; 4) When the spectral norm of Softmax is close to 0, it may suppress the propagation of perturbations, and the spectral norm of the concatenation operation is proportional to the sum of the squares of the spectral norms of the input tensor, which may implicitly introduce gradient expansion; 5) The spectral norm of the linear change layer (convolution or full connection) is the main source of perturbation amplification.

**Analysis of Deep Neural Networks** We analyzed the following four classic deep network structures, including VGG, Densenet, Resnet and Transformer (ViT), and the specific results are as shown in

Table 1: Spectral norm of the basic components

| Name | Function | $\|J\|_2$ |
|---|---|---|
| ReLU | $\max(0, x)$ | $= 1$ |
| Max Pooling | $\max_{(m,n) \in N_k(i,j)} x_{m,n,c}$ | $= 1$ |
| Average Pooling | $\frac{1}{k^2} \sum_{(m,n) \in N_k(i,j)} x_{m,n,c}$ | $= \frac{1}{k}$ |
| Convolutional | $\sum_{i=0}^{k-1} \sum_{j=0}^{k-1} \sum_{c=1}^{C_{\text{in}}} W_{i,j,c,c'} X_{h'+i,w'+j,c}$ | $\approx \|W\|_2$ |
| Fully Connected | $Wx + b$ | $= \|W\|_2$ |
| Batch Normalization | $\gamma^{(c)} \frac{x^{(c)} - \mu^{(c)}}{\sqrt{(\sigma^{(c)})^2 + \epsilon}} + \beta^{(c)}$ | $= \max_c \frac{|\gamma^{(c)}|}{\sqrt{(\sigma^{(c)})^2 + \epsilon}}$ |
| Layer Normalization | $\gamma \odot \frac{x - \mu}{\sqrt{\sigma^2 + \epsilon}} + \beta$ | $\leq \max_i \frac{|\gamma^{(i)}|}{\sqrt{\sigma^2 + \epsilon}}$ |
| Softmax | $\sigma(x)_i = \frac{e^{x_i}}{\sum_{j=1}^n e^{x_j}}$ | $\leq 2 \max_k \sigma(x)_k (1 - \sigma(x)_k)$ |
| Concatenation | $[X_1 \quad \cdots \quad X_n]$ | $\leq \sqrt{\sum_{i=1}^n \|X_i\|_2^2}$ |

Table 2: Analysis of spectral norm of deep network structure ($h$ is the number of attention heads)

| DNN | Estimation of Upper bound of $\|J\|_2$ | Structural complexity |
|---|---|---|
| VGG | $\prod_{i=1}^L \|W_i\|_2 \cdot \prod_{j=1}^M \|U_j\|_2$ | $O(L + M)$ |
| ResNet | $\frac{1}{2} \|W_{\text{cov}}\|_2 \prod_{l=1}^L (1 + \|W_{l,1}\|_2 \|W_{l,2}\|_2) \|U\|_2$ | $O(L)$ |
| DenseNet | $\|W_L\|_2 \prod_{k=1}^{L-1} (1 + \|W_k\|_2)$ | $O(L)$ |
| Transformer | $\prod_{l=1}^L (1 + \sqrt{h} \max_i \|W_i^V\|_2 \|W^O\|_2 + \|W_{l1}\|_2 \|W_{l2}\|_2)$ | $O(L)$ |

Table 2 (see App. H for more details). In Table 1, since the spectral norm of the linear change layer (convolution or fully connected) is the main source of perturbation amplification, we only estimate the upper bound in the form of the spectral norm of the convolution or fully connected layer.

Analyzing the results in Table 2, we conclude that: 1) The spectral norm of VGG is the product of the spectral norms of each layer, which grows exponentially with the network depth; 2) The introduction of the residual structure in Resnet makes the $(1 + \|W_{l,1}\|_2 \|W_{l,2}\|_2)$ term make the upper bound grow linearly, especially when $\|W_{l,1}\|_2 \|W_{l,2}\|_2 < 1$; 3) DenseNet has a linear growth similar to Resnet, but has more cross-layer links and the network weights of each layer are reused; 4) The terms referenced by the attention mechanism in Transformer may significantly increase the upper bound.

We take the most classic models among the four models to compare their structural complexity: VGG16 ($L = 13, M = 3$), DenseNet121 ($L = 59$), ResNet18 ($L = 12$), ViT-B-16 ($L = 12$). Therefore, we roughly conclude that the robustness ranking of the models is

$$\text{DenseNet121} < \text{VGG16} < \text{ResNet18} \leq \text{ViT-B-16}. \tag{11}$$

### 3.3 PRACTICAL ALGORITHMS WITH WHITE-BOX SETTINGS

Since in the theoretical analysis, we only approximately estimated the upper bound of the model, ignoring the actual spectral norm values of each component, and we also ignored the fact that the spectral norm also depends on the input of the model. Therefore, below we will evaluate the robustness of the model on a certain data set by solving the spectral norm of $F(x)$.

Let $q_k = \nabla_x \log p(y_k|x)$ and $\lambda_k = p(y_k|x)$, we can write the Fisher information matrix in a more compressed form

$$F(x) = Q \Lambda Q^T, \tag{12}$$

where $Q = [q_1 \quad q_2 \quad \cdots \quad q_K]$ and $\Lambda = \text{diag}(\lambda_1, \lambda_2, \cdots, \lambda_K)$.

**Direct Eigendecomposition** Considering the properties of the spectral norm $\|AB\|_2 = \|BA\|_2$, we can get $\|F(x)\|_2 = \|P\|_2$, where $P = \Lambda^{1/2} Q^T Q \Lambda^{1/2}$ is a symmetric matrix. The time complexity and space complexity of solving $\|P\|_2$ directly through eigenvalue decomposition are $O(dK^2 + K^3)$ and $O(dK)$ respectively, which is suitable for cases where $K$ is small.

**Power Iteration** The power iteration algorithm (as shown in Algorithm 1 of App. J) is a simple algorithm for finding the leading eigenvalue of a matrix and its associated eigenvector. Although the most time-consuming operation of the algorithm is the matrix multiplication, the matrix $F(x)$ has a special form of eigenvalue decomposition, and we can calculate $F(x)b_t$ very efficiently. Note that the initial value is set to the approximate value to accelerate the iterative algorithm's convergence process. Due to the special structure of $Q\Lambda Q^T$, we can obtain the time complexity and space complexity of the power iteration algorithm as $O(TdK)$ and $O(dK)$ respectively. Note that when the iteration error $\|\lambda_t - \lambda_{\text{prev}}\|)/\|\lambda_t\|_2 < \epsilon$, where $\epsilon$ is a given threshold, the algorithm will exit midway.

**Hutchinson Approximation Algorithm** We adopt Hutchinson algorithm (as shown in Alg. 2 of the App. J) Hutchinson (1989) to estimate the principal eigenvalue of the matrix $\lambda_{\max}$

$$\|F(x)\|_2 = \lambda_{\max}(F(x)) \approx \max_i \frac{z_i^T F(x) z_i}{z_i^T z_i}, \tag{13}$$

where $z_i$ is a random vector (such as a Rademacher vector with elements of $\pm 1$) or a Gaussian variable.

**Theorem 4** *Hutchinson (1989) Let* $R(A, x_i) = \frac{x_i^T A x_i}{x_i^T x_i}$, *given* $M$ *independent random vectors* $x_1, \cdots, x_M$ *(Rademacher vectors or Gaussian variables), when* $M \to \infty$, *then* $\hat{\lambda}_{\max} = \max_{i=1}^m R(A, x_i)$ *will converge to* $\lambda_{\max}(A)$ *with high probability. For any given* $\delta$ *value, when*

$$M \geq \frac{\log \frac{1}{\delta}}{p_\epsilon}, \tag{14}$$

*then*

$$P(\hat{\lambda}_{\max} \geq \lambda(A) - \epsilon) = 1 - (1 - p_\epsilon)^M \geq 1 - \delta, \tag{15}$$

*where* $p_\epsilon = P(R(A, x_i) \geq \lambda_{\max}(A) - \epsilon)$.

Theorem 2 shows that even if the probability $p_\epsilon$ of a single sample falling into the target interval is very low, we can still ensure high probability convergence to $\lambda_{\max}(A)$ by moderately increasing $M$.

**Theorem 5** *Hutchinson (1989) Let* $u, v \in \mathbb{R}^n$, *where* $u$ *is a random unit vector and* $v$ *is a fixed unit vector (such as the principal eigenvector of a matrix), then the probability that* $u$ *is aligned with* $v$ *decays exponentially with* $n$. *Specifically, we have*

$$P(|u^T v| \geq t) \leq 2 \exp\left(-cnt^2\right), \tag{16}$$

*where* $c$ *is a universal constant.*

Theorem 5 shows that when the dimension of the random vector grows, the probability of the random vector aligning with $\lambda_{\max}$ will decay exponentially. If the random vector generated by $F(x) = Q\Lambda Q^T$ as the input of Hutchinson is in a high-dimensional space of $d$ dimensions, then the probability of it aligning with the spectral norm will be very low. Therefore, as with direct eigenvalue decomposition, we also consider using $P = \Lambda^{1/2} Q^T Q \Lambda^{1/2}$ as the input of Hutchinson. The time complexity of Hutchinson algorithm for calculating the spectral norm of FIM is $O(MdK)$, and Hutchinson algorithm can be highly parallelized since each random vector is independent of each other.

The theoretical analysis in Appendix J and experimental verification in Appendix K show that we can significantly reduce the space complexity and approximation error of the model by indirectly estimating $\|F\|_2$ through $P$.

### 3.4 PRACTICAL ALGORITHMS WITH BLACK-BOX SETTINGS

Below we will use Hutchinson's algorithm and finite differences to estimate the robustness measure $\|F(x)\|_2$ in a black-box setting.

For any Gaussian random vector $v \sim N(0, I)$, the directional derivative of the gradient $\nabla_x \log p(y|x)$ can be approximated by symmetric difference ($u = v/\|v\|_2$)

$$u^T \nabla_x \log p(y|x) \approx \frac{\log p(y|x + hu) - \log p(y|x - hu)}{2h}, \tag{17}$$

where $h$ is a small positive constant (such as $10^{-3}$). The quadratic form $u^T F(x) u$ of FIM can be decomposed into

$$u^T F(x) u = u^T E_{p(y|x)}[\nabla_x \log p(y|x) \nabla_x \log p(y|x)^T] u = E_{p(y|x)}[(u^T \nabla_x \log p(y|x))^2], \quad (18)$$

where $u^T \nabla_x \log p(y|x)$ can be estimated using first-order finite differences (Eqn. (17)).

## 4 EXPERIMENTS

### 4.1 DATASETS AND SETTINGS

To estimate the robustness of the model, we use the basic models of four classic models, including VGG16, ResNet18, DenseNet121, and ViT_B_16, and train them on three different styles of datasets (CIFAR10, MNIST, and Tiny-ImageNet). For unified processing, the images in the three datasets are resized to 224×224 size images during training. The optimizer uses the AdamW optimizer in PyTorch, where the learning rate is uniformly set to 3e-5. The model obtained by using only the training set (without using any pre-trained model) for all models is called the clean model $M_{\text{clean}}$. Subsequently, the model we obtain through the two adversarial training algorithms, CW or PGD, is the adversarial model, denoted as $M_{\text{CW}}$ or $M_{\text{PGD}}$. We also validate the effectiveness of our metrics on large-scale datasets like CIFAR100, ImageNet, and special types of data such as medical data [1].

### 4.2 EVALUATION METRICS AND SETTINGS

Assume that the model $M$ is tested on the test set $D$, and $a(x)$ represents the perturbation sample generated by the clean input $x$. We will mainly use the spectral norm robustness $\|F(x)\|_2$, Lipschitz constant, CLEVER score, and robustness metrics based on adversarial attacks including PGD Madry et al. (2018) and C&W Carlini & Wagner (2017a).

**PGD and CW** Below we introduce the two metrics PGD and CW, which are two classic adversarial attack methods. We often use the attack success rate under PGD and CW attacks as an indicator to evaluate the robustness of the model, where the attack success rate (ASR) is defined as follows:

$$\text{ASR} = \frac{|\{(x,y)|M(a(x)) \neq y, (x,y) \in D\}|}{|\{(x,y)|M(x) = y, (x,y) \in D\}|}. \quad (19)$$

In the experiments, we use **torchattacks** [2] to calculate PGD and CW values. In PGD, the maximum perturbation $\epsilon$ is set to $8/255$, the step size $\alpha$ is $2/255$, the number of attack steps is 20 and random initialization is performed. CW uses the following parameters: box constraint parameter $c = 1$, confidence $\kappa = 0$, the number of attack steps is 20 and the learning rate $\text{lr} = 0.01$ of the Adam optimizer. It is worth noting that PGD contains random factors, while CW does not contain randomness.

**CLEVER score** The maximum perturbation radius in the CLEVER algorithm is set to 0.1, and the distance norm in the neighborhood definition and the norm in the gradient both use the 2-norm. When the CLEVER algorithm estimates the Lipschitz constant at each data point $x$, 100 points are sampled in the neighborhood of point $x$ to find the maximum value of the gradient norm.

$R_{\text{spec}}$ **and Lipschitz constant** We approximate the Lipschitz constant of the model $f(x)$ by the gradient at point $x$, where the gradient is implemented by automatic differentiation in pytorch. When calculating the robustness based on the spectral norm, we also count the average value of $\|F(x)\|_2$ and the average value of $1/\|F(x)\|_2$. The former is positively correlated with other metrics, while the latter corresponds to $R_{\text{spec}}$ and is negatively correlated with other metrics.

### 4.3 REASONABLENESS OF OUR ROBUSTNESS METRIC

We use the clean model $M_{\text{clean}}$ (ResNet18) as the benchmark and use CW adversarial training to obtain a model $M_{\text{CW}}$. Based on our intuition, $M_{\text{CW}}$ should be more robust than $M_{\text{clean}}$. Since the

---

[1]https://www.kaggle.com/datasets/tawsifurrahman/covid19-radiography-database
[2]https://adversarial-attacks-pytorch.readthedocs.io/en/latest/attacks.htmlmodule-torchattacks.attacks.pgd

Lipschitz constant $L(x)$, CLEVER, CW, PGD and spectral norm $\|F(x)\|_2$ are positively correlated, the value of $M_{\text{CW}}$ on the above indicators should be smaller than the corresponding value of $M$. We counted the percentage reduction of the metric on the model $M_{\text{CW}}$ compared to the metric on $M_{\text{clean}}$, and the results are shown in Table 3 with 500 samples.

As can be seen from Table 3, the reduction value of our spectral norm metric $\|F(x)\|_2$ is very close to the CW estimate. It is worth noting that the attack success rate on PGD decreases the least, because we use CW to perform adversarial attacks in training, but use PGD to implement attacks in testing, which shows that the CW metric is not transferable. At the same time, the estimated values of $L(x)$ and CLEVER are relatively close.

Table 3: Robustness comparison using adversarial training model $M_{\text{CW}}$ and clean model $M_{\text{clean}}$

| Model | $L(x)$ | CLEVER | CW | PGD | $R_{\text{norm}}$ | $R_{\text{spec}}$ |
|---|---|---|---|---|---|---|
| None-Attack ($M_{\text{clean}}$) | 0.50 | 3.52 | 93.64 | 99.24 | 2.38 | 46.46 |
| CW-Attack ($M_{\text{CW}}$) | 0.29 | 2.02 | 29.60 | 86.67 | 0.82 | 186.46 |
| Reduction (%) | 42.00 | 42.61 | 68.39 | 12.67 | 65.55 | - |

Comparing the results in Table 3, we can see that the estimated values of $\|F(x)\|_2$, $L(x)$, and CLEVER are very stable when the size of the data set changes, while the fluctuations of CW and PGD are relatively large. This is because CW and PGD are essentially discrete functions of the input $x$, where accuracy functions are not differentiable with respect to the input.

## 4.4 ROBUSTNESS OF DIFFERENT MODELS ON THE SAME DATASET

We use CIFAR10 as a benchmark to analyze how the six metrics rank the models (as shown in Table 4). We sort the four metrics in descending order of $L(x)$, and we can see that our spectral norm $\|F(x)\|_2$ obtains the same ranking results as $L(x)$ and CLEVER, while the results of CW are exactly the same as our $R_{\text{spec}}$. This shows that the two metrics $\|F(x)\|_2$ and $R_{\text{spec}}$ we proposed can replace CLEVER and CW respectively to some extent. PGD uses different attack methods in training and testing, so the results are not referenceable (See App. K.4 for more comparisons on large-scale datasets).

Table 4: Comparison of ranking results of 4 models on 6 metrics on the CIFAR10 dataset

| Models | $L(x)$ | CLEVER | CW | PGD | $R_{\text{norm}}$ | $R_{\text{spec}}$ |
|---|---|---|---|---|---|---|
| DenseNet121 | 0.47 | 2.93 | 54.55 | 94.81 | 2.18 | 5.16 |
| ResNet18 | 0.29 | 1.99 | 22.97 | 89.19 | 0.77 | 124.61 |
| ViT_B_16 | 0.25 | 1.35 | 39.39 | 96.97 | 0.61 | 77.36 |
| VGG16 | 0.07 | 1.11 | 14.29 | 55.95 | 0.09 | 97685.6 |

## 4.5 ROBUSTNESS OF THE SAME MODEL ON DIFFERENT DATASETS

Comparing the robustness of the same model across multiple datasets (in Tab. 25) shows that our metrics and other metrics produce consistent results for Medical Data and CIFAR100: CIFAR100 > Medical Data. However, the data distribution in ImageNet varies significantly, leading to inconsistent results when compared with other datasets. The results show that ImageNet is as difficult to attack as CIFAR100.

Table 5: Comparison of robustness ranking results of ResNet18 using 6 metrics on 3 datasets

| Dataset | $L(x)$ | CLEVER | CW | PGD | $R_{\text{norm}} \downarrow$ | $R_{\text{spec}}$ |
|---|---|---|---|---|---|---|
| Medical Data | 0.57 | 5.43 | 37.08 | 98.88 | 5.95 | 36.28 |
| ImageNet | 0.17 | 2.29 | 95.24 | 100.0 | 1.11 | 1.44 |
| CIFAR100 | 0.29 | 1.81 | 62.07 | 94.83 | 0.73 | 5.69 |

## 4.6 ROBUSTNESS COMPARISON BETWEEN BLACK-BOX SETTING AND WHITE-BOX SETTING

To compare the robustness metrics under two different settings, we use only the model output $p(y_k|x)$ in the black-box setting and explicitly use the model gradient $\nabla \log p(y_k|x)$ in the white-box setting. The results for both settings are shown in Table 6. We can draw the following conclusions: 1) Since the dimensionality of the vectors in the black-box setting is higher than that in the white-box setting (data comparison coefficient), and since we indirectly compute the matrix $P$ in the white-box setting, the estimated eigenvalues are an order of magnitude lower than those in the white-box setting; 2) Our metrics yield consistent conclusions in both settings: for the ResNet-18 model, the robustness comparison across datasets is CIFAR100 > Medical Data.

Table 6: Comparison of robustness ranking results of ResNet18 using 6 metrics on 3 datasets

| Dataset | $R_{\mathrm{norm}}$(white) | $\hat{R}_{\mathrm{norm}}$(black) |
|---|---|---|
| Medical Data | 5.95 | 0.0056 |
| CIFAR100 | 0.73 | 0.0032 |

## 4.7 COMPARISON OF RUNNING TIMES

We use the adversarial training model $M_{\mathrm{CW}}$ on CIFAR100 to test 5 metrics and run them 5 times on 500 samples to calculate the average running time of the 5 metrics, as shown in Tab. 7. Since CLEVER greatly approximates the gradient of the loss function, the maximum eigenvalue of the gradient can be easily solved, so it has the fastest running time. Our $R_{\mathrm{norm}}$ and Lipschitz constant $L(x)$ are both based on the gradient of the model, but $R_{\mathrm{norm}}$ calculate the spectral norm of $F(x)$ instead of the spectral norm of the gradient, which takes more time than the estimation of $L(x)$. Although we can achieve fast estimation of $\|F(x)\|_2$ through parallel sampling of the Hutchinson algorithm, **due to the limitations of the GPU memory** , we have to convert large-scale batch sampling into multiple batches of small-scale sampling, which makes our algorithm slightly slower.

Table 7: Comparison of running times of two models on CIFAR100 with multiple metrics

| Model | $L(x)$ | CLEVER | CW | PGD | $R_{\mathrm{norm}}$(white) | $\hat{R}_{\mathrm{norm}}$(black) |
|---|---|---|---|---|---|---|
| ResNet18 | 131.09 | 24.65 | 96.12 | 83.48 | 267.13 | 66.16 |
| ViT_B_16 | 494.74 | 41.19 | 172.08 | 233.70 | 309.73 | 379.22 |

## 5 CONCLUSION

This paper proposes a unified information-theoretic framework to quantify the robustness of deep neural networks using Fisher information. Building on the connection between the KL divergence of the posterior probability and the Fisher Information Matrix (FIM), we propose the maximum eigenvalue of the FIM, or its inverse, as a principled and interpretable robustness metric. We analyze the connections and differences between our metric and several classical metrics. We further analyze upper bounds on the spectral norms of common architectural components (e.g., ReLU and convolution) and compare the robustness of popular architectures including VGG, ResNet, DenseNet, and Transformer. To achieve scalable computation, we use three algorithms to compute the spectral norm of the FIM, making it applicable to scenarios of various scales. Furthermore, we propose a new algorithm that implements robustness estimation in the black-box setting with the Hutchinson algorithm and finite differences. Extensive experiments on datasets of varying sizes and types validate our theoretical results. Overall, our metric is well-founded, independent of attack algorithms, and applicable to both white-box and black-box settings.

However, FIM is data-dependent, which means that robustness evaluation may vary for different test sets or input domains, and comparisons across data distributions remain challenging, which will be our future work. Despite these limitations, our framework lays the foundation for a more rigorous understanding of deep learning robustness, paving the way for future work on robust model design and evaluation.

## ETHICS STATEMENT

Potential risks and mitigations include:

- **Misuse in adversarial settings:** While our metrics are useful for evaluating robustness, malicious actors may exploit insights from the Jacobian or FIM properties to design more powerful attacks. We mitigate this by focusing on defensive applications and encouraging transparency in robustness benchmarks.
- **Over-reliance on theoretical guarantees:** While principled, our bounds and metrics are not exhaustive (e.g., they may not cover all perturbation types). We emphasize that our approach should complement rather than replace empirical testing and highlight the need for multifaceted robustness evaluation.
- **Computational cost:** Despite the efficiency of our algorithm, estimating the Fisher spectrum for very large models may still be resource-intensive. We provide guidance on the trade-off between accuracy and computational overhead of robustness estimation.

## REPRODUCIBILITY STATEMENT

All experiments were performed on a GeForce RTX 3090 with 24 GB video memory to fairly compare the performance and running time of all algorithms. The datasets used in our experiments are all publicly available datasets on the Internet, including commonly used datasets in computer vision. For datasets from uncommon sources such as medical data, we provide links to the data. We performed a simple normalization on the images following the conventional normalization method for images in the field of image classification. For details, see the anonymous code. All experiments ensure the reproducibility of the results by fixing the random seed, including model initialization and data generation, for verifying the theoretical results of the theorem. For specific code, please see the link: https://anonymous.4open.science/r/8F4D7E6R/.

## THE USE OF LARGE LANGUAGE MODELS

In this work, large language models (LLMs) were primarily used as a general-purpose assist tool to aid and polish the writing of the manuscript. LLMs were not involved in research ideation, experimental design, data analysis, or the generation of any novel scientific content.

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

**We know that** $\|F(x)\|_2 = \lambda_{\max}(F(x))$**, but for the convenience of description, we use both the spectral norm** $\|F(x)\|_2$ **and the maximum eigenvalue** $\lambda_{\max}(F(x))$ **in the text.**

## A   BROADER IMPACTS

This work advances the theoretical understanding and practical evaluation of model robustness and will have impacts in multiple areas:

- **Safety-critical applications:** By providing a principled metric to quantify robustness that does not rely on adversarial attacks, our framework can help design more reliable models for high-risk applications (e.g., autonomous systems, healthcare, and finance). Improved robustness metrics may help reduce the risk of catastrophic failures caused by adversarial perturbations or distribution shifts.

- **Transparency and interpretability:** Our theoretical connections between Fisher information, Jacobian variance, and robustness provide interpretable insights into model behavior. This is in line with the growing demand for explainable AI, especially in regulated industries where understanding model vulnerabilities is critical for certification and deployment.

- **Model selection and benchmarking:** The proposed metric $1/\lambda_{\max}(F(x))$ provides an interpretable tool for comparing different architectures (e.g., VGG vs. Transformer) and selecting models with inherent robustness, reducing reliance on empirical adversarial testing.

- **Efficiency of robustness evaluation:** The scalable algorithms (e.g., power iteration, Hutchinson approximation) enable efficient robustness evaluation of large models, reducing the computational barrier compared to attack-based evaluation. This can make robustness testing more accessible to resource-constrained researchers and practitioners.

By combining theoretical guarantees with practical tools, this work contributes to the broader goal of building trustworthy AI systems. We hope that our framework will inspire further research to unify geometric and probabilistic perspectives on robustness analysis.

## B   CONNECTIONS AND DIFFERENCES WITH OTHER WORK

### B.1   SPECTRAL NORM OF FIM AND LIPCHITZ CONSTANT

We define the Lipschitz constant $L(x)$ in the neighborhood $B_2(x, r) = \{y \mid \|y - x\|_2 < r\}$ of point $x$: Suppose function $f : \mathbb{R}^n \to \mathbb{R}^m$, for a neighborhood of point $x \in \mathbb{R}^n$, if there exists a constant $L(x) > 0$ such that $y, z \in B(x, r)$, then

$$\|f(y) - f(z)\| \leq L(x)\|y - z\|. \tag{20}$$

For a differentiable function $f$, according to the mean value theorem, for any $y, z \in B_2(x, r)$, there exists $\xi$ on the line connecting $y$ and $z$ such that

$$f(y) - f(z) = \nabla f(\xi)^T (y - z). \tag{21}$$

According to the properties of the spectral norm, we have

$$\|f(y) - f(z)\|_2 \leq \|\nabla f(\xi)\|_2 \|y - z\|_2 \leq \sup_{\xi \in B_2(x,r)} \|\nabla f(\xi)\|_2 \|y - z\|_2. \tag{22}$$

By the definition of local Lipschitz continuity, the Lipschitz constant $L(x)$ at a point $x$ is

$$L(x) = \sup_{\xi \in B_2(x,r)} \|\nabla f(\xi)\|_2. \tag{23}$$

Let $J_f(x) = \nabla f(x)$, then by Eqn. (69), we have

$$F(x) \leq \|B\|_2 \|J_f(x)\|_2^2 \leq \|B\|_2 \left( \sup_{\xi \in B_2(x,r)} \|\nabla f(\xi)\|_2 \right)^2 = \|B\|_2 L(x)^2. \tag{24}$$

where $B = \text{diag}(p) - pp^T$ and $L(x)$ is the the Lipschitz constant.

## B.2 SPECTRAL NORM OF FIM AND CLEVER SCORE

In the CLEVER algorithm Weng et al. (2018) (Algorithm 1), we assume that the classifier output is $f(x)$, then the probability output $p(y|x) = \text{softmax}(f(x))$. Let the true category of sample $x$ be $j$, and the category predicted by the model be $c$, then we can define the function

$$g(x) = f_c(x) - f_j(x). \tag{25}$$

Next we calculate the posterior probability of the class $y$

$$p(y|x) = \frac{e^{f_y(x)}}{\sum_i e^{f_i(x)}} = \frac{e^{f_y(x) - \max_k f_k(x)}}{\sum_i e^{f_i(x) - \max_k f_k(x)}} = \frac{e^{f_y(x) - f_c(x)}}{\sum_i e^{f_i(x) - f_c(x)}}. \tag{26}$$

When $f_c(x) \gg f_i(x), i \neq c$, then we have $\sum_i e^{f_i(x) - f_c(x)} \approx 1$, we can approximately calculate $p(j|x)$

$$p(j|x) \approx e^{f_j(x) - f_c(x)} = e^{-g(x)}, \quad p(c|x) \approx 1. \tag{27}$$

So the cross entropy loss at a point $x$ is $-\log p(j|x) \approx g(x)$. Therefore, the CLEVER algorithm approximates the gradient norm of the cross entropy loss function with respect to the input $\|\nabla g(x)\|$, which is the CLEVER score of the point $x$.

In practice, the CLEVER algorithm calculates the Lipchitz constant $L(x)$ of the cross entropy loss at point $x$ according to Eqn. (23) by sampling points in the neighborhood $\mathcal{N}_p(x)$ (defined with $p$-norm) of $x$ ($1/q + 1/q = 1$)

$$L(x) = \max_{z \in \mathcal{N}_p(x)} \|\nabla g(z)\|_q \approx \|\nabla g(x)\|_q, \tag{28}$$

Usually we take $p = q = 2$.

When the loss function optimizes the model, it will cause the posterior probability of the true label $p(j|x)$ to be as large as possible, so $p(j|x)$ will be equal to $p(c|x)$ or its value is second only to $p(c|x)$, so we only consider the two terms in FIM (notice that $p(j|x) \approx e^{-g(x)}$ and $p(c|x) \approx 1$)

$$\begin{aligned}
F(x) &= \sum_{y=1}^{K} p(y|x) \left[ \nabla \log p(y|x) \nabla \log p(y|x)^T \right] \\
&\approx p(j|x) \nabla \log p(j|x) \nabla \log p(j|x)^T + p(c|x) \nabla \log p(c|x) \nabla \log p(c|x)^T \\
&\approx e^{-g(x)} \nabla g(x) \nabla g(x)^T. 
\end{aligned} \tag{29}$$

At this time, the principal eigenvector of $F(x)$ is $\nabla g(x)$, and the maximum eigenvalue is the Rayleigh Quotient

$$\|F(x)\|_2 = \lambda_{\max}(F(x)) \approx \frac{e^{-g(x)} \nabla g^T(x) \nabla g(x) \nabla g(x)^T \nabla g(x)}{\nabla g(x)^T \nabla g(x)} = e^{-g(x)} \|\nabla g(x)\|_2^2, \tag{30}$$

where $\|\nabla g(x)\|_2$ is an approximate estimate of the CLEVER score.

## B.3 SPECTRAL NORM OF FIM AND RANDOMIZED SMOOTHING ALGORITHM

The randomized smoothing algorithm Cohen et al. (2019) explicitly assumes that the perturbation noise follows a Gaussian distribution $\epsilon \sim N(0, \sigma^2 I)$ (see Theorem 1)

$$p(\epsilon) \propto \exp \left\{ -\frac{\|\epsilon\|_2^2}{\sigma^2} \right\} \tag{31}$$

This assumption allows the authors to devise adversarial attacks against the $l_2$ norm. Furthermore, we can establish a connection between $l_\infty$-norm attacks and the multivariate uniform distribution, and between $l_1$-norm attacks and the Laplace distribution.

Thus, **the use of randomized smoothing relies on the assumption of the perturbed probability distribution (Gaussian distribution)**, which generally works better against adversarial attacks on the $l_2$-norm.

We now present the relationship between random smoothing and our FIM-based metric $\|F(x)\|_2$ : In the random smoothing method, the certification radius is defined as follows ($\Phi^{-1}$ is the inverse CDF of the standard normal distribution)

$$r = \frac{\sigma}{2}[\Phi^{-1}(p_A) - \Phi^{-1}(p_B)], \quad p_A = P(f(x+\delta) = c_A), \quad p_B = \max_{c \neq c_A} P(f(x+\delta) = c) \quad (32)$$

We present the proof as follows:

**Probability Difference** : Let $p_A = p(y|x)$ and $p_B = p(y|x+\delta)$, according to Pinsker's inequality, we have

$$p_A - p_B \leq \sqrt{\frac{1}{2}D_{KL}(p_A\|p_B)} \quad (33)$$

Since the sqrt function is a concave function, using the Jensen inequality we have

$$E_\delta(p_A - p_B) \leq E_\delta\sqrt{\frac{1}{2}D_{KL}(p_A\|p_B)} \leq \sqrt{\frac{1}{2}E_\delta D_{KL}(p_A\|p_B)}. \quad (34)$$

For Gaussian perturbations $\delta \sim \mathbb{N}(0, \sigma^2 I)$, we have approximately

$$E_\delta[D_{KL}(p_A\|p_B)] \approx \frac{\sigma^2}{2}\|F(x)\|_2. \quad (35)$$

**Taylor Expansion** : By Taylor expansion of the inverse CDF at point $p = 0.5$, we have

$$\Phi^{-1}(p_A) - \Phi^{-1}(p_B) \approx \sqrt{2\pi}(p_A - p_B). \quad (36)$$

Finally we have

$$E_\delta[r] = \frac{\sigma}{2}E[\Phi^{-1}(p_A) - \Phi^{-1}(p_B)] \approx \frac{\sqrt{2\pi}\sigma}{2}E[p_A - p_B] \leq \frac{\sqrt{2\pi}\sigma^2}{4}\sqrt{\|F(x)\|_2} \quad (37)$$

B.4 SUMMARY ON THE RELATIONSHIP BETWEEN THE THREE METRICS

All three metrics are directly related to the gradient norm, which is used to measure the local sensitivity and stability of the model. Specifically, we list the differences between our method and the norm constraint-based method and the random smoothing method as follows

Table 8: The differences between our metric and other types of metrics

| Method | Random Smoothing | Norm Constraints | Our $\|F(x)\|_2$ |
|---|---|---|---|
| Starting Point | Centrality of Probability | Worst-case analysis | Information Geometry |
| Theoretical guarantee | Probabilistic Guarantee | Deterministic Guarantee | Expectation Sensitivity |
| Assumptions | Gaussian distribution | Maximizing the loss | Any distribution |

At the same time, the relationship between them is as follows:

- **Spectral norm $\|F(x)\|_2$ of FIM and the Lipschitz constant $L(x)$ of the model** :

$$\|F(x)\|_2 \leq B(x)\|L(x)\|^2. \quad (38)$$

- **Spectral norm $\|F(x)\|_2$ of FIM and the CLEVER score $\max_{z \in \mathcal{N}_p(x)} \|\nabla g(z)\|_2$ ($p(c|x) \approx 1$):**

$$\|F(x)\|_2 \approx e^{-g(x)}\|\nabla g(x)\|_2^2. \quad (39)$$

- **The Lipschitz constant $L(x)$ of the model and the CLEVER score** The former is the Lipschitz constant of the model $f(x)$, while CLEVER is the Lipschitz constant of the cross-entropy loss function.

- **Certification radius $r$ of the random smoothing and the spectral norm $\|F(x)\|_2$ of the FIM** : $\|F(x)\|_2$ limits the upper bound of the expectation of $r$

$$E_\delta[r] \leq \frac{\sqrt{2\pi}\sigma^2}{4}\sqrt{\|F(x)\|_2}. \quad (40)$$

## C   PROOF ON KL DIVERGENCE UNDER GENERAL DISCRETE DISTRIBUTION

**Theorem 6** *For any two class confidence distributions $p(y|x)$ and $p(y|x')$, where $x$ and $x'$ are the inputs of the model and $y$ is the class label of the model output, we have*

$$KL(p(y|x), p(y|x')) \approx \frac{1}{2}(x' - x)^T F(x)(x' - x) = \frac{1}{2}\delta^T F(x)\delta, \tag{41}$$

*where $F(x)$ is the Fisher information matrix defined as follows*

$$F(x) = \mathbb{E}_{p(y|x)}[\nabla_x \log p(y|x)\nabla_x \log p(y|x)^T]. \tag{42}$$

For any two discrete probability distributions $p(y|x)$ and $p(y|x')$, we have

$$\mathrm{KL}(p(y|x), p(y|x')) \approx \frac{1}{2}(x' - x)^T F(x)(x' - x), \tag{43}$$

where

$$F(x) = \mathbb{E}_{p(y|x)}[\nabla_x \log p(y|x)\nabla_x \log p(y|x)^T]. \tag{44}$$

First, we perform Taylor's second-order expansion of the function $\log p(y|x')$ at point $x$

$$
\begin{aligned}
\log p(y|x') &\approx \log p(y|x) + (\nabla_x \log p(y|x))^T (x' - x) \\
&\quad + \frac{1}{2}(x' - x)^T \nabla_x^2 \log p(y|x)(x' - x) + o(\|x' - x\|^3),
\end{aligned} \tag{45}
$$

then substitute it into the KL divergence

$$KL(p(y|x)||p(y|x')) = \sum_{i=1}^{K} p(y_i|x)[\log p(y_i|x) - \log p(y_i|x')]$$

to get

$$
\begin{aligned}
KL(p(y|x)||p(y|x')) &= -(x' - x)^T \sum_{i=1}^{K} p(y_i|x)\nabla \log p(y_i|x) \\
&\quad - \frac{1}{2}(x' - x)^T \left( \sum_{i=1}^{K} p(y_i|x)\nabla^2 \log p(y_i|x) \right) (x' - x) \\
&\quad - o(\|x' - x\|^3),
\end{aligned} \tag{46}
$$

where $o(\|x' - x\|^3)$ is the approximate error term.

For the first term above, we have

$$\sum_{i=1}^{K} p(y_i|x)\nabla \log p(y_i|x) = \nabla \sum_{i=1}^{K} p(y_i|x) = 0. \tag{47}$$

For the second term above, we have

$$
\begin{aligned}
\nabla_x \log p(y_i|x) &= \nabla_x p(y_i|x)/p(y_i|x), \\
\nabla_x^2 \log p(y_i|x) &= \frac{p(y_i|x)\nabla_x^2 p(y_i|x) - \nabla_x p(y_i|x)\nabla_x p(y_i|x)^T}{p(y_i|x)^2} \\
&= \frac{\nabla_x^2 p(y_i|x)}{p(y_i|x)} - \nabla_x \log p(y_i|x)\nabla_x \log p(y_i|x)^T.
\end{aligned} \tag{48}
$$

Further we get

$$\sum_{i=1}^{K} p(y_i|x)\nabla_x^2 \log p(y_i|x)$$

$$= \sum_{i=1}^{K}\{\nabla_x^2 p(y_i|x) - p(y_i|x)[\nabla_x \log p(y_i|x)\nabla_x \log p(y_i|x)^T]\},$$

$$= \nabla_x^2 \sum_{i=1}^{K} p(y_i|x) - \mathbb{E}_{p(y|x)}[\nabla_x \log p(y|x)\nabla_x \log p(y|x)^T],$$

$$= -\mathbb{E}_{p(y|x)}[\nabla_x \log p(y|x)\nabla_x \log p(y|x)^T] = -F. \tag{49}$$

Finally, we arrive at our conclusion.

## D  ANALYSIS OF KL DIVERGENCE APPROXIMATION ERROR

### D.1  THEORETICAL ANALYSIS

We express the approximation error as third-order remainder $R_3(\delta, x)$.

$$\text{KL}(p(y|x), p(y|x')) = \frac{1}{2}\delta^T F(x)\delta + R_3(\delta, x), \tag{50}$$

where $\delta = x' - x$. Assume there exists a constant $M > 0$ such that

$$\left|\frac{\partial^3 KL(p(y|x)\|p(y|x+\delta))}{\partial \delta_i \partial \delta_j \partial \delta_k}\right| \leq M, \quad \forall i, j, k, \tag{51}$$

Then the upper bound of the remainder is

$$|R_3(\delta, x)| \leq \frac{M}{6}\sum_{i,j,k}|\delta_i \delta_j \delta_k|. \tag{52}$$

Below we use the perturbation $l_\infty$ norm and $l_2$ norm to represent the upper bound of the approximation error respectively.

$l_\infty$ **upper bound** : We have

$$|R_3(\delta, x)| \leq \frac{M}{6}\sum_{i,j,k}|\delta_i \delta_j \delta_k| \leq \frac{Md^3}{6}\|\delta\|_\infty^3. \tag{53}$$

$l_2$ **upper bound** : We have

$$|R_3(\delta, x)| \leq \frac{M}{6}\sum_{i,j,k}|\delta_i \delta_j \delta_k| \leq \frac{M}{6}\left(\sum_i |\delta_i|\right)^3 = \frac{M}{6}\|\delta\|_1^3 \leq \frac{M}{6}\left(\sqrt{d}\|\delta\|_2\right)^3 = \frac{Md^{3/2}}{6}\|\delta\|_2^3. \tag{54}$$

Then we can conclude that

$$|R_3(\delta, x)| \leq \frac{Md^3}{6}\|\delta\|_\infty^3, \quad |R_3(\delta, x)| \leq \frac{Md^{3/2}}{6}\|\delta\|_2^3. \tag{55}$$

Since we usually consider robustness on the entire dataset, we can replace $\|\delta\|_2$ or $\|\delta\|_\infty$ in the upper bound with its upper bound $\theta$.

For the given dataset, we can replace $\|\delta\|_2$ or $\|\delta\|_\infty$ in the above formula with its upper bound $\theta$ .

### D.2  EXPERIMENTAL ESTIMATION

We randomly sample 500 samples on CIFAR10 using four classic models with CW adversarial training, where $\|\delta\|_\infty \leq \theta$, as shown in Table 1. Table 2 shows the results of ResNet18 on three datasets.

The results in both tables show that the approximation error and the proportionality coefficient of both are very small. Therefore, in practice, the approximation error can be ignored.

Table 9: Approximation error of multiple models on CIFAR10

| Model | $R_3(\delta, x)$ | $\|F(x)\|_2$ | $\frac{R_3(\delta,x)}{\|F(x)\|_2}$ | $\theta$ |
|---|---|---|---|---|
| ViT$_{B1}$6 | 2.7e-5 | 0.61 | 4.4e-5 | 8/255 |
| ResNet18 | 6.8e-4 | 0.77 | 8.8e-4 | 8/255 |
| VGG16 | 5.8e-5 | 0.09 | 6.4e-4 | 8/255 |
| DenseNet121 | 1.7e-3 | 2.18 | 7.7e-4 | 8/255 |

Table 10: Approximation error of multiple datasets on the ResNet18 model

| Dataset | $R_3(\delta, x)$ | $\|F(x)\|_2$ | $\frac{R_3(\delta,x)}{\|F(x)\|_2}$ | $\theta$ |
|---|---|---|---|---|
| Tiny-Imagenet | 1.7e-3 | 0.51 | 3.3e-3 | 4/255 |
| MNIST | 3.3e-5 | 0.01 | 3.3e-3 | 76/255 |
| CIFAR10 | 6.8e-4 | 0.77 | 8.8e-4 | 8/255 |

# E    STATISTICAL SIGNIFICANCE OF FISHER INFORMATION MATRIX

**Theorem 7** *For a deep learning model whose last layer uses a  **softmax**  function to implement classification tasks, where the input vector of softmax is $f(x)$, the Fisher information matrix is*

$$F(x) = var(J(x)), \tag{56}$$

*where $J(x)$ is the gradient matrix (Jacobian matrix) of the vector $f(x)$ with respect to the input $x$ and var represents the variance of the matrix random variable.*

According to Theorem 1, the Fisher information matrix $F$ measures the sensitivity of the model output distribution $p(y|x)$ to the input $x$. For classification tasks, $F$ is defined as

$$F(x) = \mathbb{E}_{p(y|x)}[\nabla_x \log p(y|x) \nabla_x \log p(y|x)^T]. \tag{57}$$

Next we need to estimate the maximum eigenvalue of the Fisher information matrix for some models.

For classification, we assume the model outputs $k$ probabilities $y_i$

$$y_i = p(y = i|x) = \frac{e^{f_i(x)}}{\sum_{k=1}^{K} e^{f_k(x)}}, \tag{58}$$

then

$$\log p(y = i|x) = f_i(x) - \log \sum_{k=1}^{K} e^{f_k(x)}. \tag{59}$$

Its gradient with respect to the input $x$ is (let $f_i = f_i(x)$)

$$
\begin{aligned}
\nabla_x \log p(y = i|x) &= \nabla_x f_i - \sum_{k=1}^{K} \left( \sum_{k=1}^{K} e^{f_k} \right)^{-1} e^{f_i} \nabla_x f_i \\
&= \nabla_x f_i - \sum_{k=1}^{K} p(y = k|x) \nabla_x f_k \\
&= \sum_{k=1}^{K} (1_{k=i} - p_k) \nabla_x f_k,
\end{aligned}
\tag{60}
$$

where $p_k = p(y = k|x)$ and $1_{i=k}$ is the indicator function.

We obtain the Fisher information matrix is

$$
\begin{aligned}
F(x) &= \mathbb{E}_{p(y|x)}[\nabla_x \log p(y|x) \nabla_x \log p(y|x)^T] \\
&= \sum_{k=1}^{K} p_k [\nabla_x \log p(y=k|x) \nabla_x \log p(y=k|x)^T] \\
&= \sum_{k=1}^{K} p_k \left[ \left( \sum_j (1_{k=j} - p_j) \nabla_x f_j \right) \left( \sum_i (1_{k=i} - p_i) \nabla_x f_i \right)^T \right] \\
&= \sum_{k=1}^{K} p_k \left[ \left( \sum_j 1_{k=j} \nabla_x f_j - \mathbb{E}[\nabla_x f] \right) \left( \sum_i 1_{k=i} \nabla_x f_i - \mathbb{E}[\nabla_x f] \right) \right]^T \\
&= \sum_{k=1}^{K} p_k \sum_{j,i} 1_{k=i} 1_{k=j} \nabla_x f_i \nabla_x f_j^T - \mathbb{E}[\nabla_x f] \sum_{k=1}^{K} p_k \sum_i 1_{k=i} \nabla_x f_k^T \\
&\quad - \sum_{k=1}^{K} p_k \sum_j 1_{k=j} \nabla_x f_j \mathbb{E}[\nabla_x f^T] + \sum_{k=1}^{K} p_k \mathbb{E}[\nabla_x f] \mathbb{E}[\nabla_x f^T] \\
&= \sum_{k=1}^{K} p_k \nabla_x f_k \nabla_x f_k^T - \mathbb{E}[\nabla_x f] \sum_{k=1}^{K} p_k \nabla_x f_k^T \\
&\quad - \sum_{k=1}^{K} p_k \nabla_x f_k \mathbb{E}[\nabla_x f]^T + \mathbb{E}[\nabla_x f] \mathbb{E}[\nabla_x f]^T \\
&= \mathbb{E}[\nabla_x f \nabla_x f^T] - \mathbb{E}[\nabla_x f] \mathbb{E}[\nabla_x f]^T \\
&= \mathrm{var}(J(x)). \tag{61}
\end{aligned}
$$

# F  GENERAL ANALYSIS OF MODEL ROBUSTNESS

**Theorem 8** *For any deep network-based classifier $h : x \to softmax(f(x))$, where softmax is the softmax function, the spectral norm $\|F(x)\|_2$ of its Fisher information matrix with respect with the input $x$ has the following upper bound*

$$
\|F(x)\|_2 = \lambda_{\max}(F(x)) = \max_{\|v\|_2=1} v^T F(x) v \le 2 \max_k p_k(1 - p_k) \|J(x)\|_2^2, \tag{62}
$$

*where $J_f(x)$ is the Jacobian matrix of the output $f(x) \in \mathcal{R}^K$ with respect to the input $x \in \mathcal{R}^d$. Let $B = diag(p) - pp^T$. When the principal eigenvector $w_1$ of $B$ is aligned with the principal left singular vector of $J(x)$, then there exists a principal right singular vector $v = J_f(x)^T w_1 / \|J_f(x)\|_2$ of $J(x)$ such that $\|F(x)\|_2 = 2 \max_k p_k(1 - p_k) \|J_f(x)\|_2^2$.*

To facilitate our estimation of the maximum eigenvalue of the Fisher information matrix, we rewrite it as

$$
\begin{aligned}
F(x) &= \sum_{k=1}^{K} p_k [\nabla_x \log p(y=k|x) \nabla_x \log p(y=k|x)^T] \\
&= \sum_{k=1}^{K} p_k \left[ \left( \sum_j (1_{k=j} - p_j) \nabla_x f_j \right) \left( \sum_i (1_{k=i} - p_i) \nabla_x f_i \right)^T \right] \\
&= \sum_{k=1}^{K} p_k \sum_{j,i} (1_{k=j} - p_j)(1_{k=i} - p_i) \nabla_x f_j \nabla_x f_i^T \\
&= \sum_{j,i=1}^{K} \left( \sum_{k=1}^{K} p_k (1_{k=j} - p_j)(1_{k=i} - p_i) \right) \nabla_x f_j \nabla_x f_i^T \\
&= J_f(x)^T B J_f(x) \tag{63}
\end{aligned}
$$

where $B_{ij} = \sum_{k=1}^{K} p_k(1_{k=j} - p_j)(1_{k=i} - p_i)$ and $J_f(x)$ is the Jacobian matrix of the $K$ outputs with respect to the $d$ inputs.

Now let's discuss $B_{ij} = \sum_{k=1}^{K} p_k(1_{k=j} - p_j)(1_{k=i} - p_i)$:

1) When $i = j$, we have

$$
\begin{aligned}
\sum_{k=1}^{K} p_k(1_{k=j} - p_j)(1_{k=i} - p_i) &= \sum_{k=1}^{K} p_k(1_{k=i} - p_i)^2 \\
&= \sum_{k=1}^{K} p_k(1_{k=i}^2 - 21_{k=i}p_i + p_i^2) \\
&= p_i - 2p_i^2 + p_i^2 \\
&= p_i(1 - p_i)
\end{aligned}
\tag{64}
$$

2) When $i \neq j$, then

$$
\begin{aligned}
\sum_{k=1}^{K} p_k(1_{k=j} - p_j)(1_{k=i} - p_i) &= \sum_{k=1}^{K}(p_k 1_{k=j} 1_{k=i} - p_k 1_{k=j} p_i \\
&\quad - p_k 1_{k=i} p_j + p_k p_j p_i) \\
&= 0 - p_j p_i - p_i p_j + p_j p_i \\
&= -p_i p_j
\end{aligned}
\tag{65}
$$

Finally, we get a matrix B with dimension $K \times K$

$$
B = \mathrm{diag}(p) - pp^T.
\tag{66}
$$

We use the Gershgorin disk theorem Golub & Loan (2013) to estimate the range of eigenvalues. For $B$, the center of the $i$-th Gershgorin disk is $B_{ii} = p_i(1 - p_i)$, and the radius is $\sum_{j \neq i} |B_{ij}| = \sum_{j \neq i} p_i p_j = p_i(1 - p_i)$. Therefore, each eigenvalue satisfies

$$
|\lambda - p_i(1 - p_i)| \leq p_i(1 - p_i),
\tag{67}
$$

which means $\lambda \in [0, 2p_i(1 - p_i)]$. Then we have

$$
\|B\|_2 \leq 2p_i(1 - p_i).
\tag{68}
$$

Finally, we estimate the largest eigenvalue of the matrix $F(x) = J_f(x)^T B J_f(x)$, which is equal to the Rayleigh quotient

$$
\begin{aligned}
\lambda_{\max}(F(x)) &= \max_{\|v\|=1} (J_f(x)v)^T B(J_f(x)v) \\
&\leq \|J_f(x)\|_2 \|v\|_2 \|B\|_2 \|J_f(x)\|_2 \|v\|_2 \\
&= \|B\|_2 \|J_f(x)\|_2^2.
\end{aligned}
\tag{69}
$$

Assume that the model output is a classification probability vector $p = [p_1, p_2, \cdots, p_K]^T$, and let $Y$ be a random class label (one-hot vector), then we have

$$
\mathbb{E}[Y] = p, \quad \mathbb{E}[YY^T] = \mathrm{diag}(p).
\tag{70}
$$

So we have

$$
B = \mathrm{cov}(Y) = \mathbb{E}[YY^T] - E[Y]E[Y]^T.
\tag{71}
$$

Next we discuss the condition that there exists $v(\|v\|_2 = 1)$ such that $\lambda_{\max}(F(x)) = \|B\|_2 \|J_f(x)\|_2^2$. Let $y = J_f(x)v$, where $\|v\|_2 = 1$, then we have

$$
\lambda_{\max}(F(x)) = \max_{\|v\|=1} y^T B y.
\tag{72}
$$

When $y^* = cw_1(c > 0) = J_f(x)v$, where $w_1(\|w_1\|_2 = 1)$ is the main eigenvector of $B$, then we have

$$\lambda_{\max}(F(x)) = \|y^*\|_2^2 \|B\|_2 = c^2 \|B\|_2. \tag{73}$$

We look for the maximum value of $c$ and get $\lambda_{\max}(F(x))$. Furthermore, we let $w_1 = \frac{1}{c} J_f(x)v$, then

$$w_1^T w_1 = \frac{1}{c^2} v^T J_f(x)^T J_f(x)v = 1. \tag{74}$$

So we have

$$c = \sqrt{v^T J_f(x)^T J_f(x)v} \tag{75}$$

We immediately get the optimal value $c^*$ of $c$ to be

$$c^* = \max_{\|v\|_2=1} \sqrt{v^T J_f(x)^T J_f(x)v} = \|J_f(x)\|_2, \tag{76}$$

where $v$ is the right singular vector corresponding to the largest singular value of $J_f(x)$. Since $\|J_f(x)\|_2 w_1^T w_1 = w_1^T J_f(x)v = \|J_f(x)\|_2$, so when $w_1$ and $v$ are the left and right singular vectors corresponding to the maximum singular value of $J_f(x)$, we have

$$\lambda_{\max}(F(x)) = \|B\|_2 \|J_f(x)\|_2^2. \tag{77}$$

When the principal eigenvector $w_1$ of $B$ is the principal left singular vector of $J_f(x)$, then

$$\|J_f(x)\|_2 w_1 = J_f(x)v \quad \rightarrow \quad \|J_f(x)\|_2 J_f(x)^T w_1 = J_f(x)^T J_f(x)v = \|J_f(x)\|_2^2 v$$
$$\rightarrow \quad v = J_f(x)^T w_1 / \|J_f(x)\|_2. \tag{78}$$

So there exists $J_f(x)^T w_1 / \|J_f(x)\|_2$ such that the equation holds. However, $w_1$ is the principal eigenvector of $\|B\|_2$, and is usually unlikely to be the principal left singular vector of $J_f(x)$.

# G $\|J\|_2$ ESTIMATION OF BASIC MODULES

## G.1 CONVOLUTION LAYER

*Theorem: For convolution operations on multi-channel images, the spectral norm $\|J_\Psi\|_2$ of the Jacobian matrix of the convolution operator $\Psi$ is approximately the spectral norm $\|W\|_2$ of the convolution kernel $W$, i.e. $\|J_\Psi\|_2 \approx \|W\|_2$.*

1) When the convolution operator's padding is 'SAME' and circular padding is used, where the stride $s$ is 1, so the input and output of the convolution operator have the same size. For the convolutional mapping $\Psi : \mathbb{R}^{H \times W \times C_{\text{in}}} \rightarrow \mathbb{R}^{H \times W \times C_{\text{out}}}$:

$$\Psi_{h',w',c'} = \sum_{i=0}^{k-1} \sum_{j=0}^{k-1} \sum_{c=1}^{C_{\text{in}}} W_{i,j,c,c'} X_{h'+i,w'+j,c}. \tag{79}$$

We divide $J_\Psi$ into blocks according to the output channel $c'$ and the input channel $c$, then each block $[J_\Psi]_{c',c} \in \mathbb{R}^{HW \times HW}$ can be a circulant matrix with circulant filled.

Under the loop filling condition, the Jacobian matrix can be expressed as a double loop structure

$$J_\Psi^{\text{circ}} = \sum_{i=0}^{k-1} \sum_{j=0}^{k-1} \Pi_H^i \otimes \Pi_W^j \otimes W_{i,j}, \tag{80}$$

where $\otimes$ denotes the Kronecker product, $W_{i,j} \in C_{\text{out}} \times C_{\text{in}}$ is a tensor slice of the matrix $W$, $\Pi_H$ denotes the circulant shift matrix of $H \times H$

$$\begin{bmatrix} 0 & 0 & \cdots & 0 & 1 \\ 1 & 0 & \cdots & 0 & 0 \\ 0 & 1 & \cdots & 0 & 0 \\ \vdots & \vdots & \ddots & \vdots & \vdots \\ 0 & 0 & \cdots & 1 & 0 \end{bmatrix}. \tag{81}$$

Let $\omega_1 = e^{-2\pi i/H}$ and $\omega_2 = e^{-2\pi i/W}$, where $i$ is the imaginary unit, we diagonalize the cyclic shift matrices separately

$$\Pi_H = F_H \Lambda_H F_H^*, \quad \Pi_W = F_W \Lambda_W F_W^*, \tag{82}$$

where $\Lambda_H = \mathrm{diag}(1, \omega_1, \cdots, \omega_1^{H-1})$ and $\Lambda_W = \mathrm{diag}(1, \omega_2, \cdots, \omega_2^{W-1})$. We substitute them into Eqn. (80) and obtain

$$
\begin{aligned}
J_\Psi^{\mathrm{circ}} &= \sum_{i=0}^{k-1}\sum_{j=0}^{k-1} (F_H \Lambda_H F_H^*) \otimes (F_W \Lambda_W F_W^*) \otimes W_{i,j}, \\
&= \sum_{i=0}^{k-1}\sum_{j=0}^{k-1} (F_H \Lambda_H F_H^*)^i \otimes (F_W \Lambda_W F_W^*)^j \otimes (I_{\mathrm{out}} W_{i,j} I_{\mathrm{in}}) \\
&= \sum_{i=0}^{k-1}\sum_{j=0}^{k-1} (F_H \Lambda_H^i F_H^*) \otimes (F_W \Lambda_W^j F_W^*) \otimes (I_{\mathrm{out}} W_{i,j} I_{\mathrm{in}}) \\
&= \sum_{i=0}^{k-1}\sum_{j=0}^{k-1} (F_H \otimes F_W \otimes I_{\mathrm{out}})(\Lambda_H^i \otimes \Lambda_W^j \otimes W_{i,j})(F_H \otimes F_W \otimes I_{\mathrm{in}}) \\
&= (F_H \otimes F_W \otimes I_{\mathrm{out}}) \left( \sum_{i=0}^{k-1}\sum_{j=0}^{k-1} \Lambda_H^i \otimes \Lambda_W^j \otimes W_{i,j} \right) (F_H \otimes F_W \otimes I_{\mathrm{in}}), \\
&= (F_H \otimes F_W \otimes I_{\mathrm{out}}) \hat{W} (F_H \otimes F_W \otimes I_{\mathrm{in}}), \tag{83}
\end{aligned}
$$

where $\hat{W} = \sum_{i=0}^{k-1}\sum_{j=0}^{k-1} \Lambda_H^i \otimes \Lambda_W^j \otimes W_{i,j}$.

Notice that $\Lambda_H^i \otimes \Lambda_W^j = \mathrm{diag}(\mu_{0,0}^{i,j}, \mu_{0,1}^{i,j}, \cdots, \mu_{H-1,M-1}^{i,j})$, where $\mu_{u,v}^{i,j} = \omega_1^{ui}\omega_2^{vj}$. We simplify $\Lambda_H^i \otimes \Lambda_W^j \otimes W_{i,j}$ into diagonal blocks to obtain

$$
\begin{aligned}
\hat{W} &= \sum_{i=0}^{k-1}\sum_{j=0}^{k-1} \mathrm{blkdiag}(\mu_{0,0}^{i,j} W_{i,j}, \mu_{0,1}^{i,j} W_{i,j}, \cdots, \mu_{H-1,W-1}^{i,j} W_{i,j}) \\
&= \mathrm{blkdiag}(\hat{W}_{0,0}, \hat{W}_{0,1}, \cdots, \hat{W}_{H-1,W-1}), \tag{84}
\end{aligned}
$$

where $\hat{W}_{p,q}$ is the two-dimensional Discrete Fourier Transform (DFT) of the convolution kernel $W$ at frequency $(p, q)$

$$\hat{W}_{p,q} = \sum_{i=0}^{k-1}\sum_{j=0}^{k-1} \mu_{p,q}^{i,j} W_{i,j}. \tag{85}$$

Therefore we have

$$\|J_\Psi\|_2 = \|J_\Psi^{\mathrm{circ}}\|_2 = \max_{p,q} \sigma_{\max}(\hat{W}_{p,q}) = \max_{p,q} \|\hat{W}_{p,q}\|_2 = \|W\|_2. \tag{86}$$

2) When the convolution operator uses zero padding, $W$ is a Toeplitz matrix (corresponding to non-circular convolution). According to the asymptotic spectral theory of the Toeplitz matrix (Grenander-Szegő theorem) Grenander & Szego (1958), when $H, W \gg k$, the spectral norm of the Toeplitz matrix $W$ converges to the $l_\infty$ norm of its sign function (i.e., the Fourier transform of the convolution kernel $W$)

$$\lim_{n\to\infty} \|W\|_2 = \|\hat{W}\|_\infty = \max_{u,v} \|\hat{W}_{u,v}\|_2 = \|J_\Psi\|_2. \tag{87}$$

3) Assuming the stride $s$ in the convolution operator is $s \geq 1$ and the padding method is **VALID** (i.e. no padding), the output size of the convolution operator is

$$H' = \left\lfloor \frac{H-k}{s} \right\rfloor + 1 \leq H, \quad W' = \left\lfloor \frac{W-k}{s} \right\rfloor + 1 \leq W. \tag{88}$$

For any matrix $A$, without loss of generality, we delete the last $k$ rows of $A$ to obtain $B$

$$A = \begin{bmatrix} a_1^T \\ a_2^T \\ \vdots \\ a_n^T \end{bmatrix}, \quad B = \begin{bmatrix} a_1^T \\ a_2^T \\ \vdots \\ a_{n-k}^T \end{bmatrix}, \tag{89}$$

so we have

$$\|A\|_2^2 = \max_{\|v\|_2=1} v^T A^T Ax = \sum_{i=1}^{n} (a_i^T v)^2 \geq \max_{\|v\|_2=1} v^T B^T Bx = \sum_{i=1}^{n-k} (a_i^T v)^2 = \|B\|_2^2. \tag{90}$$

That is, the spectral norm of the submatrix is less than or equal to the spectral norm of the original matrix, e.g. $\|B\|_2 \geq \|A\|_2$.

Note that $J$ can be regarded as a submatrix obtained by deleting some rows and columns from the Jacobian matrix of the complete convolution ($s = 1$, padding= **SAME** ), and the spectral norm of the submatrix is smaller than the spectral norm of the original matrix, so there is

$$\|J_\Psi\|_2 \leq \max_{p,q} \|\hat{W}_{p,q}\|_2 = \|W\|_2. \tag{91}$$

In summary, we can use $\|W\|_2$ to approximate the spectral norm of the Jacobian matrix of the convolution operator,i.e. $\|J_\Psi\|_2 \approx \|W\|_2$.

## G.2 ReLU Layer

The ReLU function is defined as $\text{ReLU}(x) = \max(0, x)$, so its derivative is

$$\frac{d}{dx}\text{ReLU}(x) = \begin{cases} 1, & x > 0 \\ 0, & x \leq 0. \end{cases} \tag{92}$$

For an input vector $x \in \mathbb{R}^n$, the ReLU Jacobian matrix $J_{\text{ReLU}} \in \mathbb{R}^{n \times n}$ is a diagonal matrix

$$J_{\text{ReLU}} = \text{diag}(1_{x>0}). \tag{93}$$

We immediately get

$$\|J_{\text{ReLU}}\|_2 = 1. \tag{94}$$

## G.3 Max Pooling Layer

Considering the input tensor $X \in \mathbb{R}^{H \times W \times C}$ and the stride of the max pooling layer is 2 and the pooling layer size is $2 \times 2$, the output $Y \in \mathbb{R}^{(H/2) \times (W/2) \times C}$ is

$$Y_{i,j,c} = \max(X_{2i-1,2j-1,c}, X_{2i-1,2j,c}, X_{2i,2j-1,c}, X_{2i,2j,c}) \tag{95}$$

Furthermore, the Jacobian matrix $J \in \mathbb{R}^{((H/2)(W/2)C) \times (HWC)}$ describes the gradient relationship of the output $Y$ to the input $X$

$$J_{(i,j,c),(k,l,m)} = \frac{\partial Y_{i,j,c}}{\partial X_{k,l,m}} \tag{96}$$

Given $Y_{i,j,c}$, if $k, l, c = \arg\max(X_{2i-1,2j-1,c}, X_{2i-1,2j,c}, X_{2i,2j-1,c}, X_{2i,2j,c})$, then $Y_{i,j,c} = X_{k,l,c}$, i.e. $\frac{\partial Y_{i,j,c}}{\partial X_{k,l,c}} = 1$; otherwise, $\frac{\partial Y_{i,j,c}}{\partial X_{k,l,c}} = 0$.

So each row has exactly one 1 (corresponding to the maximum value), and all the others are 0, so the vectors in each row are orthogonal to each other, and we immediately get

$$JJ^T = I_{(H/2)(W/2)C} \tag{97}$$

and

$$\|J\|_2 = \sqrt{\lambda_{\max}(J^T J)} = \sqrt{\lambda_{\max}(JJ^T)} = 1. \tag{98}$$

### G.4 AVERAGE POOLING

Suppose we input a tensor $X \in \mathbb{R}^{H \times W \times C}$, where $H$ is the height, $W$ is the width, and $C$ is the number of channels. For two-dimensional average pooling, we use a pooling window such as $k \times k$ to slide on the input with a certain step size, and calculate the average of all elements in each window as the output.

To simplify the analysis, we assume that the input is a vector $x \in \mathbb{R}^n$, and average pooling divides $x$ into $m = n/k$ ($k$ can divide $n$) windows of size $k$, then we have

$$y_i = \frac{1}{k} \sum_{j=(i-1)k+1}^{ik} x_j, \quad , i = 1, 2, \cdots, m. \tag{99}$$

We calculate the partial derivative of $y_i$ with respect to $x_j$ and obtain

$$\frac{\partial y_i}{\partial x_j} = \begin{cases} 1/k, & (i-1)k + 1 \le j \le ik, \\ 0, & \text{otherwise.} \end{cases} \tag{100}$$

Further, we will write the Jacobian matrix of the average pooling into a block matrix form based on the above results

$$J = \text{blkdiag}\left(\frac{1}{k}1_k^T, \frac{1}{k}1_k^T, \cdots, \frac{1}{k}1_k^T\right). \tag{101}$$

We know that $\|J\|_2$ is the square root of the eigenvalue of $J^T J$, so we calculate $J^T J$

$$J^T J = \text{blkdiag}\left(\frac{1}{k^2}1_k 1_k^T, \frac{1}{k^2}1_k 1_k^T, \cdots, \frac{1}{k^2}1_k 1_k^T\right), \tag{102}$$

where each diagonal block is a $k \times k$ matrix with all elements $\frac{1}{k^2}$. The rank of the matrix $\frac{1}{k^2}$ is 1, and its non-zero eigenvalues are

$$\lambda_{max}\left(\frac{1}{k^2}1_k 1_k^T\right) = \frac{1_k^T \left(\frac{1}{k^2}1_k 1_k^T\right) 1_k}{1_k^T 1_k} = \frac{1}{k}. \tag{103}$$

That is, $J^T J$ has an $m$-th eigenvalue $\frac{1}{k}$ and an $n - m$ eigenvalue $0$. Therefore, we have

$$\|J\|_2 = \frac{1}{\sqrt{k}}. \tag{104}$$

We generalize it to two-dimensional pooling, then the pooling window is $k \times k$, so each element corresponds to the average of $k^2$ inputs, and we have similar conclusions

$$\|J\|_2 = \frac{1}{k}, \tag{105}$$

where the window size $k$ is usually set to 2 in the construction of deep learning models.

### G.5 BATCH NORMALIZATION (BN)

Given an input $x \in \mathbb{R}^C$ (assuming each channel $c$ is processed independently), the output $y^{(c)}$ of the BN layer is

$$y^{(c)} = \gamma^{(c)} \frac{x^{(c)} - \mu^{(c)}}{\sqrt{(\sigma^{(c)})^2 + \epsilon}} + \beta^{(c)}, \tag{106}$$

where the mean parameter $\mu^{(c)}$, the offset parameter $\beta^{(c)}$, and the variance parameter $\sigma^{(c)}$ are all constants during the inference stage.

For the convenience of analysis, we write the BN transformation in matrix form

$$y = D(x - u) + \beta, \tag{107}$$

where

$$D = \text{diag}\left(\frac{\gamma^{(1)}}{\sqrt{(\sigma^{(1)})^2 + \epsilon}}, \cdots, \frac{\gamma^{(C)}}{\sqrt{(\sigma^{(C)})^2 + \epsilon}}\right). \tag{108}$$

We immediately get

$$\|J_{\text{BN}}\|_2 = \left\|\frac{\partial y}{\partial x}\right\|_2 = \|D\|_2 = \max_c \frac{|\gamma^{(c)}|}{\sqrt{(\sigma^{(c)})^2 + \epsilon}} \tag{109}$$

Usually if $|\gamma| \gg \sigma$, there is a risk of gradient explosion, while $|\gamma| \ll \sigma$ has a risk of gradient vanishing. Therefore, in practice, we usually approximately select $\gamma \approx \sigma$ or $\gamma \approx \sqrt{\sigma^2 + \epsilon}$, where $\epsilon$ is a very small positive constant. In general, we have approximately

$$\|J_{\text{BN}}\|_2 = \max_c \frac{|\gamma^{(c)}|}{\sqrt{(\sigma^{(c)})^2 + \epsilon}} \approx O(1). \tag{110}$$

### G.6 LAYER NORMALIZATION (LN)

The Layer normalization (LN) operation on the input $x \in R^d$ is defined as ($\odot$ is the element-wise multiplication)

$$y = \gamma \odot \frac{x - \mu}{\sqrt{\sigma^2 + \epsilon}} + \beta, \quad \mu = \frac{1}{d}\sum_{i=1}^{d} x_i, \quad \sigma^2 = \frac{1}{d}\sum_{i=1}^{d}(x_i - \mu)^2, \tag{111}$$

where $\gamma, \beta \in \mathbb{R}^d$ are learnable scale and offset parameters ($\epsilon$ is a small constant).

According to the chain rule, $J_{\text{LN}} = \frac{\partial y}{\partial x}$ can be expressed as

$$J_{\text{LN}} = \text{diag}(\gamma)\frac{\partial z}{\partial x}, \quad z = \frac{x - \mu}{\sqrt{\sigma^2 + \epsilon}}. \tag{112}$$

Furthermore, we have

$$\frac{\partial z_i}{\partial x_j} = \frac{\delta_{ij} - 1/d}{\sqrt{\sigma^2 + \epsilon}} - \frac{(x_i - \mu)(x_j - \mu)}{d(\sigma^2 + \epsilon)^{3/2}}, \tag{113}$$

where $\delta_{ij}$ is defined as

$$\delta_{ij} = \begin{cases} 1, & i = j, \\ 0, & i \neq j. \end{cases} \tag{114}$$

That is

$$
\begin{aligned}
\frac{\partial z}{\partial x} &= \frac{1}{\sqrt{\sigma^2 + \epsilon}}\left(I - \frac{1}{d}11^T - \frac{(x - \mu)(x - \mu)^T}{d(\sigma^2 + \epsilon)}\right) \\
&= \frac{1}{\sqrt{\sigma^2 + \epsilon}}\left(I - \frac{1}{d}11^T - \frac{(x - \mu)(x - \mu)^T}{d\sigma^2} + \frac{(x - \mu)(x - \mu)^T}{d\sigma^2} - \frac{(x - \mu)(x - \mu)^T}{d(\sigma^2 + \epsilon)}\right) \\
&= \frac{1}{\sqrt{\sigma^2 + \epsilon}}\left(I - P + \frac{(x - \mu)(x - \mu)^T}{d\sigma^2} - \frac{(x - \mu)(x - \mu)^T}{d(\sigma^2 + \epsilon)}\right),
\end{aligned} \tag{115}
$$

where $P = I - \frac{1}{d}11^T - \frac{(x-\mu)(x-\mu)^T}{d\sigma^2}$.

Next we prove that the matrix $P$ is a projection matrix. We can observe that

$$P = aa^T + bb^T, \tag{116}$$

where $a = \frac{1}{\sqrt{d}}1$ and $b = \frac{1}{\sqrt{d}}\frac{x-u}{\sigma}$. We have

$$a^T a = 1, \quad a^T b = \frac{1}{d}\frac{\sum_{i=1}^{d} x_i - d\mu}{\sqrt{\sigma^2}} = 0, \quad b^T b = \frac{1}{d}\frac{\sum_{i=1}^{d}(x_i - u)^2}{\sigma^2} = \frac{\sigma^2}{\sigma^2} = 1. \tag{117}$$

Then
$$P^2 = (aa^T + bb^T)(aa^T + bb^T) = aa^T + bb^T = P. \tag{118}$$

According to the properties of the projection matrix $P$, we have $I - P$ is also a projection matrix, so the eigenvalue of $I - P$ is either 1 or 0. That is

$$\|I - P\|_2 = 1. \tag{119}$$

According to the properties of the spectral norm, we have

$$
\begin{aligned}
\|J_{\mathrm{LN}}\|_2 &\leq \|\mathrm{diag}(\gamma)\|_2 \left\| \frac{\partial z}{\partial x} \right\|_2 \\
&= \frac{1}{\sqrt{\sigma^2 + \epsilon}} \|\mathrm{diag}(\gamma)\|_2 \left\| I - P + \frac{(x-\mu)(x-\mu)^T}{d\sigma^2} - \frac{(x-\mu)(x-\mu)^T}{d(\sigma^2 + \epsilon)} \right\|_2 \\
&\leq \frac{1}{\sqrt{\sigma^2 + \epsilon}} \|\mathrm{diag}(\gamma)\|_2 \left( \|I - P\|_2 + \left\| \frac{(x-\mu)(x-\mu)^T}{d\sigma^2} - \frac{(x-\mu)(x-\mu)^T}{d(\sigma^2 + \epsilon)} \right\|_2 \right) \\
&\leq \frac{1}{\sqrt{\sigma^2 + \epsilon}} \|\mathrm{diag}(\gamma)\|_2 \left( 1 + \frac{\epsilon d\sigma^2}{d\sigma^2(\sigma^2 + \epsilon)} \right) \\
&= \frac{1}{\sqrt{\sigma^2 + \epsilon}} \max_i \gamma^{(i)} \left( 1 + \frac{\epsilon}{\sigma^2 + \epsilon} \right). 
\end{aligned}
\tag{120}
$$

Usually we have $\epsilon \ll \sigma^2$, and thus $\frac{\epsilon}{\epsilon + \sigma^2} \to 0$. Finally we have

$$\|J_{\mathrm{LN}}\|_2 \leq \max_i \frac{|\gamma^{(i)}|}{\sqrt{\sigma^2 + \epsilon}} \approx O(1). \tag{121}$$

### G.7 SOFTMAX FUNCTION

The softmax function $\sigma$ is defined as

$$\sigma(z)_i = \frac{e^{z_i}}{\sum_{j=1}^n e^{z_j}}, \quad i = 1, 2, \cdots, n. \tag{122}$$

Its Jacobian matrix $J_\sigma(z)$ is a $n \times n$ matrix, where

$$\sigma(z)_{ij} = \frac{\partial \sigma_i}{\partial z_j} = \sigma_i(\delta_{ij} - \sigma_j). \tag{123}$$

Therefore, we can represent it in matrix form

$$J_\sigma = \mathrm{diag}(\sigma(z)) - \sigma(z)\sigma(z)^T, \tag{124}$$

which is a symmetric matrix.

We use the Gershgorin disk theorem Golub & Loan (2013) to estimate the range of eigenvalues. For $J$, the center of the $i$-th Gershgorin disk is $J_{ii} = \sigma_i(1 - \sigma_i)$, and the radius is $\sum_{j \neq i} |J_{ij}| = \sum_{j \neq i} \sigma_i \sigma_j = \sigma_i(1 - \sigma_i)$. Therefore, each eigenvalue satisfies

$$|\lambda - \sigma_i(1 - \sigma_i)| \leq \sigma_i(1 - \sigma_i), \tag{125}$$

which means $\lambda \in [0, 2\sigma_i(1 - \sigma_i)]$. Then we have

$$\|J_{\mathrm{softmax}}\|_2 \leq 2\sigma_i(1 - \sigma_i). \tag{126}$$

When $\sigma_i = 1/2$, the upper bound $2\sigma_i(1 - \sigma_i)$ takes the maximum value $1/2$. That is, $\|J\|_{\mathrm{softmax}} \leq \frac{1}{2}$.

Note that when the dimension $d$ of the vector is very high, then $\sigma_k$ are approximately equal, and we have

$$2\sigma_k(1 - \sigma_k) \approx \frac{2}{d}\left(1 - \frac{1}{d}\right) \approx \frac{2}{d}, \tag{127}$$

which leads $\|J_\sigma\|_2 \leq \frac{2}{d}$.

### G.8 Block Matrix

Assume that the block matrix $M \in R^{m \times (n_1 + n_2)}$ is composed of two sub-matrices $A \in R^{m \times n_1}$ and $B \in R^{m \times n_2}$ horizontally concatenated

$$M = [A \quad B], \tag{128}$$

the spectral norm of a matrix $M$ is its maximum singular value, defined as

$$\|M\|_2 = \max_{\|x\|_2 = 1} \|Mx\|_2, \tag{129}$$

where $x \in R^{n_1 + n_2}$, and defined as

$$x = \begin{bmatrix} x_1 \\ x_2 \end{bmatrix}. \tag{130}$$

So we have

$$\begin{aligned} \|Mx\|_2 &= \|Ax_1 + Bx_2\| \\ &\leq \|A\|_2 \|x_1\|_2 + \|B\|_2 \|x_2\|_2 \\ &\leq \sqrt{\|A\|_2^2 + \|B\|_2^2} \sqrt{\|x_1\|_2^2 + \|x_2\|_2^2} \\ &= \sqrt{\|A\|_2^2 + \|B\|_2^2}. \end{aligned} \tag{131}$$

On the other hand, if we let $x_1$ be the main right singular vector of A, we have

$$\|M\|_2 = \max_{\|x\|_2 = 1} \|Mx\|_2 \geq \left\| M \begin{bmatrix} x_1 \\ 0 \end{bmatrix} \right\|_2 = \|Ax_1\|_2 = \|A\|_2. \tag{132}$$

Similarly, if we let $x_2$ be the main right singular vector of matrix $B$, we have $\|M\|_2 \geq \|B\|_2$. Therefore

$$\|M\|_2 \geq \max(\|A\|_2, \|B\|_2). \tag{133}$$

That is

$$\max(\|A\|_2, \|B\|_2) \leq \|M\|_2 \leq \sqrt{\|A\|_2^2 + \|B\|_2^2} \leq \sqrt{2} \max\{\|A\|_2, \|B\|_2\}. \tag{134}$$

Furthermore, we can generalize to a block matrix consisting of $n$ matrices

$$\max_i(\|A_i\|_2) \leq \|[A_1 \quad A_2 \quad \cdots \quad A_n]\|_2 \leq \sqrt{\sum_{i=1}^{n} \|A_i\|_2^2} \leq \sqrt{n} \max_i\{\|A_i\|_2\}. \tag{135}$$

## H Robustness Analysis of Classical Deep Learning Networks

Since the components ReLU, Max Pooling, Average Pooling, BN and LN usually have a constant spectral norm upper bound $O(1)$, for the convenience of discussion, we mainly focus on the spectral norm upper bounds of convolutional layers, fully connected layers and concatenation layers.

### H.1 VGGNet

VGGNet is mainly composed of consecutive convolutional layers and fully connected layers, each of which is followed by ReLU activation and maximum pooling. Assuming that VGGNet has L convolutional layers and M fully connected layers, we have

$$\|J_f\|_2 \leq \prod_{i=1}^{L} \|W_i\|_2 \cdot \prod_{j=1}^{M} \|U_j\|_2, \tag{136}$$

where $W_i$ is the convolution kernel and $U_j$ is the weight of the fully connected layer.

Since VGGNet is very deep and has no residual connections, the upper bound of $\|J_f\|_2$ will grow or decay exponentially with depth (depending on the size of $\|W_i\|_2$). It is worth noting that VGG16 and VGG19 contain 13 convolutional layers, 3 fully connected layers and 16 convolutional layers, 3 fully connected layers, respectively.

## H.2 RESNET

The core innovation of ResNet is the residual connection, which is used to solve the gradient vanishing problem of deep networks. It mainly includes the initial convolution layer, the maximum pooling layer, the residual block and the global average pooling + fully connected layer. ResNet18 and ResNet50 contain 8 residual blocks and 16 residual blocks, respectively.

Suppose a residual block is

$$f_i(x) = x + g_i(x), \tag{137}$$

where $g_i$ is the composite function of the convolutional layer. The Jacobian matrix of the function $f_i$ is

$$J_{f_i} = I + J_{g_i}. \tag{138}$$

Therefore, we have

$$\|J_{f_i}\|_2 \leq \|I\|_2 + \|J_{g_i}\|_2 = 1 + \|J_{g_i}\|_2. \tag{139}$$

Assuming that the function $g_i$ is a combination of convolution + ReLU + convolution [3], then

$$\|J_{g_i}\| \leq \|W_{i,1}\|_2 \cdot \|W_{i,2}\|_2. \tag{140}$$

ResNet usually consists of an initial convolutional layer followed by multiple residual blocks, a global average pooling layer and a fully connected layer. So we end up with (assuming $\|J_{BN}\|_2 \leq 1$)

$$\|J_{\text{resnet}}\|_2 \leq \frac{1}{2}\|W_{\text{cov}}\|_2 \prod_{l=1}^{L}(1 + \|W_{l,1}\|_2\|W_{l,2}\|_2)\|U\|_2, \tag{141}$$

where $W_{\text{cov}}$ and $U$ are the weights of the initial convolutional layer and the fully connected layer, respectively.

ResNet still grows with depth, but more modestly than VGGNet's exponential product.

## H.3 DENSENET

DenseNet121 contains 4 dense blocks, a total of 58 convolution layers, and DenseNet169 also contains 4 dense blocks, but is deeper than DenseNet121 and contains 82 convolution layers.

In dense blocks, each layer is the concatenation of the outputs of all previous layers. Suppose the output of the $l$-th layer is

$$X_l = H_l(\text{concat}(X_0, X_1, \cdots, X_{l-1})) \tag{142}$$

According to the properties of the Jacobian matrix, we have ($X_0$ is the input of the network)

$$
\begin{aligned}
\|J_L\|_2 &= \left\|\frac{\partial X_L}{\partial X_0}\right\|_2 \\
&= \left\|\frac{\partial H_L}{\partial \text{cat}(X_0, X_1, \cdots, X_{L-1})}\frac{\partial \text{cat}(X_0, X_1, \cdots, X_{L-1})}{\partial X_0}\right\|_2 \\
&\leq \left\|\frac{\partial(\text{cov} \cdot \text{ReLU} \cdot \text{BN})}{\partial \text{cat}}\right\|_2 \left\|\begin{bmatrix} I \\ J_1 \\ \cdots \\ J_{L-1} \end{bmatrix}\right\|_2, \quad \text{Eqn.}(135) \\
&\leq \|W_L\|_2\sqrt{1 + \sum_{l=1}^{L-1}\|J_l\|_2^2} \\
&\leq \|W_L\|_2(1 + \sum_{l=1}^{L-1}\|J_l\|_2) \\
&= \|W_L\|_2 S_{L-1}, \tag{143}
\end{aligned}
$$

---

[3] The residual blocks of ResNet are basic block and bottleneck block, where the former contains 2 convolutional layers, while the latter contains 3 convolutional layers.

where $S_{L-1} = \sum_{k=0}^{L-1} \|J_k\|_2$ and $J_0 = I$.

We now use mathematical induction to prove

$$S_L \leq \prod_{l=1}^{L}(1 + \|W_l\|_2). \tag{144}$$

When $L = 1$, we have

$$S_1 = 1 + \|J_1\|_2 \leq 1 + \|W_1\|_2. \tag{145}$$

Furthermore, we assume that the conclusion holds when $k = l$, that is $S_{l-1} \leq \prod_{k=1}^{l-1}(1 + \|W_k\|_2)$. Then using the conclusion $J_l \leq \|W_l\|_2 S_{l-1}$ from Eqn. (143), we immediately have

$$
\begin{aligned}
S_l &= S_{l-1} + \|J_l\|_2 \\
&\leq S_{l-1} + \|W_l\|_2 S_{l-1} \\
&= S_{l-1}(1 + \|W_l\|_2) \\
&\leq \prod_{k=1}^{l-1}(1 + \|W_k\|_2)(1 + \|W_l\|_2) \\
&= \prod_{k=1}^{l}(1 + \|W_k\|_2).
\end{aligned}
\tag{146}
$$

So, we get the upper bound of $\|J_L\|_2$ as

$$\|J_L\|_2 \leq \|W_L\|_2 \prod_{k=1}^{L-1}(1 + \|W_k\|_2). \tag{147}$$

### H.4 TRANSFORMER

Vision Transformer (ViT) is a vision model based on the Transformer architecture, which divides images into fixed-size patches and performs global modeling through multi-head attention (MHA). ViT-B-16 is the basic version, using a $16 \times 16$ patch size. ViT-B-16 contains $L = 12$ layers of Transformer Encoder. Next we discuss the spectral norm of the Jacobian matrix of the Encoder model.

The input sequence $X = (x_1, x_2, \cdots, x_n)$ is transformed by the embedding layer and positional encoding to

$$H^{(0)} = \text{Embed}(X) + \text{PositionalEncoding}, \tag{148}$$

where $H^{(0)} \in R^{n \times d}$ and $d$ is the model dimension.

The encoder is composed of $L$ identical layers stacked together, each layer contains:

- Multi-Head Attention (MHA)

$$\text{MHA}(H) = \text{Concat}(\text{head}_1, \cdots, \text{head}_h)W^O. \tag{149}$$

  Each attention head $\text{head}_i = \sigma\left(\frac{Q_i K_i^T}{\sqrt{d_k}}\right)V_i$, where $Q_i = HW_i^Q$, $K_i = HW_i^K$, $V_i = HW_i^V$ and $W^Q, W^K, W^V \in \mathbb{R}^{d \times d_k}$.

- Feed-Forward Network (FFN)

$$\text{FFN}(H) = \text{ReLU}(HW_1 + 1b_1^T)W_2 + 1b_2^T, \tag{150}$$

- Residual Connection and Layer Normalization

$$
\begin{aligned}
H^{(l)} &= \text{LN}(H^{(l-1)} + \text{MHA}(H^{(l-1)})) \tag{151} \\
H^{(l)} &= \text{LN}(H^{(l)} + \text{FFN}(H^{(l)})). \tag{152}
\end{aligned}
$$

Given an input $H \in \mathbb{R}^{n \times d}$, where $n$ is the sequence length and $d$ is the feature dimension, Self-Attention will be represented as follows

$$Q = HW^Q, \quad K = HW^K, \quad V = HW^V,$$

$$S = \frac{QK^T}{\sqrt{d_k}}, \quad A = \sigma(S), \quad \text{Attention}(Q, K, V) = AV, \tag{153}$$

where $W^Q, W^K, W^V \in \mathbb{R}^{d \times d_k}$ are the learnable weight matrices, $A \in \mathbb{R}^{n \times n}$ is the row-normalized attention weight matrix, and $\text{Attention}(Q, K, V) \in \mathbb{R}^{n \times d_k}$ is the output of Self-Attention.

We first calculate the gradient with respect to the value $H$ from $V$

$$\left\| \frac{\partial \text{Attention}}{\partial H} \right\|_2 \le \left\| \frac{\partial \text{Attention}}{\partial V} \right\|_2 \left\| \frac{\partial V}{\partial H} \right\|_2 = \|A\|_2 \|W^V\|_2. \tag{154}$$

Note that $A \in \mathbb{R}^{m \times n}$ is a row-normalized matrix, and all elements of A are positive. Therefore, according to the Gerschgorin disk theorem, we have that any eigenvalue of the matrix A is located in a Gerschgorin disk

$$|\lambda - A_{ii}| \le \sum_{j \ne i} |A_{ij}|, \quad i = 1, 2, \cdots, m. \tag{155}$$

That is, $-1 + A_{ii} \le \lambda \le 1$. So we immediately get $\|A\|_2 \le 1$. At the same time, we observe that $A\mathbf{1} = \mathbf{1}$, then 1 is an eigenvalue of $A$, and thus $\|A\|_2 = 1$.

Next we calculate the gradient of **Attention** with respect to $H$ from the attention weight $A$

$$\left\| \frac{\partial \text{Attention}}{\partial H} \right\|_2 = \left\| \frac{\partial \text{Attention}}{\partial A} \frac{\partial \sigma}{\partial S} \frac{\partial S}{\partial H} \right\|_2$$

$$\le \|V\|_2 \cdot \frac{1}{n} \cdot \left\| \frac{1}{\sqrt{d_k}} \left( \frac{\partial Q}{\partial H} K^T + Q \frac{\partial K^T}{\partial H} \right) \right\|_2$$

$$\le \frac{2}{n\sqrt{d_k}} \|W^V\|_2 \|W^Q\|_2 \|W^K\|_2 \|H\|_2^2. \tag{156}$$

The input $H$ is normalized, so $\|H\|_2$ is bounded. In general, $n$ and $d_k$ are very large, then we have

$$\|J_{\text{attn}}\|_2 \le \|W^V\|_2 + \frac{2}{n} \frac{\|W^V\|_2 \|W^Q\|_2 \|W^K\|_2 \|H\|_2^2}{\sqrt{d_k}} \approx \|W^V\|_2. \tag{157}$$

According to the estimation of the spectral norm of the block matrix (as shown in Eqn. (135)), we have ($h = 8$)

$$\|J_{\text{MHA}}\|_2 \le \sqrt{h} \max_i \|W_i^V\|_2 \|W^O\|_2. \tag{158}$$

According to the properties of the spectral norm, we immediately have

$$\|J_{\text{FFN}}\|_2 = \left\| \frac{\partial \text{FFN}}{\partial H} \right\|_2 \le \|W_2\|_2 \|J_{\text{ReLU}}\|_2 \|W_1\|_2 = \|W_1\|_2 \|W_2\|_2. \tag{159}$$

Note that when we use a transformer for classification, we do not use the decoding layer. Combining our previous analysis and conclusions, we have

$$\|J_{\text{transformer}}\|_2 \le \prod_{l=1}^{L} (1 + \sqrt{h} \max_i \|W_i^V\|_2 \|W^O\|_2 + \|W_{l1}\|_2 \|W_{l2}\|_2). \tag{160}$$

# I PROPERTIES OF HUTCHINSON'S ALGORITHM

## I.1 CONVERGENCE OF HUTCHINSON'S ALGORITHM FOR SOLVING SPECTRAL NORM

**Theorem 9** *Let $R(A, x_i) = \frac{x_i^T A x_i}{x_i^T x_i}$, given $m$ independent random vectors $x_1, \cdots, x_m$, when $m \to \infty$, then $\hat{\lambda}_{\max} = \max_{i=1}^{m} R(A, x_i)$ will converge to $\lambda_{\max}(A)$ with high probability. For any given $\delta$*

*value, when*

$$m \geq \frac{\log \frac{1}{\delta}}{p_\epsilon}, \tag{161}$$

*then*

$$P(\hat{\lambda}_{\max} \geq \lambda(A) - \epsilon) = 1 - (1 - p_\epsilon)^m \geq 1 - \delta, \tag{162}$$

*where $p_\epsilon = P(R(A, x_i) \geq \lambda_{\max}(A) - \epsilon)$.*

Defining Rayleigh entropy $R(A, x) = \frac{x^T A x}{x^T x}$, then for a symmetric matrix $A$, we have

$$\lambda_{\min}(A) \leq R(A, x) \leq \lambda_{\max}(A), \quad \forall x \neq 0. \tag{163}$$

Furthermore, we have $\lambda_{\max}(A) = \max_{x \neq 0} R(A, x)$.

**Coverage of random vectors** Assume $z \sim N(0, I_n)$ is a standard Gaussian random variable, $v_{\max}$ is the largest eigenvector (unit vector) of $A$, then $z = x^T v_{\max}$ is also a Gaussian random variable. We know that the probability of a continuous random variable at any single point is 0, so we have

$$P(z = 0) = P(x^T v_{\max} = 0) = 0. \tag{164}$$

That is, $P(x^T v_{\max} \neq 0) = 1$, so $x$ can be decomposed into

$$x = (x^T v_{\max}) v_{\max} + x_\perp, \quad x_\perp \perp v_{\max}. \tag{165}$$

When $x \to v_{max}$, that is, $R(A, x) \to \lambda_{\max}(A)$.

**Concentration of Rayleigh Entropy** Since $R(A, x)$ is a continuous function and $R(A, v_{\max}) = \lambda_{\max}(x)$, there exists a neighborhood $B_\delta(v_{\max})$ of $v_{\max}$ such that for any $x$

$$\|x - v_{\max}\| \leq \delta \Rightarrow R(A, x) \geq \lambda_{\max}(A) - \epsilon. \tag{166}$$

It is worth noting that the probability that a Gaussian random variable $x$ falls in the neighborhood is positive

$$P(\|x - v_{\max}\| < \delta) > 0. \tag{167}$$

So we have

$$P(R(A, x) \geq \lambda_{\max}(A) - \epsilon) \geq P(\|x - v_{\max}\| < \delta) > 0. \tag{168}$$

**Convergence of the maximum value** We take $m$ independent random variables $x_1, \cdots, x_m$ and define

$$\hat{\lambda} = \max_{i=1}^{m} R(A, x_i). \tag{169}$$

Since $P(R(A, x_i) \geq \lambda_{\max}(A) - \epsilon) = p_\epsilon > 0$, then

$$P(R(A, x_i) \leq \lambda_{\max}(A) - \epsilon) = 1 - p_\epsilon, \quad i = 1, 2, \cdots, m. \tag{170}$$

So we obtain

$$P(\hat{\lambda}_{\max} \leq \lambda(A) - \epsilon) = P\left(\max_{i=1}^{m} R(A, x_i) \leq \lambda(A) - \epsilon\right) = (1 - p_\epsilon)^m. \tag{171}$$

When $m \to \infty$, then $(1 - p_\epsilon)^m \to 0$. In other words, there is

$$P(\hat{\lambda}_{\max} \geq \lambda(A) - \epsilon) = 1 - (1 - p_\epsilon)^m \to 1. \tag{172}$$

**High probability convergence** Now, given $\delta \in (0, 1)$, we ask for the probability

$$
\begin{aligned}
P(\hat{\lambda}_{\max} \geq \lambda(A) - \epsilon) \quad &= \quad 1 - (1 - p_\epsilon)^m \geq 1 - \delta \\
&\Rightarrow \quad (1 - p_\epsilon)^m \leq \delta \\
&\Rightarrow \quad \left(\frac{1}{1 - p_\epsilon}\right)^m \geq \frac{1}{\delta} \\
&\Rightarrow \quad m \log\left(\frac{1}{1 - p_\epsilon}\right) \geq \log\left(\frac{1}{\delta}\right) \\
&\Rightarrow \quad m \geq \frac{\log \frac{1}{\delta}}{-\log(1 - p_\epsilon)}
\end{aligned}
\tag{173}
$$

Since $-\log(1 - p_\epsilon)$ is a convex function of $p_\epsilon$, according to the convex function $f(x)$ satisfying $f(x) \geq f(0) + f'(0)x$, we know that

$$-\log(1 - p_\epsilon) \geq p_\epsilon. \tag{174}$$

So when

$$m \geq \frac{\log \frac{1}{\delta}}{p_\epsilon} \geq \frac{\log \frac{1}{\delta}}{-\log(1 - p_\epsilon)}, \tag{175}$$

we have

$$P(\hat{\lambda}_{\max} \geq \lambda(A) - \epsilon) = 1 - (1 - p_\epsilon)^m \geq 1 - \delta. \tag{176}$$

## I.2 ALIGNMENT OF RANDOMLY SAMPLED VECTORS WITH THE PRINCIPAL EIGENVECTOR OF A MATRIX

**Theorem 10** *Let $u, v \in \mathbb{R}^n$, where $u$ is a random unit vector and $v$ is a fixed unit vector (such as the principal eigenvector of a matrix), then the probability that $u$ is aligned with $v$ decays exponentially with $n$. Specifically, we have*

$$P(|u^T v| \geq t) \leq 2 \exp\left(-cnt^2\right), \tag{177}$$

*where $c$ is a universal constant.*

Let $v \in \mathbb{R}^n$ be a fixed unit vector corresponding to the principal eigenvector of the matrix $A$, and $u$ be a uniform random unit. Below we use $u^T v$ to denote the degree of alignment of $u$ with $v$.

**Expectation and variance of the inner product** Since u is uniformly randomly distributed, its direction distribution is symmetrical, that is, for any $u^v = c$, there exists $(-u)^T v = -c$. Therefore

$$\mathbb{E}(u^T v) = 0. \tag{178}$$

Furthermore, we calculate the variance $\text{var}(u^T v)$ of $u^T v$

$$
\begin{aligned}
\text{var}(u^T v) = \mathbb{E}[(u^T v - \mathbb{E}(u^T v))^2] &= \mathbb{E}[(u^T v)^2] \\
&= \mathbb{E}\left[\left(\sum_{i=1}^n u_i v_i\right)^2\right] \\
&= \sum_{i=1}^n v_i^2 \mathbb{E}[u_i^2] + \sum_{i \neq j} v_i v_j \mathbb{E}[u_i u_j].
\end{aligned} \tag{179}
$$

Note that since $u$ is uniformly randomly distributed, all $\mathbb{E}[u_i^2]$ are equal, as shown by $\sum_{i=1}^n u_i^2 = 1$

$$\sum_{i=1}^n \mathbb{E}[u_i^2] = 1 \Rightarrow \mathbb{E}[u_i^2] = \frac{1}{n}. \tag{180}$$

Therefore, the variance of $u^T v$ is

$$\text{var}(u^T v) = \frac{1}{n} \sum_{i=1}^n v_i^2 = \frac{1}{n}. \tag{181}$$

Let the standard deviation $\sigma = \sqrt{\text{var}(u^T v)} = \frac{1}{\sqrt{n}}$. For most probability distributions, such as the Gaussian distribution, $u^T v$ will have a probability of 95% falling within the interval $[-2\sigma, +2\sigma]$. Therefore, $|u^T v|$ is usually no more than $O(\frac{1}{\sqrt{n}})$.

**Concentration Inequality** Levy's lemma Milman & Schechtman (1986) states that for a Lipschitz function on a high-dimensional sphere, its values are highly concentrated near the desired value. Specifically,

**Lemma 1** *Levy's Lemma: Assume that $f : \mathbb{S}^{n-1} \to \mathbb{R}$ is an L-Lipschitz function (i.e., $|f(u) - f(u')| \leq L\|u - u'\|_2$), and $u$ is uniformly distributed on the unit sphere $\mathbb{S}^{n-1}$, then*

$$P(|f(u) - \mathbb{E}[f]| \geq t) \leq 2\exp\left(-\frac{cnt^2}{L^2}\right), \tag{182}$$

*where c is a universal constant (e.g. $c = 1/2$).*

We define the function $f(u) = u^T v$, where $v$ is a fixed unit vector, so we have

$$|f(u) - f(u')| = |(u - u')^T v| \leq \|u - u'\|_2 \|v\|_2 = \|u - u'\|_2. \tag{183}$$

That is, the function $f$ is an L-Lipschitz function, where $L = 1$. From the symmetry of $f$, we know that $\mathbb{E}[f] = 0$, so applying Levy's Lemma we can get

$$P(|u^T v| \geq t) \leq 2\exp\left(-cnt^2\right). \tag{184}$$

It can be seen that the alignment of the random unit vector $u$ with the fixed unit vector (the principal eigenvector of the matrix) decays exponentially as $d$ increases.

## J  OVERVIEW AND ANALYSIS OF ALGORITHMS

We innovatively apply three algorithms based on the low-rank structure of $F(x)$. We make full use of the associative property of matrix multiplication and the property of the spectral norm $\|AA^T\| = \|A^T A\|$ in our algorithm to indirectly estimate $B(x) = \Lambda^{1/2} Q^T Q \Lambda^{1/2}$ rather than $F(x) = Q\Lambda Q^T$. In the power iteration algorithms, when computing $b_{t+1} = F(x)b_t$, we compute $(Q(\Lambda(Q^T b_t)))$ instead of $Q\Lambda Q^T b_t$. Computing according to $(Q(\Lambda(Q^T b_t)))$ has lower space complexity. Note that the indirect estimation makes the approximation error of the Hutchinson algorithm smaller.

The following table compares the space complexity and time complexity of direct and indirect estimation of $F(x)$ ($d \gg K$)

Table 11: Time and space complexity analysis of indirect estimation of $\|F\|_2$

| Indirect Estimation | Space complexity | Time complexity |
|---|---|---|
| Eigendecomposition | $O(dK)$ | $O(dK^2 + K^3)$ |
| Power Iteration | $O(dK)$ | $O(TdK)$ |
| Hutchinson Approximation | $O(dK)$ | $O(dK)$ |

Table 12: Time and space complexity Analysis of direct estimation of $\|F\|_2$

| Direct Estimation | Space complexity | Time complexity |
|---|---|---|
| Eigendecomposition | $O(d^2)$ | $O(Kd^2 + d^3)$ |
| Power Iteration | $O(d^2)$ | $O(TdK)$ |
| Hutchinson Approximation | $O(dK)$ | $O(dK)$ |

Overall, our innovative application of the three algorithms generally significantly reduces the time and space complexity of the algorithms, making our robustness metrics more feasible in large-scale practical applications.

## K  THEORETICAL VERIFICATION EXPERIMENT

We use two common and popular datasets for image classification: CIFAR10 Krizhevsky (2009) and MNIST LeCun et al. (1998). CIFAR10 contains 10 categories and a total of 60,000 color images of size $32 \times 32$. MNIST is a handwritten digit image dataset containing 60,000 training images and 10,000 test images, each sample size is $28 \times 28$ pixels. Our programs are all run on a server equipped with a GeForce RTX 3090 with 24G video memory. We select 4 classic base models including DenseNet121, VGG16, ResNet18, ViT-B-16 to study our proposed robustness metric based on the spectral norm.

---

**Algorithm 1** Power Iteration for the Principal Eigenvalue

---

1: **Input** : $Q$ and $\Lambda$
2: Initialize $j = \arg\max_k p(y_k|x)$, $b_0 = \nabla_x \log p(y_j|x) \in \mathbb{R}^d$
3: $\lambda_{\text{prev}} = p(y_j|x)$ and $b_0 = b_0/\|b_0\|$
4: **for** $t = 0, 1, 2, \cdots, T$ **do**
5: $\quad b_{t+1} = Q(\Lambda(Q^T b_t))$
6: $\quad \lambda_t = b_t^T b_{t+1}$
7: $\quad$ **if** $(\|\lambda_t - \lambda_{\text{prev}}\|)/\|\lambda_t\|_2 < \epsilon$ **then**
8: $\quad\quad$ **break**
9: $\quad$ **end if**
10: $\quad \lambda_{\text{prev}} = \lambda_t$
11: $\quad b_{t+1} = \frac{b_{t+1}}{\|b_{t+1}\|}$
12: **end for**
13: **return** $\lambda_t$

---

---

**Algorithm 2** Hutchinson Algorithm for the Principal Eigenvalue

---

1: **Input** : $Q$ and $\Lambda$
2: $\Gamma = \Lambda^{1/2}$
3: **for** $i = 1, 2, \cdots, M$ **do**
4: $\quad$ Generate a random vector $z$ where $z_i \sim N(0,1)$
5: $\quad y = Q\Gamma z$
6: $\quad \lambda_i = \frac{y^T y}{z^T z}$
7: **end for**
8: **return** $\max_i \lambda_i$

---

### K.1 ANALYSIS OF SPECTRAL ROBUSTNESS MEASURE

**FIM and variance of Jacobian matrix** To verify the theorem $F(x) = \text{var}(J(x))$, we estimate the variance of $J(x)$ by sampling $J(x)$ to verify its correctness. We design a toy model consisting of a simple single-layer convolution layer + fully connected layer + softmax layer. By generating random inputs of different batch sizes, we calculate the estimated variance $\hat{\sigma}^2(x)$ of $F(x)$ and $J(x)$ respectively, and finally estimate the approximate error of $\frac{\|F(x) - \hat{\sigma}^2(x)\|_F}{\|F(x)\|_F}$ through relative error. The results in Table 13 conform to the law of large numbers: as the number of samples increases, the estimated value of the random variable approaches its true value.

Table 13: Error between FIM and variance of Jacobian matrix vs sampling size

| #Size | 32 | 64 | 128 | 256 | 512 | 1024 |
|-------|-----|-----|-----|-----|-----|------|
| Error | 1.8552 | 1.5554 | 1.3644 | 1.2530 | 1.1921 | 1.1600 |

**Model robustness and number of model layers** Through model analysis, we know that when the model components are the same, the model robustness $R_{\text{spec}}$ is inversely proportional to the number $L$ of layers of the model components (e.g. $O(1/L)$). Resnet18 and Resnet34 have the same components (Basic Block), as shown in Table 14, when the number of layers of the model increases, the robustness metric of the model decreases.

Table 14: Comparison of robustness measures for models with the same components

| $R_{\text{spec}}$ | **ResNet18** | **ResNet34** |
|-------------------|--------------|--------------|
| CIFAR10 | 9.610 | 3.162 |
| MNIST | 1.285 | 0.763 |

**Analysis on the metric $R_{\text{spec}}$**  From the results in Table 16, we can see that ViT has the highest robustness metric, while DenseNet has the worst robustness. The performance of the robustness metric on the CIFAR10 dataset is consistent with our theoretical results, while in the results on MNIST, VGG16 is more robust than ResNet18. This may be because the gradient of VGG16 on simple MNIST images is smoother, making its spectral norm smaller.

**Robustness metric $R_{\text{spec}}$ and robustness metric based on Lipschitz constant**  To analyze the relationship between $R_{\text{spec}}$ and the classic robustness measure based on the Lipschitz constant, we count the estimated value $1/\|F(x)\|_2$ and the Lipschitz constant on each sample $x$ in the data set, and then calculate the Pearson correlation coefficient of the two sequences, as shown in the table. This further verifies that there is a linear correlation and consistency of evaluation between our measure $R_{\text{spec}}$ and Lipschitz constant robustness measure.

Table 15: Comparison of robustness measures for multiple models

| Dataset | CIFAR10 | MNIST |
|---|---|---|
| ViT-B-16 | 11.44 | 35.66 |
| ResNet18 | 9.61 | 1.28 |
| VGG16 | 1.04 | 5.07 |
| DenseNet | 1.40 | 0.06 |

Table 16: Pearson and Spearman values (in brackets) analysis between $R_{\text{spec}}$ and Lipschitz constant robustness measure

| Dataset | CIFAR10 | MNIST |
|---|---|---|
| ViT-B-16 | 0.90 (0.95) | 0.88 (0.88) |
| ResNet18 | 0.86 (0.90) | 0.82 (0.84) |
| VGG16 | 0.85 (0.89) | 0.80 (0.82) |
| DenseNet | 0.84 (0.91) | 0.80 (0.83) |

### K.2  SOLUTION ON SPECTRAL NORM OF FIM

**Comparison of algorithm running times**  We propose to use three different types of algorithms to solve the spectral norm of the FIM matrix to cope with models of different sizes.

We set the number of parameters to 1e5, the number of iterations in the power iteration algorithm and the number of samples in the Hutchinson algorithm to 1000. Then, we randomly generate a Gaussian distribution and run 10 times with the number of categories to average the running time, and obtain Table 17. From Table 17, we can see that when the category (model output) scale is small, we can directly resort to eigenvalue decomposition, which is usually faster; when the category is of medium size, power iteration may have an advantage; and when the category scale is very large, the Hutchinson algorithm may be more efficient.

Table 17: Performance (Time) comparison of algorithms vs classes (D: direct eigenvalue decomposition, P: power iteration, and H: Hutchinson algorithm)

| #Classes(K) | D (s) | P (s) | H (s) |
|---|---|---|---|
| 10 | 0.0016 | 0.0136 | 0.2988 |
| 100 | 0.0070 | 0.0233 | 0.0265 |
| 1000 | 0.2649 | 0.1964 | 0.1378 |
| 10000 | 30.3127 | 1.3501 | 1.2813 |

**Hutchinson's convergence**  Theorem 4 shows that when the number of samplings $M$ approaches $\infty$, $\hat{\lambda}_{\max} = \max_{i=1}^m R(A, x_i)$ will approach $\lambda_{\max}(A)$ with high probability. Next we will verify this conclusion through experiments.

We set the number of categories $K = 10$ and the dimension of the parameters to $1e5$, and then randomly generate the Gaussian distribution matrix $Q$ and the diagonal matrix $\Lambda$. When the Hutchinson random algorithm is run 10 times in parallel on the GPU with different sampling times, the statistical average approximation error $100 * |\hat{\lambda}_{\max} - \lambda_{\max}|/\lambda_{\max}$ and average running time are shown in the table. We can clearly see that when $M$ increases, the approximation error continues to decrease.

Table 18: Approximation Error of Hutchinson's Algorithm

| #Samples (M) | Time (s) | Error (%) |
|---|---|---|
| 10 | 0.3100 | 35.66 |
| 100 | 0.0038 | 26.06 |
| 1000 | 0.0055 | 12.69 |
| 10000 | 0.0515 | 9.26 |

**Comparison of Hutchinson's algorithm for solving $\|Q\Lambda Q^T\|_2$ and $\|\Lambda^{1/2}Q^TQ\Lambda^{1/2}\|_2$**  Theorem 5 tells us that the probability of aligning a random vector generated by the Hutchinson algorithm with the true value $\lambda_{\max}$ decays exponentially with the dimensional size $n$. The following experiment

shows why we choose to use $\Lambda^{1/2}Q^TQ\Lambda^{1/2}$ as the input of the Hutchinson algorithm instead of $Q\Lambda Q^T$, even though they theoretically have the same spectral norm.

In Hutchinson, we choose the dimension of the parameter as $d = 10000$ and the number of samplings as 1000. We randomly generate Q and $\Lambda$ that follow a Gaussian distribution, select different numbers of categories, and use $\Lambda^{1/2}Q^TQ\Lambda^{1/2}$ and $Q\Lambda Q^T$ as the input of Hutchinson to run the Hutchinson algorithm, as shown in Table 19. It can be seen that the approximation error using $\Lambda^{1/2}Q^TQ\Lambda^{1/2}$ as input is related to the number of categories, while the approximation error using $Q\Lambda Q^T$ as input is related to the dimension of the parameter.

Table 19: Approximation error of Hutchinson's algorithm with $Q\Lambda Q^T$ and $\Lambda^{1/2}Q^TQ\Lambda^{1/2}$ as input

| #Classes(K) | $\Lambda^{1/2}Q^TQ\Lambda^{1/2}$ | $Q\Lambda Q^T$ |
|---|---|---|
| 10 | 13.91 | 99.83 |
| 100 | 60.75 | 99.52 |
| 1000 | 73.85 | 97.19 |

However, the dimension of the parameters is constant, so we see that as the number of categories increases, the approximation error (or probability of alignment) of $\Lambda^{1/2}Q^TQ\Lambda^{1/2}$ as input decreases, while the approximation error of $Q\Lambda Q^T$ as input remains approximately the same.

**Sampling in Hutchinson approximation algorithm** Hutchinson has good theoretical properties by generating Rademacher random variables, but in practice, sampling Gaussian random variables has better convergence properties. The following experimental results (Table 20) show that Gaussian random variables have lower approximation errors than Rademacher random variables.

We generate a matrix $Q$ with dimensions $100000 \times 10$ that follows a Gaussian distribution and a diagonal matrix $\Lambda$ that follows a Gaussian distribution, and then sample Gaussian random variables and Rademacher random variables for different times, and compare their approximation errors as shown in Table 20. The results in Table 20 show that Gaussian sampling is much better than Rademacher sampling.

Table 20: Approximation error (%) of Hutchinson algorithm under Gaussian sampling and Rademacher sampling

| #Samples (M) | Normal | Rademacher |
|---|---|---|
| 10 | 32.13 | 56.66 |
| 100 | 26.51 | 61.86 |
| 1000 | 12.06 | 54.09 |
| 10000 | 9.68 | 59.04 |

## K.3 VARIANCE OF ROBUSTNESS MEASURE ESTIMATE

According to the description and setting of the estimation measure above, CLEVER and PGD contain a certain amount of randomness because they need to randomly sample data points. The estimates of other metrics $R_{\text{spec}}$, CW, and Lipschitz constant are all deterministic metrics.

Below we use the results of the clean model $M_{\text{clean}}$ on three data sets and four models as shown in Tables 21 and 22. For each experiment, we sample 500 data points on the data set to count the variance of 5 repeated experiments. From Tables 21 and 22, it can be seen that DenseNet121 has the largest variance on the MNIST data set, and the variances of the others are very small.

Table 21: Variance of PGD measure

| Dataset | ViT_B_16 | ResNet18 | VGG16 | DenseNet121 |
|---|---|---|---|---|
| CIFAR10 | 1.00±0.0000 | 1.00±0.0000 | 0.77±0.0006 | 0.99±0.0006 |
| MNIST | 0.89±0.0000 | 0.91±0.0000 | 0.01±0.0000 | 0.96±0.0013 |
| Tiny-ImageNet | 0.99±0.0000 | 0.99±0.0000 | 1.00±0.0000 | 1.00±0.0010 |

## K.4 COMPARISON OF DIFFERENT TYPES OF DATA

Below we further give the results on CIFAR100, Medical Data (covid19-radiography-database from Kaggle [4]) and ImageNet in Tab. 23 and 24. Comparing the robustness metrics of the two models,

---

[4]https://www.kaggle.com/datasets/tawsifurrahman/covid19-radiography-database

Table 22: Variance of CLEVER Score

| Dataset | ViT_B_16 | ResNet18 | VGG16 | DenseNet121 |
|---|---|---|---|---|
| CIFAR10 | 2.42±0.0005 | 5.09±0.0006 | 2.31±0.0004 | 6.11±0.0041 |
| MNIST | 2.82±0.0035 | 3.17±0.0007 | 0.40±0.0001 | 11.99±0.0105 |
| Tiny-ImageNet | 2.59±0.0003 | 2.39±0.0002 | 13.14±0.0021 | 4.46±0.0027 |

ViT_B_16 and ResNet18, on the same dataset (Medical Data or CIFAR100), we can observe that our metrics and most other metrics give consistent results: ResNet18 < ViT_B_16.

Table 23: Comparison results of the ResNet18 and VIT_B_16 models on Medical data

| Model | $L(x)$ | CLEVER | CW | PGD | $\|F(x)\|_2$ | $R_{\text{spec}}$ |
|---|---|---|---|---|---|---|
| ResNet18 | 0.57 | 5.43 | 37.08 | 98.88 | 5.95 | 36.28 |
| ViT_B_16 | 0.29 | 2.10 | 20.25 | 98.73 | 2.11 | 375.44 |

Table 24: Comparison results of the ResNet18 and VIT_B_16 models on CIFAR100

| Model | $L(x)$ | CLEVER | CW | PGD | $\|F(x)\|_2$ | $R_{\text{spec}}$ |
|---|---|---|---|---|---|---|
| ResNet18 | 0.29 | 1.81 | 62.07 | 94.83 | 0.73 | 5.69 |
| ViT_B_16 | 0.23 | 1.21 | 65.22 | 93.48 | 0.55 | 23.05 |

## K.5 Comparison of robustness metrics on small-scale datasets

We use ResNet18 as the basis to analyze how the six metrics rank the dataset. The results are shown in Tab. 25.

MNIST is a grayscale image with a simple input space, which results in a flat gradient of the model loss function, thus: 1) The model may be prone to overfitting on MNIST, resulting in $L(x) = 0$; 2) Adversarial attacks are difficult to take effect, and the success rate of attacks is extremely low; 3) The model output is very certain, so $\|F(x)\|_2$ is extremely small; 4) $R_{\text{spec}}$ is extremely large, and there are many outliers in the robust value.

If we exclude the outlier data MNIST, our metrics $\|F(x)\|_2$ and $R_{\text{spec}}$ achieve consistent results with other metrics, including $L(x)$, CLEVER, and CW.

Table 25: Comparison of robustness ranking results of ResNet18 using 6 metrics on 3 datasets

| Dataset | $L(x)$ | CLEVER | CW | PGD | $R_{\text{norm}} \downarrow$ | $R_{\text{spec}}$ |
|---|---|---|---|---|---|---|
| CIFAR10 | 0.29 | 2.02 | 29.60 | 86.67 | 0.82 | 186.46 |
| Tiny-Imagenet | 0.21 | 1.72 | 38.64 | 88.64 | 0.51 | 7.25 |
| MNIST | 0.0 | 1.73 | 1.0 | 2.0 | 0.01 | 24510.91 |

