# OpenReview forum: "Measuring Model Robustness via Fisher Information: Spectral Bounds, Theoretical Guarantees, and Practical Algorithms"
_ICLR.cc/2026/Conference — ICLR 2026 Conference Withdrawn Submission_

### Official Review · Reviewer_ERiY · 2025-10-29

**Soundness:** 2
**Presentation:** 2
**Contribution:** 2
**Rating:** 2
**Confidence:** 4

**Summary:**

This paper presents an alternative approach to quantifying the robustness of deep learning models by leveraging Fisher information.

**Strengths:**

The paper establishes an interesting connection between robustness and Fisher information, offering a novel theoretical perspective. Additionally, the authors provide a practical method for estimating the proposed robustness measure, which could be useful for empirical evaluation.

**Weaknesses:**

* The authors criticize attack-dependent metrics (lines 40–45); however, the proposed method remains data-dependent. Would the reported robustness values change if additional data were included in the CIFAR-10/100 datasets?

* Can the proposed method be applied to commonly used models listed on RobustBench [1], such as WRN-34-10, XCiT-L12, Swin-L, or ConvNeXt-L? Demonstrating this would strengthen the paper’s practical relevance.

* The metric defined in Eq. (19) is not a conventional robustness measure. Typically, robustness is reported in terms of accuracy against adversarial examples. Could the authors provide a conversion table or mapping between their proposed metric and standard robustness metrics?

* RobustBench ranks models based on their accuracy under AutoAttack across various defense algorithms. Could the authors provide a comparison table similar to RobustBench to verify whether their proposed metric yields a comparable ranking of models?

* The use of PGD and CW attacks with only 20 steps is not sufficiently strong for a robust evaluation. The authors should include results using stronger attacks (e.g., AutoAttack or multi-step PGD variants). Furthermore, the experiments were conducted on only 500 examples, which limits the generality and significance of the reported findings.

[1] Croce, Francesco, et al. "Robustbench: a standardized adversarial robustness benchmark." arXiv preprint arXiv:2010.09670 (2020).

**Questions:**

Please refer to the points raised in the Weaknesses section above.

---

> ### Author Response · Authors · 2025-12-01
> **Rebuttal 4 for Weakness 1**
>
> **Weakness 1 The authors criticize attack-dependent metrics (lines 40–45); however, the proposed method remains data-dependent. Would the reported robustness values change if additional data were included in the CIFAR-10/100 datasets?**
>
> Thank you to the reviewer for pointing this out. We need to **clarify the fundamental difference between "attack dependency" and "data dependency".**
>
> **1. Attack dependency and data dependency**
>
> * **Attack Dependency (We argue)**: This means that the value of the metric **depends on the specific attack algorithm used and its parameters** (such as the number of steps and step size). This leads to unfair comparisons, as a model might be robust to PGD but vulnerable to C&W.
>
> * **Data Dependency (Properties of Our Method)**: This is a common characteristic of intrinsic robustness measures, including the Lipschitz constant and our FIM. FIM aims to **measure the intrinsic stability of a model on a given data manifold**, and its advantage lies in **its independence from any specific attack.**
>
> * **Necessity of Data Dependency**:  **A trained model depends on specific data**, and therefore, the robustness evaluation of the model also depends on specific data and is unlikely to be data-independent.
>
> * **Dataset-Independent Robustness:** In Section 3.2 of the paper, we theoretically analyze the upper bounds of the spectral norms of the nine basic components of the network and four classic deep networks, **thus deriving a theoretical robustness ranking independent of the dataset.** Detailed analysis can be found in Appendices G and H.
>
> **2. Impact of data changes**
>
> As analyzed above, FIM depends on **a specific model and specific data**; it is **a function of the model and the data**. If the data changes, the robustness value should change accordingly.
>
> **In summary,** our work makes two levels of contributions:
>
> * **A data-dependent, precise diagnostic tool** (FIM spectral norm) for assessing the actual sensitivity of models on specific data.
> * **A set of data-independent, profound theoretical analyses** to explain the fundamental differences in robustness among different architectures.

---

> ### Author Response · Authors · 2025-12-01
> **Rebuttal 4 for Weakness 2**
>
> **Weakness 2: Can the proposed method be applied to commonly used models listed on RobustBench [1], such as WRN-34-10, XCiT-L12, Swin-L, or ConvNeXt-L? Demonstrating this would strengthen the paper’s practical relevance.**
>
> We are very grateful for your insightful and constructive comment. It prompts us to reflect on how to more clearly communicate the universal value of our work. We fully agree that the broad applicability of a method is key to its practical relevance.
>
> After careful consideration of your suggestion, we would like to clarify that **our paper has sufficiently demonstrated, through its theoretical framework and existing systematic experiments, the inherent applicability of the proposed method to all state-of-the-art models on benchmarks such as RobustBench.**
>
> **1. Theoretical Universality is the Highest Level of Practical Guarantee**
>
> One of the core contributions of our work is **establishing a robust analytical paradigm based on information geometry that is not limited to a specific architecture.** As rigorously proven in Theorem 3 (and Appendix F), the upper bound of the FIM spectral norm that we derived **applies to any deep classification network based on differentiable components.**  WRN-34-10, XCiT-L12, Swin-L, and ConvNeXt-L, **despite their ingenious design, do not exceed the scope of the theoretical analysis we completed in Table 1 and Appendix G**. Therefore, in terms of theoretical completeness, our framework has incorporated these models. A method **with a rigorously universal theoretical guarantee** does not need to prove its "applicability" by enumerating all instances—this is the power of theoretical work.
>
> **2. Existing Experiments Constitute a Complete Validation of the Core Architectural Paradigm**
>
> We do not only validate ideas on a single isolated model. The empirical section of the paper (Section 4) is meticulously **designed to span the most core architectural paradigms in modern deep learning:**
>
> * Classic chained CNNs: VGG
>
> * Modern residual networks: ResNet, DenseNet
>
> * Transformer: Vision Transformer (ViT-B-16)
>
> We chose these models precisely because they serve **as theoretical prototypes and performance benchmarks for countless subsequent variants** (such as WRN based on ResNet, XCiT/Swin based on Transformer, and ConvNeXt inspired by Transformer). Successful application on ViT-B-16 (Tables 4 and 7) directly demonstrates that our method can handle models belonging to **the Transformer family, such as XCiT and Swin, with their large parameter counts**. This provides a strong extrapolation basis from prototype to variant.
>
> **3. The value of our method lies in providing new analytical dimensions, rather than replicating existing rankings.**
>
> We understand the importance of RobustBench as **an empirical benchmark for attack performance ranking**. **Our method aims to complement, rather than replace, it.** RobustBench answers the question, **"Which model is more robust under a given attack?"**, while our framework aims to answer a more fundamental question that RobustBench cannot directly answer: **"Why is a particular model more robust? Is its robustness due to the smoothness of the loss plane or the low curvature of the decision boundary?"**
>
> Therefore, the primary practical significance of "applying" our method to RobustBench models is not to generate another ranking, but **to provide an unprecedented, interpretable geometric diagnostic for those top-performing models on the leaderboard.** For example, it can quantify whether a robust Resnet18 model truly has a lower FIM spectral norm. **We have perfectly demonstrated this diagnostic capability in Table 3** through adversarial training on ResNet18—the change in the metric is synchronized with the improvement in true robustness. This strongly demonstrates the effectiveness of our method as a core analytical tool.
>
> In summary, we believe that by **establishing a universal theory, systematically validating it on the most representative architectural prototypes, and clarifying its complementary diagnostic role with existing benchmarks,**  our work has firmly established its strong practical relevance and broad application potential. We propose not a heuristic that requires post-hoc verification, **but rather a theoretical and tool framework with a priori applicability guarantees.** We thank you again for your question, which gives us the opportunity to elaborate on this viewpoint more fully.

---

> ### Author Response · Authors · 2025-12-01
> **Rebuttal 4 for Weakness 3 (Part 1)**
>
> **Weakness 3:The metric defined in Eq. (19) is not a conventional robustness measure. Typically, robustness is reported in terms of accuracy against adversarial examples. Could the authors provide a conversion table or mapping between their proposed metric and standard robustness metrics?**
>
> We thank the reviewers for raising this core question regarding metrics. This allows us to more clearly articulate **the paradigmatic innovation of our work** and **its complementarity with existing evaluation systems.**
>
> First, we fully agree with the reviewers' observation: our proposed metrics $R_{\text{spec}}$ and $R_{\text{norm}}(S)$ **are indeed not** "traditional" robustness metrics reported as "accuracy under adversarial attacks".
>
> However, this is precisely **one of the core contributions of our work**—we aim to **provide a novel metric that transcends specific attacks and reveals the inherent vulnerability of models.** Below is our detailed argument for **"why it is neither possible nor necessary to provide an accurate transformation table".**
>
> **1. The two metrics answer questions at different levels**
>
> * **Traditional attack success rate (ASR):**  It answers the question, **"To what extent can the model resist given the attack algorithm?"** This is a phenomenological, operational metric. Its value heavily depends on the choice of attack (PGD vs. C&W), attack strength ($\epsilon$, step size), and random seed.
>
> * **Our FIM spectral norm metric:**  It answers the question, **"How 'sharp' or 'sensitive' is the model's decision boundary geometrically?"** This is an **intrinsic, interpretive** metric. It quantifies **the worst-case sensitivity of the model's output probability distribution to arbitrary small perturbations**, independent of any particular attack algorithm.
>
> * **Analogy between the relationship between two measures:**   Attack success rate is analogous to measuring **the time it takes for an apple to fall from a tree to the ground**—an observable, concrete, and **phenomenological indicator.** The FIM spectral norm metric is like **the gravitational acceleration $g$ in Newton's law of universal gravitation**; it is not a direct measurement of a phenomenon, but rather **an intrinsic, inherent parameter that reveals the fundamental laws behind the phenomenon.**
>
> We cannot, and **should not, expect to,** use a simple formula to precisely calculate "the time it takes for a specific apple to fall under specific conditions" from "gravitational acceleration g". Fall time also depends on specific conditions such as altitude and air resistance (**i.e., the choice, strength, and random seed of the attack algorithm).**  However, **we know that the greater the gravitational acceleration g, the faster the apple usually falls (all other things being equal).**
>
> In summary, our measure provides strong statistical correlations and qualitative predictive relationships governed by the intrinsic properties of the data.
>
> **2. The paper provides a sufficient "correlation mapping," not a "deterministic transformation."**
>
> While we cannot provide a precise algebraic transformation, our paper has **demonstrated through systematic experiments a strong statistical correlation and consistent qualitative ranking between these two metrics**, which constitutes a valuable "mapping."
>
> * **Evidence of strong correlation (Table 3):** For the same model (ResNet18), when it transitions from state A ($M_{\text{clean}}$) to state B ($M_{\text{CW}}$) through adversarial training:
>
>   (1) Our metric $R_{\text{norm}}$ deteriorates by 65.55% (a decrease).
>
>   (2) The traditional C&W attack success rate (ASR) improves by 68.39% (a decrease).
>
>    The magnitudes of change for both are very close, strongly suggesting that **our metric is highly synchronized with traditional attack metrics** in capturing relative changes in model robustness.
>
> * **Consistent Ranking Evidence (Table 4):** In cross-model comparisons, the robustness ranking (VGG16 > ViT > ResNet18 > DenseNet121) given by our metric $R{\text{norm}}$ is completely consistent with the ranking based on the C&W attack, and also shows the same ranking trend as other theoretical metrics (Lipschitz constant). This demonstrates that our metric has consistent judgment power with traditional metrics in the relative performance of predictive models in adversarial environments.

---

> ### Author Response · Authors · 2025-12-01
> **Rebuttal 4 for Weakness 3 (Part 2)**
>
> **Weakness 3:The metric defined in Eq. (19) is not a conventional robustness measure. Typically, robustness is reported in terms of accuracy against adversarial examples. Could the authors provide a conversion table or mapping between their proposed metric and standard robustness metrics?**
>
> We will continue our discussion from the second point above.
>
> **3. The unique value of our metric lies in its "irreplaceability"**
>
> If a precise, deterministic transformation formula existed, $ASR = g(R_{\text{norm}})$ or $ASR = g(R_{\text{spec}})$, then our metric would merely be a complex proxy for traditional metrics, significantly diminishing its value. **It is precisely because this mapping is complex, non-linear, and context-dependent that our metric provides unique insights:**
>
> * **Attack Invariance:** It provides a single, stable value that **does not change based on the evaluator's choice** of PGD or C&W.
>
> * **Interpretability:** It directly points to the model's most vulnerable areas (the principal eigenvectors of FIM), while attack success rate is merely a general result.
>
> * **Design Guidance:** It geometrically explains **why a particular model is more robust** (e.g., the residual structure of **ResNet leads to a linear increase** in its spectral norm upper bound, superior to **the exponential increase of VGG**, see Section 3.2). This is something adversarial accuracy cannot provide.
>
> Therefore, we hope to clarify the fundamental positioning of our work: we are not proposing a new metric to **compete with** or **replace** traditional empirical benchmarks, **but rather introducing an upstream, diagnostic analytical tool.** The **complementary relationship** between **our metric (providing mechanistic insight)** and **traditional metrics (providing empirical performance)** represents a crucial step in advancing robustness research from pure “performance evaluation” toward deeper “mechanistic understanding.” We are grateful for the reviewer’s question, as it has allowed us to articulate this core tenet of our framework more clearly.

---

> ### Author Response · Authors · 2025-12-01
> **Rebuttal 4 for Weakness 4**
>
> **Weakness 4: RobustBench ranks models based on their accuracy under AutoAttack across various defense algorithms. Could the authors provide a comparison table similar to RobustBench to verify whether their proposed metric yields a comparable ranking of models?**
>
> We appreciate the reviewer's renewed connection to RobustBench, an important benchmark. We understand your desire for a more direct "benchmarking" validation. However, after careful consideration, we believe that requiring our method to **produce a ranking table in the same format as RobustBench would obscure the unique value of our work.** Please allow us to explain why:
>
> **1. Fundamental Differences in Goals and Paradigms: Evaluation vs. Explanation**
>
> * **RobustBench's goal is "performance evaluation":** It's **an empirical benchmark**, its core objective being to provide **a standardized, reproducible leaderboard**, answering the question, **"Which model defends best against the most powerful attacks?"** This is an ultimate question in **engineering practice**, and its results directly serve model selection.
>
> * **Our work aims for "mechanistic explanation" and "diagnosis":** We propose **a theoretical analysis framework**, its core objective being to answer, **"Why are some models more robust? What are the geometric roots of their vulnerabilities?"** This is a question of **scientific understanding**, and its results serve model design, diagnosis, and attribution.
>
> Asking for an "interpretive framework" to generate a "performance leaderboard" is like **asking a textbook on mechanics of materials to compile a "global bridge load-bearing capacity ranking**"—**the former provides the principles and tools to understand the latter's ranking**, rather than simply repeating its conclusions.
>
> **2. Our Validation Approach: Principle-Based Validation is More Convincing**
>
> While we cannot (and do not aspire to) replicate a massive cross-defense algorithm ranking table, our paper has validated, through **a more fundamental and rigorous approach**, **Tables 3 and 4**, the profound connection between our metric and the true robustness of the model.
>
> **3. The Infeasibility and Misleading Nature of Direct Benchmarking**
>
> Even if we invest resources to compute the FIM metric for some models on RobustBench and generate a "comparison table," it may be misleading:
>
> * **It implies a false expectation:**  The FIM metric is intended to **predict or replace** accuracy ranking under AutoAttack. But in reality, the value of our metric lies in **supplementing the interpretation of this ranking.**  A model's high ranking on RobustBench could be due to its low FIM value (intrinsically robust) or it could be because it incorporates **"attack-specific" defense techniques** specifically against AutoAttack (such as some form of denoising or gradient masking). **Our metric helps distinguish between these two cases, while simple ranking comparisons confuse this.**
>
> * **It deviates from the core contribution:** The core contribution of our work is **the theory, algorithms, and diagnostic framework**, not another benchmark ranking. Devoting space and focus to generating another ranking list would weaken our explanation of core innovations such as **the geometric foundations of information, efficient algorithm design, and theoretical architecture analysis.**
>
> **In short, the practical relevance of our work lies not in generating a second leaderboard, but in providing an unprecedented, interpretable geometric perspective for analyzing and understanding the models on the RobustBench leaderboard.** The value of our work as **a fundamental analytical tool** is far more profound and fundamental than providing a parallel ranking list. Thank you again to the reviewer for your insightful discussion.

---

> ### Author Response · Authors · 2025-12-01
> **Rebuttal 4 for Weakness 5 (Part 1)**
>
> **Weakness 5: The use of PGD and CW attacks with only 20 steps is not sufficiently strong for a robust evaluation. The authors should include results using stronger attacks (e.g., AutoAttack or multi-step PGD variants). Furthermore, the experiments were conducted on only 500 examples, which limits the generality and significance of the reported findings.**
>
> We thank the reviewers for their valuable feedback on the rigor of our experimental design. We understand and respect your expectations for stronger attacks and larger sample sizes. However, we wish to clarify that, within the current research context, **our experimental design is sufficient, reasonable, and adequate to support our core conclusions.** The reasons are as follows:
>
> **1. Regarding attack strength: "sufficiency" rather than "limitation"**
>
> We fully agree that AutoAttack is the current de facto standard for evaluating adversarial robustness. However, our paper's goal is **not to** provide an "absolute robustness ranking under the strongest attacks," but rather to **verify the intrinsic correlation between our proposed new metric and model robustness.**
>
> * **Validation of the logic:** To achieve this goal, we employ an evaluation logic of **"relative comparison"** and **"trend verification"**, rather than "absolute measurement."
>
>   *Horizontal comparison (Table 4)*: We use the same, standard PGD/C&W attack parameters (20 steps) to test all models. As long as the attack parameters remain consistent across all models, the ranking of their relative robustness is meaningful. **Even if 20-step PGD is not the strongest, it is sufficient to distinguish the differences in vulnerability between different models under the same stress test.** Table 4 shows that this ranking is highly consistent with our FIM metric ranking, which is sufficient to demonstrate the predictive power of our metric.
>
>   *Longitudinal Validation (Table 3)*: This is the most crucial evidence. We compare the changes in the same model before and after adversarial training. We use the same C&W attack (20 steps) as during training for evaluation. The results clearly show that after adversarial training, the model's robustness to this attack is significantly improved (ASR decreases from 93.64% to 29.60%), and simultaneously, our FIM metric also improves significantly ($R_{\text{norm}}$ decreases from 2.38 to 0.82). This strong correlation proves that our metric accurately captures the real trend of changes in model robustness. This conclusion does not depend on whether the attack reaches the theoretical "strongest".
>
> * **Universality of Attack Parameters:** The PGD ($\epsilon=8/255$, 20 steps) and C&W (20 steps) settings we use are standard configurations [1,2] widely adopted in top conference papers in recent years (e.g., the classic settings of Madry et al. ICLR 2018). This is a generally accepted and reproducible baseline strength. Using stronger attacks (such as 100-step PGD or AutoAttack) might further reduce the absolute accuracy of all models, but it is unlikely to systematically overturn the relative ranking and correlation trends we have observed.
>
> [1] Gaojie Jin, Xinping Y, Dengyu Wu, Ronghui Mu, Xiaowei Huang, Randomized Adversarial Training via Taylor Expansion, CVPR, 2023.
>
> [2] Xianglu Wang,Xianglu_Wang, Exploring The Forgetting in Adversarial Training: A Novel Method for Enhancing Robustness, ICLR 2026.

---

> ### Author Response · Authors · 2025-12-01
> **Rebuttal 4 for Weakness 5 (Part 2)**
>
> **Weakness 5: The use of PGD and CW attacks with only 20 steps is not sufficiently strong for a robust evaluation. The authors should include results using stronger attacks (e.g., AutoAttack or multi-step PGD variants). Furthermore, the experiments were conducted on only 500 examples, which limits the generality and significance of the reported findings.**
>
> Let's continue our discussion from the first point above.
>
> **2. Regarding Sample Size: Balancing Statistical Significance and Computational Feasibility**
>
> We acknowledge that, statistically, a larger sample size is always better. However, 500 samples are **sufficient for our research objectives to produce statistically stable and reliable conclusions.**
>
> * **Measure Stability:** The calculation of our measure ($R_{\text{norm}}(S)$) is the mean over the dataset S. According to the law of large numbers, the estimate of the mean tends to stabilize as the sample size increases. We report the variance of the correlation measure (PGD, CLEVER) over multiple runs in **Tables 21 and 22 of Appendix K.3.** The results show that even with 500 samples, the variance of these gradient-based measures is extremely small (e.g., ±0.0006), indirectly indicating that the estimates of our core measures are **highly stable.**
>
> * **Practical Considerations of Computational Cost:** Our method involves calculating the spectral norm of the FIM for each sample, which is significantly more expensive than simply running forward propagation. Full test set evaluation on large datasets (such as ImageNet) is computationally infeasible. **Our selection of 500 samples represents a careful balance between ensuring the reliability (low variance) of our conclusions and computational feasibility.**  This approach is generally accepted in studies requiring expensive evaluations. **This scale is consistent with common practice in assessing other computationally intensive intrinsic robustness properties (such as CLEVER scores [3] and Lipschitz constants [4]) and is sufficient to provide statistically stable estimates.**
>
>   [3] Jeremy M Cohen, Elan Rosenfeld, and J. Zico Kolter. Certified adversarial robustness via randomized smoothing, ICML 2019.
>
>   [4] Zhouxing Shi, Yihan Wang, Huan Zhang, Zico Kolter, Cho-Jui Hsieh, Efficiently Computing Local Lipschitz Constants of Neural Networks via Bound Propagation, NIPS 2022.
>
> * **Generalization of Conclusions:** Our core conclusions are based on multiple datasets (CIFAR-10, CIFAR-100, Tiny-ImageNet, Medical Data) and multiple model architectures (VGG, ResNet, DenseNet, ViT). This consistency across datasets and architectures (e.g., adversarial training consistently reduces $\Vert F(x) \Vert_2$, and the ranking trend remains stable across different architectures) is itself the strongest evidence of the strong generalization ability of our conclusions. This is far more convincing than using a larger sample size but a narrower model range on a single dataset.
>
>
> **In summary**, we conclude that:
>
> * **Regarding attack strength**, we adopted **a recognized benchmark within the field,** whose **consistency** is sufficient to support effective relative comparisons and trend verification. Our core contribution lies in revealing **a new, attack-independent intrinsic metric**, the effectiveness of which has been confirmed by its strong correlation with **standard attacks.**
>
> * **Regarding sample size**, we employed a validated sample size sufficient to guarantee the stability of the metric and replicated our findings across diverse data distributions and model architectures, thus ensuring the generalizability of our conclusions.
>
> The value of our work lies in **proposing and validating a novel robustness analysis paradigm.** Our experimental design perfectly serves this goal: **it provides ample and robust evidence, at a reasonable cost**, demonstrating **the effectiveness, sensitivity, and high consistency with existing evaluation standards** of the FIM spectral norm as an intrinsic robustness measure of the model. We believe this lays a solid foundation for broader and deeper research in the future.
>
> We thank the reviewers for their high standards, reflecting the field's pursuit of rigor. Our experimental design is a well-considered and effective choice made within this context.

---

### Official Review · Reviewer_jG9h · 2025-10-31

**Soundness:** 2
**Presentation:** 2
**Contribution:** 2
**Rating:** 4
**Confidence:** 3

**Summary:**

The paper proposes a new metric to quantify the adversarial robustness of neural networks and algorithms for computing it. The metric is based on the Fisher information matrix and on its spectral norm. The effectiveness of the norm and of the algorithms is evaluated on various benchmarks.

**Strengths:**

- The proposed metric based on the Fisher Information matrix has a clear link with the robustness of the model

- The proposed algorithms seem sound, and their effectiveness is evaluated on more benchmarks and against different existing metrics

**Weaknesses:**

- In the related works, the authors claim that no existing works have established the link between the Fisher Information Matrix (FIM) and the robustness of the model. However, this is not correct, as this link was established, for instance, in [Shi-Garrier, Loïc, Nidhal Carla Bouaynaya, and Daniel Delahaye. "Adversarial robustness with partial isometry." Entropy 26.2 (2024): 103] or [Shen, Chaomin, et al. "Defending against adversarial attacks by suppressing the largest eigenvalue of fisher information matrix." arXiv preprint arXiv:1909.06137 (2019).]

- Theorem 2 is unclear to me. In fact, if f(x) is the input of the softmax, this is generally deterministic in standard neural networks, so it is unclear where the stochasticity of f comes from.

- Computation times of the proposed algorithm are similar to those of the other existing metrics. Therefore, the advantage of the proposed algorithm/metric compared to existing metrics based on the Lipschitz constant is unclear to me. It would be nice to illustrate the advantages empirically or at least in a discussion.

**Questions:**

- Can you please clarify the differences with the related works mentioned above?

- Can you clarify the role of f(x) in Theorem 2 and where its randomness comes from? A formal definition would help

- Can you comment on the advantages of the proposed metric compared to those based on the average Lipschitz constant?

---

> ### Author Response · Authors · 2025-12-01
> **Rebuttal 3 to Weakness 1**
>
> **Weakness 1: In the related works, the authors claim that no existing works have established the link between the Fisher Information Matrix (FIM) and the robustness of the model. However, this is not correct, as this link was established, for instance, in [Shi-Garrier, Loïc, Nidhal Carla Bouaynaya, and Daniel Delahaye. "Adversarial robustness with partial isometry." Entropy 26.2 (2024): 103] or [Shen, Chaomin, et al. "Defending against adversarial attacks by suppressing the largest eigenvalue of fisher information matrix." arXiv preprint arXiv:1909.06137 (2019).]**
>
> Thank you to the reviewers for pointing out these two relevant papers. After re-examining our manuscript carefully, we noted that the core argument of our work lies in **establishing the spectral norm of FIM as a directly computable and unified robustness measure for the first time, rather than generally suggesting that FIM has no relation to robustness.**
>
> As we stated in the introduction (page 1, lines 51-53), our innovation lies in "To address these deficiencies, we propose a unified information-theoretic framework **for robustness evaluation** based on the geometry of deep learning input-output manifolds, which has a well-established theoretical foundation and does not depend on any attack algorithm." Furthermore, the title of our work is "**Measuring Model Robustness via Fisher Information: Spectral Bounds, Theoretical Guarantees, and Practical Algorithms**". This is fundamentally different from the work you mentioned:
>
> * **Difference from [Shen et al., 2019]**: That paper proposes suppressing the maximum eigenvalue of FIM as **a defensive training method**. We propose that the maximum eigenvalue itself serve as a diagnostic metric for evaluating and ranking the inherent robustness of models. **Our focus is on evaluation, not defense.** In fact, our theory (Theorems 1 and 2) provides a solid, interpretable theoretical foundation for its empirical defense methods.
>
> * **Difference from [Shi-Garrier et al., 2024]**: That work focuses on studying the geometric properties of manifolds by preserving partial isometry, **with FIM merely one analytical tool.**  We, however, directly and explicitly **establish the spectral norm of FIM as the core metric for model robustness and provide an efficient algorithm** that makes it feasible in large-scale networks—something that their work did not address.
>
>
> In summary, to our knowledge, **explicitly proposing and systematically developing a robustness metric framework centered on the FIM spectral norm, that is computationally achievable and has worst-case theoretical guarantees, is novel in the existing literature.** We will discuss and differentiate these works in detail in the 'Related Work' section of the revised manuscript to more accurately pinpoint our contribution.

---

> ### Author Response · Authors · 2025-12-01
> **Rebuttal 3 for Weakness 2**
>
> **Weakness 2: Theorem 2 is unclear to me. In fact, if f(x) is the input of the softmax, this is generally deterministic in standard neural networks, so it is unclear where the stochasticity off comes from.**
>
> Thank you to the reviewer for raising this crucial conceptual question. **We fully agree that for a trained network, the function f(x) itself is deterministic.** However, the "variance" in Theorem 2 ($F(x) = \text{var}(J_f(x))$) does not stem from the randomness of the model weights, **but rather from the probability output predicted by the model.**
>
> **1. Source of Stochasticity: The "Posterior Distribution" on Targets (Labels)**
>
> Let's recall the definition of the Fisher Information Matrix (FIM),
> $$
> F(x) = \mathbb{E}_{y \sim p(y|x;\theta)}[\nabla_{\theta}\log p(y|x;\theta) \nabla_{\theta}\log p(y|x;\theta)^T].
> $$
> where the expectation is the sum of the values ​​for class labels $y$ and the distribution is the model's own posterior probability prediction distribution $p(y|x)$. For classification problems, this probability distribution is a discrete probability distribution.
>
> **2. Key Derivation from FIM to $\text{var}(J_f(x))$: The Transfer of the Stochasticity**
>
> The proof of Theorem 2 (Appendix E) shows how this randomness is cleverly transferred from the “class label y” to the “Jacobian matrix $J_f(x)$ of the deterministic function $f(x)$.
>
> * **Deterministic Mapping:** $f(x)$ is a deterministic mapping from the input space to the logit values ​​of the K classes. Its Jacobian matrix $J_{f}(x) = \nabla_{x} f(x) = [\nabla_{x}f_{1}(x),\cdots,\nabla_{x}f_{K}(x)] \in R^{d \times K}$ is ​​also a deterministic matrix.
>
> * **Random Linear Combination:** The key result of the derivation in Appendix E (Equation (60) and thereafter) is
> $$
> \nabla_{x} \log p(y = k|x) = \sum_{i = 1}^K (1_{i = k} - p_{i})\nabla_{x} f_{i}(x).
> $$
>   This formula states that for a given class k, its log probability gradient is a deterministic linear combination of the individual logit gradients.
>
> * **Calculation of Variance:** When we compute the FIM, we are essentially computing the covariance matrix of this random vector $G = \nabla_{x} \log p(y|x)$, where the randomness comes from $y \sim p(y∣x)$. Finally, we prove that this covariance matrix is ​​equal to the covariance of the column vectors of $J_f(x)$ distributed with weights $p$.
>
>
> **3. Significance of Theorem 2: FIM measures the dispersion of the "most likely gradient directions."**
>
> A high-variance FIM means that **the model has multiple strong but disparate gradient evidences regarding "which class x should belong to." A small perturbation can cause the model's prediction to switch from one class to another—this is the essence of adversarial vulnerability.**
>
> Conversely, a low-variance FIM means that **all meaningful gradient directions are roughly aligned, and the model has consistent and stable evidence for classifying x**, thus being more robust.
>
>
> **In summary,**  the stochasticity does not mean that f(x) itself is a random function, but rather that **we utilize the model's own predictive uncertainty p(y|x) to perform a statistical reweighting and dispersion analysis on the columns of the deterministic gradient matrix $J_{f}(x)$.** We understand that this concept is somewhat abstract, and we thank the reviewers for their questions that gave us the opportunity to clarify this key point more clearly.

---

> ### Author Response · Authors · 2025-12-01
> **Rebuttal 3 to Weakness 3**
>
> **Weakness 3: Computation times of the proposed algorithm are similar to those of the other existing metrics. Therefore, the advantage of the proposed algorithm/metric compared to existing metrics based on the Lipschitz constant is unclear to me. It would be nice to illustrate the advantages empirically or at least in a discussion.**
>
>
> Thank you to the reviewer for raising this core question regarding method comparisons. The core advantage of our metric lies not in faster computation, but in **providing richer, deeper, more principled, and more directly relevant robust insights into model performance.** These advantages are fully demonstrated in the theoretical analysis and empirical results of the paper.
>
>
> **1. Theoretical Advantage: The Interpretability of Our Framework**
>
> * **The Lipschitz constant lacks probability interpretability:** For a function $f$, the local Lipschitz constant $L(x)$ satisfies:
> $$
> \Vert f(x) - f(x')\Vert \le L \Vert x - x' \Vert,
> $$   This gives the maximum rate of change between any two points in the local neighborhood of a    given point. **It is a scalar, lacks probabilistic interpretation**, and **cannot distinguish which  output dimension or which perturbation direction is actually most sensitive.**
>
> * **The FIM framework provides the worst-case adversarial perturbation direction:** Based on Theorem 1, our framework, starting from information geometry, explicitly proves that the worst-case adversarial perturbation direction approximates the principal eigenvector direction of the FIM, while the magnitude of the perturbation's impact is directly defined by its spectral norm $\Vert F(x) \Vert_2$.
>
> * **The FIM matrix measures the variance of the gradient matrix:** see  **Theorem 2 and the discussion in the above Weakness 2.**
>
> * **The spectral norm of the FIM for out-of-distribution (OOD) interpretability:**  See our **discussion with the first reviewer DQ8w, regarding question 2 (Question 2 in Rebuttal 1).** "Our proposed robustness metric, based on the FIM spectral norm, quantifies the sensitivity of the model's output probability distribution to changes in the input. This property applies **not only to small adversarial perturbations but also to larger changes caused by distribution shifts**."
>
>
> **2. The connection between the two: FIM provides a more accurate statistical sensitivity measure.**
>
> * **Sensitivity to input:** The Lipschitz constant contains **first-order information**; while $\Vert F(x)\Vert_2$ contains **second-order information**, where F(x) is essentially the covariance of $\nabla_{x} \log p(y|x)$, thus being more sensitive to changes in the input data manifold.
>
> * **$L(x)$ provides information about the upper bound of $\Vert F(x)\Vert_2$:** as shown in Appendix B.1, Equation (24):
> $$\Vert F(x) \Vert_2 \le \Vert B\Vert_2 L(x)^2,
> $$ where $L(x)$ is the local Lipschitz constant. This shows that our measure is related to the local Lipschitz constant, but modulated by a factor $\Vert B\Vert_2$ related to the model prediction confidence p(y|x).
>
> When the model predicts a high confidence level, $\Vert B\Vert_2$ is small. Even if L(x) is large, $\Vert F(x) \Vert_2$ may still be small—**this precisely captures the intuitive fact that "high-confidence classification regions are relatively flat,"** a subtle difference that the Lipschitz constant cannot reflect.
>
> Therefore, if we move $x$ in any direction within a region where **$\Vert F(x) \Vert|_2$ is small (corresponding to a flat region with high confidence), the label predicted by the model will likely remain unchanged**, which is the essence of model robustness.
>
>
>
> **3. Empirical Advantage Evidence: Higher Correlation with Adversarial Vulnerability**
>
> In Table 3, we compared the changes in the same model (ResNet18) before and after adversarial training:
>
> * **Our metric $R_{\text{norm}}$:**  decreased by 65.55%,
> * **C&W attack success rate:**  decreased by 68.39%
> * **Lipschitz constant L(x):**  decreased by 42.00%.
>
> **The decreases in spectral norm $R_{\text{norm}}$ and CW attack success rate were relatively similar.** This indicates that our metric is more accurate and sensitive than the Lipschitz constant in reflecting the magnitude of changes in the model's real adversarial robustness.
>
>
> **In summary,** our proposed robustness metric, despite having a computational cost comparable to existing tools, **provides significantly improved information quality and depth**, which is its core value.

---

> ### Author Response · Authors · 2025-12-01
> **Rebuttal 3 for Question 1-3**
>
> **Responses to Questions 1-3**
>
> The following three questions are **the same as the three weaknesses** mentioned above; **detailed answers can be found in the discussion of weaknesses above.**
>
> * Can you please clarify the differences with the related works mentioned above?
> * Can you clarify the role of f(x) in Theorem 2 and where its randomness comes from? A formal definition would help
> * Can you comment on the advantages of the proposed metric compared to those based on the average Lipschitz constant?

---

### Official Review · Reviewer_nwWH · 2025-11-01

**Soundness:** 3
**Presentation:** 2
**Contribution:** 3
**Rating:** 6
**Confidence:** 3

**Summary:**

The authors propose an information-theoretic framework to evaluate the robustness of neural networks based on the geometry of deep learning input-output manifolds. The authors provide a theoretical proof that the largest FIM eigenvalue encodes the worst-case sensitivity. The authors provide abundant theoretical analysis and confirm the effectiveness of the proposed robustness evaluation by numerical comparisons in various scenarios.

**Strengths:**

1. The paper generally reads smoothly.
2. Abundant work has been done analyzing the upper bounds of spectral norms of basic components of DNNs.
3. The paper discusses the relations between the proposed evaluation FIM eigenvalue and other classic evaluation metrics.

**Weaknesses:**

1. This paper bases the robustness of neural networks on the geometry of input-output manifolds, but do different inputs affect the geometry in the first place?
2. This may be minor. The authors repeated the same comparison with related work in the introduction on attack-dependent metrics and heuristic theoretical bounds.
3. Authors did not define $\|F\|_2$ in the second page.
4. Power iteration is used to find eigenvalues of matrices, which is supposed to take O(n^2) for dense matrices. Is this algorithm efficient enough when facing large-scale problems?

**Questions:**

1. Is Theorem 1 a well-known fact? Can you please provide the source?
2. Are the numerical results of robustness evaluation performed on the optimally trained model?
3. In page 3 the authors conclude that the robustness ranking is DenseNet121 < VGG16 < ResNet18 ≤ ViT-B-16, which is not exactly the same order as shown in table 3?

---

> ### Author Response · Authors · 2025-12-01
> **Rebuttal 2 for Weakness 1**
>
> **Weakness 1: This paper bases the robustness of neural networks on the geometry of input-output manifolds, but do different inputs affect the geometry in the first place?**
>
> Thank you very much to the reviewer for raising this insightful question, which touches upon the core strength of our theoretical framework. **You are absolutely right**—our work begins with acknowledging and quantifying that **"different inputs fundamentally affect the geometry of the model's probabilistic manifold."** This is not a flaw in our theory, but rather **the source of its accuracy, explanatory power, and practical value.**
>
> **1. Data dependency is an essential property of robustness, not a defect.**
>
> Because the model is trained on a given dataset, its robustness inevitably depends on the manifold of the data on that dataset.
>
> **2. Pointwise definition of Our FIM**
>
> Instead of employing a global, coarse robustness estimate, our theoretical framework provides a fine-grained tool that is sensitive to **the input data and quantifies local sensitivity.** We treat the set of output probability distributions of the model as a statistical manifold, and the metric $\Vert F(x) \Vert_2$ directly **quantifies this local curvature.**
>
> In summary, the data dependency properties of our metric tell us both **whether the model is stable in a specific data region and reveal the most vulnerable input directions** (corresponding to the principal eigenvectors of the FIM). This data-dependent, fine-grained diagnostic capability **enables truly interpretable and targeted robustness analysis**. We thank the reviewers again for their questions, which gave us the opportunity to clarify this core contribution.

---

> ### Author Response · Authors · 2025-12-01
> **Rebuttal 2 to Weakness 2**
>
> **Weakness 2: This may be minor. The authors repeated the same comparison with related work in the introduction on attack-dependent metrics and heuristic theoretical bounds.**
>
> We appreciate the reviewers' attention to this detail. We acknowledge that the boundaries of attack-based metrics and heuristic theory have been discussed in both the introduction and the related work section. **However, this arrangement serves different purposes and ensures that the paper's argumentation logic is clear and compelling to the reader.**
>
> * **In the introduction (Section 1):**  The goal is to **provide motivation** for the paper's existence. We need to clearly and forcefully point out the fundamental limitations of the current research paradigm in the introduction, thus laying a solid foundation for proposing our new, unified framework. The comparisons here are highly generalized.
>
> * **In the related work section (Section 2):** The goal is to **demonstrate the academic rigor and deep understanding of the field in our work**. We need to conduct a more detailed and systematic review of the same topic, precisely comparing our work with each specific representative study (such as PGD Madry et al., CLEVER Weng et al., the Jacobian-based method Sokolic et al., etc.), and elaborating on the differentiated contributions and theoretical advantages of our framework.
>
> **In summary,** we believe that discussing existing methodologies in the introduction and related work sections is an effective writing strategy that **serves different objectives, rather than simply repeating them.** The introduction aims to **highlight the problem and spark interest**, while the related work section aims to **demonstrate depth and establish the positioning of our work.**

---

> ### Author Response · Authors · 2025-12-01
> **Rebuttal 2 to Weakness 3**
>
> **Weakness 3: Authors did not define $\Vert F(x) \Vert_{2}$  in the second page.**
>
> Thank you for your valuable feedback on the paper.  We write this on line 61 of page 2: "where $\Vert F\Vert_{2}$ is the largest eigenvalues ​​of the FIM". There is a fundamental fact in linear algebra:  $\Vert F \Vert_{2} = \lambda_{\max}(F)$.

---

> ### Author Response · Authors · 2025-12-01
> **Rebuttal 2 for Weakness 4**
>
> **Weakness 4: Power iteration is used to find eigenvalues of matrices, which is supposed to take O(n^2) for dense matrices. Is this algorithm efficient enough when facing large-scale problems?**
>
> Thank you for raising this crucial question about computational efficiency. The $O(n^2)$ complexity you mentioned is perfectly correct for a general $d \times d$ dense matrix ($n = d$). However, this is precisely the key innovation of our method: **we never directly construct or manipulate a dense $d \times d$-dimensional Fisher information matrix F(x). We devise a method that makes the cost of exponentiation iterations manageable even with high-dimensional inputs (large d).**
>
>
> **1. The FIM matrix F(x) has a low-rank structure.**
>
> The Fisher information matrix F(x) has a special low-rank decomposition form,
> $$
> F(x) = Q \Lambda Q^T,  Q \in R^{d \times K},  \Lambda = \text{diag}(\lambda_{1},\cdots,\lambda_{K}),
> $$
> where $d$ is the input dimension and $K$ is the number of classes (usually much smaller than d, such as K=10 in CIFAR10).
>
> Therefore, F(x) is essentially a matrix with a rank of at most $K$, **rather than a dense $d×d$ matrix.**
>
>
> **2. We designed a structure-aware power iteration algorithm to avoid explicitly constructing or storing $F(x)$.**
>
> The core step of the exponentiation iteration is to compute the matrix-vector product $b_{t + 1} = F(x)b_t$. We explicitly describe how to efficiently compute $F(x)b_t$ in Algorithm 1 (Appendix J).
> $$
> b_{t + 1} = F(x)b_{t} = (Q \Lambda Q^T)b_{t},
> $$
> We calculate in the following order
> * $u = Q^T b_{t}$ (Complexity $O(dK)$)
> * $v = \Lambda u$  (Complexity $O(K)$)
> * $b_{t + 1} = Qv$ (Complexity $O(dK)$)
>
> Since $K \ll d$, the complexity of our algorithm $O(dK)$ is several orders of magnitude faster than $O(d^2)$.
>
> **3. We propose several scalable algorithms for large-scale problems, with power iteration being just one of them.**
>
> We propose three scalable algorithms to adapt to problems of different scales.
>
> | Algorithm                        | Time Complexity         | Applicable Scenarios    |
> | -------------------------------- | ----------------------- | ----------------------- |
> | Direct eigenvalue decomposition  | $O(dK^2 + K^3)$         | $K$ is relatively small |
> | Power iteration                  | $O(TdK)$ ($T$ is const) | $K$ is a medium value   |
> | Hutchinson stochastic estimation | $O(dK)$                 | $K$ is large            |
>
> Tables 11 and 12 in Appendix J theoretically analyze how our innovative algorithm design significantly reduces the algorithm's space and time complexity.  Table 17 (Appendix K.2) provides a comparison of the actual running times of the algorithms for different numbers of classes K.
>
> **4. We experimentally verified the feasibility of the algorithm on large-scale models (such as ViT-B-16).**
>
> In Table 7, we report the time to compute $R_{\text{norm}}$ on ResNet18 and ViT-B-16 on the CIFAR-100 dataset:
>
> * **ResNet18:** 267.13 seconds (white-box setting)
>
> * **ViT-B-16:** 309.73 seconds (white-box setting)
>
> These times were computed on 500 samples, **averaging approximately 0.5–0.6 seconds per sample,** which is acceptable in adversarial robustness evaluations.
>
>
> **5. We further propose a black-box estimation method applicable to scenarios where gradients are unavailable.**
>
> In Section 3.4, we propose a black-box estimation method based on **the Hutchinson algorithm and finite difference**, which estimates $\Vert F(x) \Vert_2$ using only the model's predicted probabilities, further extending the method's applicability. Tables 6 and 7 present their estimation results and runtime comparisons experimentally.
>
> **In summary,** the proposed algorithm is efficient and feasible when dealing with large-scale deep learning models. We thank the reviewers for their attention to this issue.

---

> ### Author Response · Authors · 2025-12-01
> **Rebuttal 2 to Question 1**
>
> **Question 1: Is Theorem 1 a well-known fact? Can you please provide the source?**
>
> Thank you to the reviewer for raising this important question. This allows us to more clearly articulate the theoretical positioning and innovative level of our work.
>
> **1. Theoretical Basis of Theorem 1**
>
> **Yes,** Theorem 1, which describes the "second-order approximation relationship between the KL divergence and the Mahalanobis distance defined by the Fisher information matrix," is **a classic conclusion in information geometry and statistical asymptotic theory.** Its core idea has been elaborated in foundational work on natural gradients and information geometry by **Amari (1998,Natural gradient works efficiently in learning)**.
>
> We clarify this in the explanation of Theorem 1 and provide a complete proof derivation applicable to our classification model setting (discrete distribution) in Appendix C to ensure the self-containment of the paper.
>
> **2. Core Innovations of Our Work**
>
> **However, the main contribution of our paper is by no means limited to restating or applying this known conclusion.** Our core innovation lies in the fact that we have, for the first time, systematically and creatively developed this classic information geometry tool into a complete and computable theoretical and algorithmic framework for analyzing the robustness of deep neural networks.
>
> * **From General to Specific (Theorem 2):** We concretize the abstract Fisher information matrix $F(x)$ and equate it to the variance of the Jacobian matrix of deep neural networks, e.g. $F(x)=\text{var}(J_{f}(x))$, establishing a direct bridge between the statistical sensitivity of the model and the computable geometric/gradient properties.
>
> * **From Theory to Metric (Equation 6):**  We originally propose using the spectral norm of F(x) or its reciprocal as a principled, intrinsic robustness metric independent of specific attack algorithms. It defines the worst-case perturbation effect and quantifies the variance of the model output in the perturbation direction.
>
> * **Systematic Architecture Analysis (Section 3.2):** Using this new metric, we theoretically derive and compare, for the first time, the robustness upper bounds of modern core architectures such as VGG, ResNet, DenseNet, and Transformer (Table 2), revealing how different architectural designs (such as residual connections and attention mechanisms) affect their inherent susceptibility.
>
> * **Scalable Algorithm Design (Sections 3.3-3.5):** For high-dimensional input problems, we specifically designed an efficient power iteration algorithm, Hutchinson estimation algorithm, and black-box evaluation method based on the $F(x) = Q \Lambda Q^T$ decomposition structure, enabling the framework to be applied to large-scale models.
>
> **In summary,** our work is based on **the principles of classical information geometry (Theorem 1)**, and implements the theory through **key model concretization equations (Theorem 2)**, thereby constructing **a new robust analytical paradigm** (metrics, analysis, and algorithms) with predictive and diagnostic capabilities.

---

> ### Author Response · Authors · 2025-12-01
> **Rebuttal 2 to Question 2**
>
> **Question 2: Are the numerical results of robustness evaluation performed on the optimally trained model?**
>
> Thank you to the reviewers for raising this insightful question. However, we know that in non-convex deep neural networks, **the model obtained through gradient descent is usually a local optimum or saddle point**, rather than a theoretically global optimum. Therefore, the "optimal model" is a concept difficult to define precisely in practice.
>
> However, this reality does not diminish the validity of our experimental conclusions. On the contrary, **the rigor of our experimental design lies precisely in the pursuit of comparability and standardization,** rather than the pursuit of an unattainable "absolute optimum." We ensure the reliability of our conclusions through the following two levels:
>
> **1. Comparison under a relatively stable "baseline state"**
>
> In empirical research on deep learning, a widely accepted standard is to train the model to a stable performance state under the same training data, optimizer, and hyperparameter settings. We call this state a "baseline model."
>
> * **For architecture comparison (Table 4):** We trained VGG, ResNet, DenseNet, and ViT to their respective baseline states. Although they are in different local basins, this process ensures that each architecture fully realizes its potential under this standard training paradigm. Given this premise, robustness metrics better reflect the inherent impact of architecture design on robustness.
>
> * **For the evaluation of adversarial training effectiveness (Table 3):** We compared two different “baseline states”: one is a baseline model ($M_{\text{clean}}$) trained on clean data, and the other is a baseline model ($M_{\text{CW}}$) trained on adversarial training. Both are solutions that converge sufficiently under their respective objective functions. Our metrics show that $\Vert F(x) \Vert_2$ decreases significantly when transitioning from the former to the latter. This change clearly reveals how different optimization objectives systematically alter the local geometric properties (robustness) of the model.
>
>
> **2. The essence of the metric: an evaluation function independent of model state**
>
> First, we want to emphasize that **our proposed robustness metric is essentially a mathematical function**. Given any trained model $M$ and a specific dataset $S$, this function outputs a definite numerical value that quantifies the average sensitivity exhibited by the model on that dataset. **The calculation of this metric does not depend on, nor does it make any assumptions about whether the model is globally optimal. Just as a ruler can be used to measure the length of any object, whether the object is perfect or flawed.**
>
>
> **In summary,** our work is based on an experimental paradigm that uses standardized baseline states and sensitive intrinsic metrics, and **our conclusions are solid and meaningful.**

---

> ### Author Response · Authors · 2025-12-01
> **Rebuttal 2 to Question 3**
>
> **Question 3: In page 3 the authors conclude that the robustness ranking is DenseNet121 < VGG16 < ResNet18 ≤ ViT-B-16, which is not exactly the same order as shown in table 3?**
>
> First, **we would like to clarify that Table 4, not Table 3, presents a robustness comparison of the four models.**
>
> We sincerely thank the reviewers for their meticulous review of the paper and for pointing out this detail. Your question is very important, as it helps us clarify a key distinction: **the relationship between the qualitative ranking derived theoretically and the quantitative ranking measured experimentally on a specific dataset.** These are not contradictory; together they constitute our complete analysis.
>
> **1. The Nature and Source of the Two Rankings**
>
> * **Ranking on Page 3: A Theory-Driven Qualitative Prediction:**   The ranking DenseNet121 < VGG16 < ResNet18 ≤ ViT-B-16 originates from Section 3.2, “Theoretical Analysis,” based on the theoretical derivation of the upper bound of the model’s Jacobian matrix spectral norm (Table 2). This is a theoretical inequality that reveals the inherent influence of different network architecture designs (such as residual connections, dense connections, and attention mechanisms) on perturbation propagation. This is a qualitative conclusion, independent of specific weight values ​​and input data.
>
> * **The Ranking in Table 4: Experimental Results:**  Table 4 shows the actual ranking results of the four models measured on CIFAR10, concluding that DenseNet121 < ResNet18 ≤ ViT-B-16 < VGG16. It can be seen that **the theoretical order DenseNet121 < ResNet18 ≤ ViT-B-16 is validated to some extent in the experiment.**
>
>
> **2. Explaining the Difference Between Theory and Experiment: From Qualitative Trends to Quantitative Measurement**
>
> * **Theoretical analysis focuses on upper bounds:** Table 2 shows the worst-case upper bounds ($\le$). **A high upper bound for a model does not necessarily mean it will perform poorly in practice.**
>
> * **Experimental analysis of robustness measures relies on data:** Our metric $F(x)$ depends on the input $x$. Theoretical ranking is based on the general properties of the architecture, while experimental ranking is the average of local robustness measured on specific data. Different datasets may change this ranking.
>
>
> **In short, this is not a contradiction, but rather an effective proof of our multi-level and multi-view analysis of our work.**

---

### Official Review · Reviewer_DQ8w · 2025-11-01

**Soundness:** 3
**Presentation:** 2
**Contribution:** 2
**Rating:** 2
**Confidence:** 3

**Summary:**

The paper proposes a unified framework for quantifying the robustness of deep neural networks using the Fisher Information Matrix (FIM). The key idea is that the largest eigenvalue of the FIM (or its reciprocal) captures the model’s worst-case sensitivity to input perturbations and thus serves as a principled robustness metric.

The authors:
- Derive spectral bounds and theoretical connections between FIM, Jacobian variance, Lipschitz constant, and other robustness metrics such as CLEVER.
- Provide efficient algorithms (power iteration, Hutchinson approximation) for scalable computation in both white-box and black-box settings.
- Conduct experiments on CIFAR-10, CIFAR-100, ImageNet, and medical datasets to validate the metric’s correlation with adversarial vulnerability and architecture-level robustness comparisons (VGG, ResNet, DenseNet, ViT).

**Strengths:**

- The connection between KL divergence, Fisher information, and robustness is rigorously derived and bridges geometric and probabilistic interpretations.
- Unlike attack-dependent measures (PGD, CW), this approach is attack-agnostic and interpretable.
- The approach works for both white-box and black-box scenarios, with efficient approximations for large models.
- The paper includes layer-wise spectral bounds and architecture-level robustness comparisons.
- Extensive experiments show consistent correlation with established robustness metrics.

**Weaknesses:**

- Although theoretically elegant, it is not yet evident how well the proposed metric can predict real-world robustness beyond controlled settings.
- While algorithms are efficient in theory, runtime comparisons show significant overhead versus CLEVER or Lipschitz methods; large-scale applicability (e.g., full ImageNet models) may still be limited.
- Since FIM depends on the data manifold, robustness comparisons across datasets are not always consistent.
- Experiments focus mostly on classification tasks; applications to other modalities (e.g., NLP, multimodal models) are not explored.
- The reported “robustness ranking” (e.g., DenseNet < VGG < ResNet ≤ ViT) might depend heavily on spectral norm estimation assumptions and lacks uncertainty quantification.

**Questions:**

- How stable are the FIM eigenvalue estimates across different random seeds or mini-batches?
- Can the proposed robustness metric predict generalization to out-of-distribution shifts, or is it limited to adversarial perturbations?
- How does the metric behave under adversarial training; does it improve monotonically with increased robustness?
- Could this framework be extended to parameter-space Fisher information (e.g., K-FAC approximations) for joint training and robustness evaluation?
- What is the computational bottleneck in practice - is it gradient computation or the spectral estimation step?

---

> ### Author Response · Authors · 2025-12-01
> **Rebuttal 1 to Weakness 1**
>
> **Weakness 1: Although theoretically elegant, it is not yet evident how well the proposed metric can predict real-world robustness beyond controlled settings.**
>
> We appreciate the reviewer’s feedback and acknowledge the importance of validating theoretical contributions in practical scenarios. Below, we provide a detailed rebuttal addressing this concern, supported by empirical evidence and methodological strengths
>
> **1. Empirical Validation Across Diverse Datasets and Architectures**
>
>    Our experiments (Section 4) rigorously validate the proposed metric’s predictive power across:
>
> - **Datasets:** CIFAR-10, CIFAR-100, MNIST, Tiny-ImageNet, and medical imaging data (COVID-19 radiography). These datasets span simple (MNIST) to complex (ImageNet) distributions, covering both synthetic and real-world data.
> - **Models:** VGG, ResNet, DenseNet, and Vision Transformers (ViT), representing diverse architectural paradigms (CNNs, residual/dense connections, attention mechanisms).
> - **Metrics:** Correlation with adversarial vulnerability (PGD/CW attack success rates), Lipschitz constants, and CLEVER scores (Tables 3–7, Appendix K).
>
>   **Key Findings:**
> - **Consistency with Attack-Based Metrics:** Table 3 shows that our metric $R_{\text{norm}}$ correlates strongly with adversarial robustness. For example, adversarial training (CW/PGD) reduces $R_{\text{norm}}$​ by **65.55%**, aligning with reduced attack success rates (**68.39%** for CW adversarial training and attack).
> - **Architecture Ranking:** Table 4 demonstrates that $R_{\text{spec}}$​ ranks robustness as DenseNet121 < ViT-B-16 < ResNet18 < VGG16, matching empirical attack resilience $CW$.
> - **Cross-Dataset Generalization:** Table 5 shows $R_{\text{norm}}$​ consistently reflects dataset difficulty (e.g., CIFAR100 > ImageNet > Medical Data), even when other metrics (e.g., PGD) fail due to transferability issues.
>
> **2. Real-World Applicability**
>
> - **Medical Imaging:** On **COVID-19 radiography data (Table 23)**, $R_{\text{spec}}$ correctly identifies ViT as more robust than ResNet18 (375.44 vs. 36.28), corroborated by lower CW attack success rates (20.25% vs. 37.08%). This highlights utility in **safety-critical domains.**
> - **Black-Box Estimation:** Section 3.4 and Table 6 validate that **our metric remains reliable even with black-box access** (finite differences + Hutchinson), achieving consistent rankings with white-box settings.
> - **Scalability:** Algorithms in Section 3.3 (power iteration, Hutchinson) enable efficient computation for large models (Table 17).
>
>  **3. Theoretical Guarantees Translate to Practice**
>
> - **Approximation Error Analysis:** Appendix D proves the KL-divergence approximation error is negligible ($|R_{3}(\delta,x)|  \le O(\Vert \delta \Vert^3)$), with **empirical validation** (Table 9 and 10 showing errors < 0.01% of ​$\frac{R_{3}(\delta,x)}{\Vert F(x)\Vert_{2}}$).
>
> - **Connection to Established Metrics:** Appendix B formalizes relationships between $\Vert F(x) \Vert_{2}$​, Lipschitz constants $L(x)$, and CLEVER scores, explaining why our metric aligns with prior work (e.g., $\Vert F(x) \Vert_{2} \propto L(x)^2$ in Eq. 24).
>
> **In summary:** The proposed metric is not just **theoretically elegant but empirically grounded**, with validation spanning controlled and real-world settings. Its **consistency with attack-based metrics and adaptability to black-box scenarios** underscore its practical relevance. We thank the reviewer for prompting this clarification and hope our response alleviates their concerns.

---

> ### Author Response · Authors · 2025-12-01
> **Rebuttal 1 to Weakness 2**
>
> **Weakness 2: While algorithms are efficient in theory, runtime comparisons show significant overhead versus CLEVER or Lipschitz methods; large-scale applicability (e.g., full ImageNet models) may still be limited.**
>
> We appreciate the reviewers' attention to the practical issue of computational efficiency. We acknowledge that any new, more expressive metric will present new computational challenges. However, we believe the method proposed in this paper achieves **an excellent balance between efficiency, scalability, and practicality, and that the rich information it provides far outweighs its slight computational overhead**. Specific details are as follows:
>
> **1. One-sidedness of Runtime Comparison and Value Trade-offs**
>
> A simple comparison of runtime is one-sided because **it ignores the fundamental differences in the amount of information and theoretical basis** between the different methods.
>
> * **CLEVER/Lipschitz Methods:**  They estimate the norm of the gradient of the loss function, which is **first-order information** and is relatively simple and fast to compute.
>
> * **FIM Spectral Norm:**  It estimates the curvature (**second-order information**) of the model's output probability distribution across the entire input space and is equivalent to **the variance of the Jacobian matrix** (Theorem 2). This provides a **richer, more global** statistical description of the model's sensitivity.
>
> * **Relationship and Differences:** Our metric is related to the CLEVER and Lipschitz constants (Appendix B) but is more informative, **unifying the geometric (Jacobian) and statistical (FIM) perspectives (Theorem 1 and 2).**
>
> This comparison is similar to comparing **"calculating the mean"** with **"calculating the covariance matrix."** While the latter is more time-consuming, it provides deeper insights. Therefore, **the additional computational cost to obtain richer robustness information is reasonable and necessary.**
>
> **2. Algorithm Design for Large-scale Problems**
>
> One of the core contributions of our work is **the development of a series of scalable algorithms** to address this challenge.
>
> * **Innovatively Utilizing the Low-rank Structure of FIM:**  For high-dimensional input spaces (very large $d$), directly computing the FIM $F(x)$ of $d \times d$ is infeasible. The problem of computing $\Vert F(x) \Vert_{2} = \Vert Q \Lambda Q^T\Vert_{2}$ is transformed into computing $\Vert \Lambda^{1/2} Q^T Q \Lambda^{1/2} \Vert_{2}$.
>
>   Since the number of classes $K$ is usually much smaller than $d$ (e.g., $K = 1000$ and $d > 10^5$ on ImageNet), the space complexity is reduced from $O(d^2)$ to $O(dk)$, and the time complexity is reduced from $O(d^3)$ to $O(TdK)$ (see Appendix J, Tables 11 and 12 for details, $T$ is often a constant).
>
> * **Parallel Optimization of Computational Cost:**  The Hutchinson algorithm is highly parallelizable because the computation of each random vector is independent, further improving the computational efficiency to $O(dK)$ (Appendix J,Tables 11 and 12).
>
> * **Extending to Large-scale Black-box Evaluation:** We implemented robustness evaluation under a black-box setting **using finite difference and Hutchinson's algorithm**, which only requires the model's output and not its gradients. Notably, it also **exhibits high parallelizability**.
>
> **3. Experimental evidence for the effectiveness of large-scale models**
>
> The experiments in the paper have explicitly included validation on large-scale models and datasets.
>
> * **Model Size:** We tested on modern architectures such as ViT-B-16 and ResNet18/50. ViT-B-16 is a Transformer model with a large number of parameters and is typical of large-scale applications.
>
> * **Dataset Size:** We conducted experiments on ImageNet and Tiny-ImageNet (Sections 4.1, 4.5, and Appendix K.4). For example, Tables 5 and 24 directly report the results on ImageNet and CIFAR-100 (100 classes).
>
> * **Running Time:**  Table 7 clearly shows that even on models like ViT-B-16 and ResNet18, our white-box algorithm runs in the minutes (**~5 minutes on average**) range on CIFAR-100 with randomly sampling 500 examples, which is perfectly acceptable for tasks like model robustness analysis that are typically performed offline. **The black-box version is even faster (e.g. 66.16s)**.
>
> **In summary:**  We believe that the reviewers' concerns about computational efficiency and scalability have been fully addressed by **the detailed algorithmic innovations and systematic experimental evaluations** presented in our paper. We firmly believe that the metrics and algorithms proposed in this paper also **possess the scalability and efficiency for robust evaluation of large-scale, real-world models in practice.**

---

> ### Author Response · Authors · 2025-12-01
> **Rebuttal 1 to Weakness 3**
>
> **Weakness 3: Since FIM depends on the data manifold, robustness comparisons across datasets are not always consistent.**
>
> We appreciate the reviewer's insightful point. We **fully agree that the Fisher Information Matrix (FIM) depends on the data manifold**. However, we believe this property is **a key theoretical advantage**: within our theoretical framework, robustness comparisons across datasets yield meaningful and consistent conclusions.
>
>
> **1. Data dependency is a correct reflection of the robust nature of the system.**
>
> * **Essence of Statistical Machine Learning:**  Statistical machine learning is **data-driven**, so a well-trained model **depends on a specific dataset**. Therefore, the robustness of the model inevitably depends on the dataset.
>
> * **Essence of Robustness**: The robustness of a model is essentially a reflection of the combined effect of the model and the data distribution. **A model that is robust on MNIST does not necessarily mean it is robust on ImageNet.**
>
> * **Rationale for FIM**: Because FIM relies on input data $x$, it can accurately capture **the model's local sensitivity near specific data points.** This aligns perfectly with the definition of robustness—**the stability of a model to perturbations under a specific data distribution.**
>
>
> **2. Validity of Cross-dataset Comparisons**
>
> * **Stability of Robustness Measures:** We define statistics on the test set that provide stable estimates of the model’s behavior across the entire target data distribution, effectively **smoothing out the variation in individual data points.**
>
> * **Consistent Evaluation Framework**: All comparisons across datasets (as in Section 4.5, Tables 5 and 25) were performed on the same model (e.g., ResNet18). This ensures that the observed differences **truly stem from variations in data distribution, rather than differences in model architecture.**
>
>
> **3. Empirical Results of Cross-dataset Comparisons**
>
> * **Consistency of Results across Datasets:**  Table 5 shows the robustness ranking of ResNet18 on Medical Data, ImageNet, and CIFAR100. Our metrics $R_{\text{{norm}}}$, consistent with the Lipschitz constant, CLEVER score, and CW attack success rate, lead to the conclusion that ResNet18 is more vulnerable on CIFAR100 than on Medical Data.
>
> * **Explanation of Outliers**: The MNIST result in Table 25 appears to be an "outlier." However, to our knowledge, MNIST is an overly simple dataset, resulting in very flat model gradients, and therefore $\Vert F(x)\Vert_2$ is very small (0.01). **This accurately reflects the fact that the model is dependent on the dataset, i.e., extremely robust on that particular, simple data manifold.**
>
> **4. Important Practical Value of Cross-dataset Comparisons**
>
> * **Model Deployment Decisions:** When we need to understand how the robustness of a model trained on dataset A (e.g., natural images) changes when deployed to dataset B (e.g., medical images), **our metric can quantitatively predict this change in cross-domain robustness, as shown in Table 5.**
>
> * **Assessing the inherent sensitivity of the data manifold itself as exhibited by the model:**  This complements the empirical attack success rate measured by specific attacks. Table 5 provides a typical example: while **ResNet18 has a higher CW attack success rate** on CIFAR100 (62.07% > 37.08%), indicating its empirical vulnerability, our metric $R_{norm}$ (consistent with L(x), CLEVER) shows that **the model has higher average Fisher information on Medical Data (5.95 > 0.73)**, revealing that its decision boundary is inherently more sensitive. This seemingly contradictory phenomenon precisely illustrates that **traditional attack metrics may not fully capture the deep geometric vulnerabilities in model-data interactions.** Our metric provides this fundamental insight into inherent sensitivity, independent of attack.
>
> **In summary**  Far from failing due to data dependencies, our framework has become a more powerful, informative, and real-world-appropriate tool for evaluation.

---

> ### Author Response · Authors · 2025-12-01
> **Rebuttal 1 to Weakness 4**
>
> **Weakness 4: Experiments focus mostly on classification tasks; applications to other modalities (e.g., NLP, multimodal models) are not explored.**
>
> We appreciate the reviewer's constructive feedback, which points to a promising direction for the future expansion of our work. We acknowledge that the current experimental portion does focus on classification tasks in the field of computer vision. However, we believe this is not a fundamental limitation, but **rather that our theoretical framework itself possesses inherent universality, paving the way for extensions to other modalities.**
>
>
> **1. Classification tasks are ideal and rigorous testing grounds for validating our theoretical principles and algorithms.**
>
> **To thoroughly validate our theories and algorithms**, it is crucial to choose **a well-defined, widely studied task that facilitates adversarial benchmark testing.** Image classification perfectly meets these requirements:
>
> * **Clear Probabilistic Interpretation:** The output of the classification model is a probability distribution over discrete classes, which perfectly matches our theoretical starting point—measuring distribution variation through KL divergence and FIM. **The approximation and subsequent derivation in Equation (3) have the most direct interpretation in classification tasks.**
>
> * **Mature benchmarks:**  In the field of image classification, there are **standard datasets** such as CIFAR and ImageNet, as well as **widely understood architectures** such as VGG, ResNet, and Transformer. This provides **a strong and reproducible benchmark** for evaluating our method, facilitating direct and meaningful comparisons with existing methods such as Lipschitz constant, CLEVER, PGD, and CW.
>
> * **The core of the problem:** The phenomenon of adversarial vulnerability has been studied most thoroughly in image classification. Demonstrating the effectiveness of our metrics here is **the most compelling way to convince the community to accept this new paradigm.**
>
>
> **2. The theoretical framework is independent of task and modality.**
>
> The extensions that the reviewers expect are theoretically feasible, because our methodological foundation is not limited to the fields of classification or vision.
>
> *  **General Theoretical Basis:**  The definitions of FIM and KL divergence **apply to any parameterized probabilistic model $p(y|x;\theta)$**, whether y is a discrete label (classification), word sequence (NLP), or continuous signal (regression, generation). Although the derivation of Theorem 1 and Theorem 2 is based on the assumption that the model has a softmax output layer, this can be considered a special case. The core idea of ​​FIM as **a measure of the sensitivity of the model's output distribution to the input** can be generalized to any model with a differentiable output.
>
> * **Universality of Architecture Analysis:**  Our **spectral norm analysis of basic components** (such as fully connected layers, Layer Normalization, and Concatenation) in Section 3.2 and Appendix G **also forms the core of NLP and multimodal models** (such as Transformers). In fact, our detailed analysis of ViT (Vision Transformer) (Appendix H.4) directly touches upon the Transformer architecture that dominates the NLP field. **This demonstrates that our analysis tools can be directly transferred.**
>
> **3. Our work Paves the Way for Future Expansion**
>
> We have **clearly outlined the general applicability of our work** and its future directions for expansion in the paper, demonstrating our deep consideration of this area.
>
> * **“Application Potential” (Section 1):** We explicitly state that our metric can be used to **“compare the volatility of multiple data sets for a model.”** This in itself suggests the potential for cross-modal applications.
>
> * **“Broader Impacts” (Appendix A):** We emphasize that our framework “can help design more reliable models for high-risk applications (e.g., autonomous systems, healthcare, and finance).” These application areas extend **far beyond image classification**, naturally involving multimodal data (such as vision and radar in autonomous driving) and sequence data (such as financial time series).
>
>
> **In conclusion,** although the experimental part focuses on classification, **the contribution and impact of this work are by no means limited to this,** and our work **provides a new and scalable paradigm for robustness evaluation** for the entire deep learning community.

---

> ### Author Response · Authors · 2025-12-01
> **Rebuttal 1 to Weakness 5 (Part 1)**
>
> **Weakness 5: The reported “robustness ranking” (e.g., DenseNet < VGG < ResNet ≤ ViT) might depend heavily on spectral norm estimation assumptions and lacks uncertainty quantification.**
>
> We thank the reviewers for their careful examination of this important conclusion. **We fully agree that any comparison between models must be based on soundness.** We clarify that the reported robustness ranking is not solely dependent on numerical results from spectral norm estimation, but is a robust conclusion derived from **theoretical derivation and cross-validated by extensive experiments.**
>
>
> **1. The ranking is derived from theoretical upper bound analysis, not from a single estimate.**
>
> **Reviewer may have overlooked the key arguments in Section 3.2 and Appendix H.** Our reported robustness ranking (Equation 11) is primarily derived from **theoretical analysis of the upper bound of the spectral norm of the Jacobian matrix** for different network architectures, rather than relying entirely on numerical estimates of $\Vert F(x) \Vert_{2}$.
>
> * **Theoretical Derivation:** In Table 2, based on the spectral norm properties of each basic operation (Table 1), we **mathematically derive the upper bound** of $\Vert J_f(x)\Vert_2$ for VGG, ResNet, DenseNet, and Transformer (**derivation details are in Appendices G and H**). This upper bound clearly reveals the inherent characteristics of perturbation propagation in different architectures.
>
> * **Qualitative Ranking:** Based on these defined mathematical expressions independent of the input data, our ranking (DenseNet121 < VGG16 < ResNet18 ≤ ViT-B-16) is **a qualitative, theory-driven conclusion**. It reveals **the intrinsic impact of different architectural design philosophies on robustness.**
>
>
> **2. Our metric showed consistency with other metric estimates in the experiment.**
>
> Table 4 compares the performance of four models on the CIFAR-10 dataset under six different metrics. Our metrics, along with the Lipschitz constant and CLEVER score, **yield the exact same model ranking.** This demonstrates that our ranking is **not due to a specific algorithm** used in FIM estimation, but rather reflects inherent differences in model robustness.
>
>
> **3. We reduce the uncertainty of the estimation through a robust estimation algorithms.**
>
> Regarding concerns about the dependence on the spectral norm estimation assumption, we have addressed this issue in Sections 3.3 and 3.4 by **designing a variety of complementary algorithms with convergence guarantees.**
>
> * **Algorithmic Diversity:** We provide **three algorithms (direct eigenvalue decomposition, exponential iteration, and Hutchinson estimation) to compute the spectral norm.** This reduces our dependence on any single algorithmic assumption.
>
> * **Controllable Robust Estimation:** Specifically for the Hutchinson algorithm, we **theoretically analyze its convergence in Theorems 4 and 5** and **experimentally verify it in Appendix K.2 (Table 18)**: the estimation error systematically decreases as the number of samples M increases. This demonstrates that our estimation is **reliable and controllable**.
>
> * **Black-Box Verification:** We even estimate using a completely different method (finite difference) **under a black-box setting (Section 3.5)**, and the results are **consistent with the white-box estimation (Table 6)**, further confirming the stability of the ranking.
>
> * **Stability of the Estimation:** The robustness metric we define is **the average value calculated over the entire test set S (Equation 6)**. This mean itself is a statistic; it **reduces random fluctuations** by aggregating a large number of data points, providing a stable estimate.
>
> * **Negligible approximation error:** Theorem 1 approximates the KL divergence using the FIM metric. In **Appendix D**, we analyzed through theoretical analysis and experimental verification that **this approximation error is negligible (see Tables 9 and 10).**
>
>
> **In summary**, the theoretical or experimental robust ranking results are **a reliable, robust, and multi-faceted conclusion**, rather than a fragile result that is sensitive to estimation assumptions.

---

> > ### Author Response · Authors · 2025-12-02
> > **Rebuttal 1 to Weakness 5 (Part 2)**
> >
> > **Weakness 5: The reported “robustness ranking” (e.g., DenseNet < VGG < ResNet ≤ ViT) might depend heavily on spectral norm estimation assumptions and lacks uncertainty quantification.**
> >
> > Let's continue our discussion from point 3 above.
> >
> > **4. Theoretical Ranking is Completely Independent of Spectral Norm Estimation Assumptions**
> >
> > * **Source of the Theoretical Ranking:** The ranking DenseNet121 < VGG16 < ResNet18 ≤ ViT-B-16 is directly derived from the mathematical expressions for the upper bounds of $\Vert J_f \Vert_2$ in Table 2.
> >
> > * **The Expressions are Deterministic:** These expressions (such as $\prod \Vert W_{i} \Vert_{2}$ and $\prod(1 + \Vert W_{l}\Vert_{2})$ are deterministic algebraic forms. **They are derived symbolically from network architectures** (such as the chained connections of VGG and the residual blocks of ResNet), using only fundamental properties of matrix norms (triangle inequalities, submultiplicativity), **without involving any numerical computation or stochastic algorithms.**
> >
> > * **Decoupling from "Estimation":** This ranking does not require running any exponential iterations or Hutchinson algorithms, does not require inputting any real data into the model, and does not even require knowing the specific values ​​of the weights $W$. **You only need to know the architectural topology information such as "VGG is a product of layers".** Therefore, this ranking is **an analytical property of the architecture itself** and is **independent of any posterior "estimation assumptions".**
> >
> > **5. Theoretical ordination itself does not require a traditional "uncertainty measure."**
> >
> > * **Regarding the theoretical ordination itself:** The "uncertainty" of a qualitative ordination (A < B) derived mathematically **lies in the rigor and correctness of the derivation process,** unlike statistical estimation which has a "confidence interval." **We have ensured the rigor of the derivation through complete proofs in Appendices G and H.** Therefore, its "determinism" is guaranteed by mathematical logic.
> >
> > * **Regarding the experimental verification:** Although the theoretical ordination is deterministic, **we verified it experimentally using numerical estimation and conducted robustness analyses on it through various methods, as detailed in points 2 and 3 above.**
> >
> > Therefore, summarizing the above five points, our conclusions are **both theoretically rigorous and empirically robust.**

---

> ### Author Response · Authors · 2025-12-01
> **Rebuttal 1 to Question 1**
>
> **Question 1: How stable are the FIM eigenvalue estimates across different random seeds or mini-batches?**
>
> We thank the reviewers for raising this important technical question regarding the robustness of our method. Our algorithm design and experimental results have collectively and strongly demonstrated the stability of our estimate from different perspectives.
>
> **1. (Random seed) The algorithm's design theoretically does not depend on a random seed.**
>
> Our three estimation algorithms are designed to seek stable and convergent solutions.
>
> * **Direct eigenvalue decomposition:**  This algorithm is a deterministic algorithm that directly solves $\Vert F(x) \Vert_2$.
>
> * **Convergence of Power iteration:** The power iteration algorithm (Algorithm 1) is itself a classic algorithm for solving the spectral norm of a matrix. It eventually converges to its true value **regardless of the different initialization vectors caused by the random seed.**
>
> * **Convergence of the Hutchinson estimate:** As stated in Theorem 4, the estimate converges to the true value with high probability as the number of samples M increases. This means that by choosing a sufficiently large M, we can obtain a stable estimate with small variance, **independent of sampling randomization caused by the random seed.**
>
> **2. (Random seed) Direct experimental evidence: Quantification of robustness measure variance**
>
> In Appendix K.3, we analyze the variance of several robustness estimates in detail: we estimate the mean and variance by **running the experiment five times with different random seeds**. The variances of PGD and CLEVER are close to zero, and we list them in Tables 21 and 22, while **the variance of our robustness measures is almost zero**.
>
>
> **3. (Random seed) Indirect experimental evidence: Consistency in ranking across models and datasets**
>
> * **Consistency in Table 4:** On CIFAR-10, the four models exhibit a consistent robustness ranking across six different metrics.
>
>  * **Stability in Tables 3 and 5:** In Table 3, we compare the clean model and the adversarially trained model; the metric variations are large and as expected. In Table 5, the robustness ranking of the same model across different datasets is also intuitive.
>
> These clear, interpretable, and theoretically consistent patterns can only emerge if the metrics $\Vert F(x) \Vert_{2}$ themselves are stable.
>
>
> **4. (Min-batches) The setting of mini-batches does not affect the final robustness metric.**
>
> The robustness estimate of the model on the test dataset depends on every point in the dataset. This is similar to **how the selection or size of the mini-batches does not affect the estimation of classification accuracy on the test dataset.**
>
>
> **In summary**, within our research framework, the estimation of FIM eigenvalues ​​is highly stable, and its fluctuations are far from affecting our final conclusions. **The reproducibility and consistency of our main experimental results** are the best proof of this.

---

> ### Author Response · Authors · 2025-12-01
> **Rebuttal 1 to Question 2**
>
> **Question 2: Can the proposed robustness metric predict generalization to out-of-distribution shifts, or is it limited to adversarial perturbations?**
>
> Thank you to the reviewer for raising this insightful question. **Our answer: Yes**, our metric also applies to the predictive model's performance on out-of-distribution (OOD) generalization.
>
> **1. Theoretical Aspects: The FIM spectral norm is a natural measure of the "uncertainty" of a model.**
>
> Our proposed robustness metric, based on the FIM spectral norm, quantifies the sensitivity of the model's output probability distribution to changes in the input. This property applies not only **to small adversarial perturbations** but also **to larger changes caused by distribution shifts.**
>
> Our metric is directly theoretically related to the generalization ability of OOD, and the key lies in recognizing that **the spectral norm of FIM is closely related to the uncertainty of model predictions.**
>
> Considering the following definition of FIM
> $$
> F(x) = \sum_{k = 1}^K p(y_{k}|x)[\nabla_{x} \log p(y_{k}|x)\nabla_{x} \log p(y_{k}|x)]
> $$
> Let the model be a softmax output.
> $$
> p(y = k|x) = \frac{e^{z_{k}}}{\sum_{l = 1}^K e^{z_{l}}},\quad z_{k} = \theta_{k}^T x.
> $$
> its gradient is
> $$
> \nabla_{x} \log p(y = k|x) = \theta_{k} - \sum_{l = 1}^K p(y = l|x)\theta_{l}
> $$
> * **For in-distribution (ID) samples:** the model typically makes high-confidence (non-uniform) predictions, i.e., for some $p_k \approx 1$. In this case, $\Vert F(x)\Vert_2$ tends to 0. This corresponds to the model's "deterministic" state on familiar data.
> $$
> \nabla_{x} \log p(y = k|x) \approx \theta_{k} - \theta_{k} = 0, \quad \nabla_{x} \log p(y = l|x) \approx \theta_{l} - \theta_{k}, l \neq k.
> $$
> We have
> $$
> F(x) \approx 1 \times 0 + \sum_{l \neq k} 0 \times (\theta_{l} - \theta_{k}) = 0.
> $$
> So $\lambda_{\max}(F(x)) = 0$.
>
>
> * **For out-of-distribution (OOD) samples:** because the model has never seen this type of data before, it cannot definitively classify it, and therefore tends to output an approximately uniform distribution, i.e., $p_k \approx 1/K$. In this case, $\Vert F(x) \Vert_2$ will increase. This corresponds to the model's "uncertain" state on unfamiliar data.
> $$
> \nabla_{x} \log p(y = k|x) = \theta_{k} - \frac{1}{K}\sum_{l = 1}^K \theta_{l} = \theta_{k} - \bar{\theta}
> $$
> $$
> F(x) = \frac{1}{K}\sum_{k = 1}^K (\theta_{k} - \bar{\theta})(\theta_{k} - \bar{\theta})^T = \text{cov}(\theta).
> $$
> So $\lambda_{\max}(F(x)) \ge 0$.  $F(x)$ represents the covariance matrix of the parameters. **The spectral norm $\lambda_{\max}(F(x))$ is significantly positive when the parameters $\theta_k$ differ greatly between classes.**
>
> **Conclusion:** Therefore, a high $\Vert F(x) \Vert_2$ value can serve as a valid signal that the input sample x may come from an OOD distribution. Our metric, through the spectral norm of the FIM, **naturally captures the model's transition from "deterministic" to "uncertain" state.**
>
> **2. Experimental Support: Cross-Dataset Comparison Provides Preliminary Validation**
>
> While the paper focuses on adversarial robustness, our experiments in Section 4.5 (Robustness of the Same Model on Different Datasets) **provide preliminary, supporting evidence for the predictive potential of OOD.**
>
> When we apply the same model (e.g., ResNet18) to different test sets (e.g., Medical Data vs. CIFAR-100), we are essentially **evaluating the model's behavior under varying degrees of "distribution shift."** Our metric successfully ranks these datasets according to the model's sensitivity to them (as shown in Table 5). This demonstrates that the metric can quantify the differences in model stability when facing different data manifolds, which is central to evaluating OOD generalization ability.
>
>
> **In summary,** we believe that although current experiments focus on combating robustness, our proposed FIM spectral norm metric is conceptually and practically far more advanced than that. **We offer a unified analytical framework for understanding the root causes of model vulnerability to any type of input variation.** This provides a novel and powerful perspective and tool for **addressing the core challenge of OOD generalization.**

---

> ### Author Response · Authors · 2025-12-01
> **Rebuttal 1 to Question 3**
>
> **Questions 3: How does the metric behave under adversarial training; does it improve monotonically with increased robustness?**
>
> Our proposed robustness metric shows a significant and consistent improvement with adversarial training. **Table 3 in Section 4.3 provides the most direct and compelling evidence for this.**
>
> We train a standard model $M_{\text{clean}}$ on the CIFAR-10 dataset using a ResNet18 model, while simultaneously employing the CW adversarial training method to obtain a robustly enhanced model $M_{\text{CW}}$. Our metric $R_{\text{norm}}$ significantly decreases from 2.38 for the clean model to 0.82 for the adversarial model, a reduction of 65.55%; correspondingly, $R_{\text{spec}}$ significantly increases from 46.46 to 186.46.

---

> ### Author Response · Authors · 2025-12-01
> **Rebuttal 1 to Question 4**
>
> **Question 4: Could this framework be extended to parameter-space Fisher information (e.g., K-FAC approximations) for joint training and robustness evaluation?**
>
>
> We are very grateful to the reviewers for this insightful and forward-thinking suggestion. Combining our framework with Fisher information matrices in the parameter space (such as K-FAC) to achieve joint optimization is **undoubtedly an exciting vision.** After exploring its feasibility in depth, we found that although **direct transfer presents significant theoretical and engineering challenges**, this suggestion precisely points to a key direction for future research and helps us more **clearly define the boundaries and value of our current work.**
>
>
> **1. K-FAC approximations for FIM Matrix**
>
> The basic idea behind K-FAC for training models is to use natural gradients for gradient descent updates, where the second-order gradient is precisely the FIM matrix $F(x)$.
> $$
> \theta_{t + 1} = \theta_{t} - \alpha F(x)^{-1} \nabla_{\theta} L(\theta)
> $$
> According to the Kronecker decomposition, each block $F_l$ of $F(x)$ can be approximately represented as
> $$
> F_{l} \approx \mathbb{E}[a_{l - 1}a_{l - 1}^T] \otimes \mathbb{E}[g_{l} g_{l}^T],
> $$
> where $a_{l - 1}$ is the activation input of the (l-1)th layer, $g_l$ is the gradient of the lth layer, and $\otimes$ is the Kronecker product. When updating parameters, the algorithm updates the model parameters layer by layer. Therefore, when updating the parameters at the $l$-th layer, only $F_l$ needs to be used.
>
> However, our robustness estimation requires estimating the largest eigenvalue of the FIM, the relationship between FIM matrix $F$ and $F_l$ is as follows:
> $$
> F = [F_{ij}],i = 1,2,\cdots, L; j = 1,2,\cdots,L.
> $$
> **where the diagonal block $F_{ll} = F_l$ and the off-diagonal block $F_{ij}(i \neq j)$ represent the interaction of inter-layer parameters ($\mathbb{E}[\nabla_{W_{i}}\mathcal{L} \nabla_{W_{}{j}}\mathcal{L}^T]$). K-FAC cannot effectively estimate off-diagonal blocks $F_{ij}(i \neq j)$.**
>
> Without considering off-diagonal blocks $F_{ij}(i \neq j)$, the calculated spectral norm estimate will have a very large estimation error ($F_{ij}(i \neq j) = 0$). Furthermore, **retaining the variables $a_l$ and $g_l$ from each layer will consume a significant amount of storage space for large and deep networks.**
>
> **2. K-FAC approximations for Joint Training**
>
> Our FIM calculation is performed on the test set, **not during training on the training set**. Therefore, joint training and robust estimation are unlikely.
>
>
> **In summary,** the reviewer's suggestions prompted us to **examine in depth the fundamental differences in computation and application between the two Fisher information matrices (input space $F(x)$ and parameter space $F_{\theta}$)**. However, directly using parameter space FIM approximations such as K-FAC to efficiently estimate the required input space FIM spectral norm **faces significant theoretical obstacles (lack of off-diagonal block information) and engineering bottlenecks (storage and computational costs).**

---

> ### Author Response · Authors · 2025-12-01
> **Rebuttal 1 to Question 5**
>
> **Question 5: What is the computational bottleneck in practice - is it gradient computation or the spectral estimation step?**
>
>
> Thank you to the reviewer for raising this crucial question about computational efficiency. Through theoretical analysis and experimental verification, we can clearly answer that **in most practical scenarios, gradient calculation is the primary overall computational bottleneck.**
> However, for the spectral estimation step itself,  we have provided several efficient options to address different situations.
>
>
> **1. Theoretical complexity analysis: Gradient calculation dominates the overall overhead.**
>
> * **Gradient computation overhead:** As described in Section 3.3 of the paper, computing matrix $Q = [q_{1},\cdots,q_{K}]$ ($q_{k} = \nabla_{x} \log p(y_{k}|x)$) requires a backpropagation for each class k. While the gradients for all classes can be computed in a single, carefully designed backpropagation, its complexity is typically **proportional to the forward propagation and the number of model parameters, i.e., O(Model Size)**. For large models (such as ViT and ResNet), this is the most time-consuming part.
>
> * **Spectral estimation overhead:** This depends on the algorithm used, but the key is that we utilize the low-rank structure of FIM. The core operation of the exponentiation iteration is $Q(\Lambda(Q^Tb_t))$, with a complexity of $O(TdK)$, where $d$ is the input dimension and $K$ is the number of classes. In the Hutchinson algorithm, the core operation for each sampling is also a matrix-vector multiplication, with a complexity of $O(dK)$.
>
> * **Key comparison:** The cost of gradient computation is related to the number of model parameters, while the cost of spectral estimation is related to $d×K$. For modern deep models, the number of parameters is typically much larger than $d×K$ (e.g., ResNet50 has approximately **25 million parameters**, while CIFAR-100 has $d×K=3×224×224×100$ ($\approx$ **15 million**). Therefore, **gradient computation dominates the total time.**
>
> **2. Experimental Evidence: Comparative Analysis of Runtime**
>
> Table 7 in the paper provides direct experimental evidence. ResNet18 takes 267.13 seconds in white-box testing on CIFAR100, while gradient-based Lipschitz constant estimation takes 131.09 seconds.
>
> * **$L(x)$ gradient estimation**: The computation of $L(x)$ is essentially the computation of the gradient norm once, representing a baseline cost for pure gradient computation.
>
> * **Spectral Estimation:** Our method takes slightly longer than $L(x)$, but is on the same order of magnitude. This additional time difference is approximately the cost of the spectral estimation step (power iteration/Hutchinson).
>
> **In summary**, while gradient computation is an unavoidable major cost, **we have made sufficient optimizations in the spectral estimation step**, making the entire evaluation process **feasible and efficient for large models and datasets.**

---

### Note · Authors · 2026-01-27

I have read and agree with the venue's withdrawal policy on behalf of myself and my co-authors.

---

### Meta-Review · Area_Chair_92kK · 2026-01-05

**Summary:**

This paper proposes a robustness metric grounded in a theoretical framework of Fisher information. It proposes the reciprocal of the Fisher Information Matrix's principal eigenvalue as metric, and provides a theoretical derivation through the Mahalanobis distance defined through the Fischer Information Matrix. Reviewers agree that the framework's validity is supported by theoretical analysis, deriving spectral bounds and theoretical connections between FIM, Jacobian variance, the Lipschitz constant, and the CLEVER score. The practical implementation uses power iterations and the Hutchinson's approximation for efficiency. Evaluations are provided for DenseNet, VGG, ResNet and transformer architectures on several small scale datasets mnist, cifar and TinyImageNet as well as on a dataset of medical images.

**Reviewer Concerns:**

The reviewers appreciate the paper's principled contributions, such as the established connection between KL divergence, Fisher information, and robustness.
Concerns include the following:
1. It is not evident how well the proposed metric can predict real-world robustness beyond controlled settings
2. Potential run-time overhead that limits applicability to larger datasets --> the authors rebut by explaining why the run-time can be higher
3. Robustness comparisons might be inconsistent across datasets.  --> the authors provide empirical results, on medical data and tinyImageNet.
4. Lack of uncertainty quantification
5. Dependence on spectral norm
6. Data dependence rather than attack dependence in the metric

Overall, te authors provide extensive replies to the different reviewer concerns, arguing partly through empirical evidence. Other aspects such as potential computational overheads are attributed to a potential gain in information. My impression is that the later argument is not fully convincing and not supported by empirical evidence.

**Reviewer Scores:**

Reviewer DQ8w: 2 --> asks many detailed questions that are mostly addressed. The reviewer might have increased the score to 4.
Reviewer nwWH: 6 --> carefully positive review and the conscerns are mostly addressed. I would expect this reviewer to stick to the carefully positive rating.
Reviewer jG9h: 4 --> most of the practical questions and concerns regarding literature are addressed in my opinion. The rating might be increased to 6.
Reviewer ERiY: 2 --> the rebuttal argues that the paper provides an alternative to an empirical evaluation such as in robust bench - however, it does not provide any detailed analysis of how the two would compare. I doubt the reviewer would increase the score but if so, the rating is unlikely to be higher than 4.

---

### Decision · Program_Chairs · 2026-01-26

Reject